# Low-Rank Thinning

**Annabelle Michael Carrell** [1]   **Albert Gong** [2]   **Abhishek Shetty** [3]   **Raaz Dwivedi** [2]   **Lester Mackey** [4]

## Abstract

The goal in thinning is to summarize a dataset using a small set of representative points. Remarkably, sub-Gaussian thinning algorithms like Kernel Halving and Compress can match the quality of uniform subsampling while substantially reducing the number of summary points. However, existing guarantees cover only a restricted range of distributions and kernel-based quality measures and suffer from pessimistic dimension dependence. To address these deficiencies, we introduce a new low-rank analysis of sub-Gaussian thinning that applies to any distribution and any kernel, guaranteeing high-quality compression whenever the kernel or data matrix is approximately low-rank. To demonstrate the broad applicability of the techniques, we design practical sub-Gaussian thinning approaches that improve upon the best known guarantees for approximating attention in transformers, accelerating stochastic gradient training through reordering, and distinguishing distributions in near-linear time.

## 1. Introduction

This work is about thinning, finding a small set of representative points to accurately summarize a larger dataset. Here, we use the term "dataset" liberally to refer to any collection of points, be they experimental observations (as in Sec. 6), stochastic gradients (as in Sec. 5), or key-value pairs (as in Sec. 4). State-of-the-art thinning techniques provably improve upon uniform subsampling but only for restricted classes of kernel-based quality measures and with pessimistic dependence on the data dimension (see, e.g.,

Harvey & Samadi, 2014; Phillips & Tai, 2020; Alweiss et al., 2021; Dwivedi & Mackey, 2024; 2022; Shetty et al., 2022; Li et al., 2024). We introduce a new analysis for sub-Gaussian thinning algorithms that applies to any kernel and shows that one can efficiently identify a better-than-uniform set of representative points whenever the kernel or data matrix is nearly low-rank. This opens the door to a variety of impactful applications including approximate dot-product attention in transformers, accelerating model training through gradient reordering, and distinguishing distributions with deep kernels in near-linear time.

In Sec. 2, we introduce our formal definition of thinning, two kernel-based measures of thinning quality, and a suite of candidate thinning algorithms. Sec. 3 presents our main theorem relating thinning quality to low-rank properties of a dataset and its induced kernel matrix. In Sec. 4, we translate the problem of approximating attention into a thinning problem and develop a practical solution, Thinformer, with state-of-the-art quality guarantees. In Sec. 5, we develop a thinned stochastic gradient reordering rule that provably accelerates training and bridges the theory-practice gaps left open by prior work. Finally, in Sec. 6 we derive new and improved power guarantees for practical thinned hypothesis tests that distinguish distributions in near-linear time. Each section also includes its own discussion of related work.

---

[1]University of Cambridge [2]Cornell Tech [3]MIT [4]Microsoft Research New England. Correspondence to: Annabelle Carrell <ac2411@cam.ac.uk>, Albert Gong <agong@cs.cornell.edu>, Abhishek Shetty <ashetty1995@gmail.com>, Raaz Dwivedi <dwivedi@cornell.edu>, Lester Mackey <lmackey@microsoft.com>.

**Notation.** For each $n \in \mathbb{N}$ and $a, b \in \mathbb{R}$, we define $[n] \triangleq \{1, \ldots, n\}$, $a \wedge b \triangleq \min(a, b)$, and $a \vee b \triangleq \max(a, b)$. We let $\|\mathbf{A}\|_{\mathrm{op}}$, $\|\mathbf{A}\|_{\mathrm{max}}$, and $\|\mathbf{A}\|_{2,\infty}$ respectively represent the maximum singular value, absolute entry, and row Euclidean norm of a matrix $\mathbf{A}$ and let $\lambda_r(\mathbf{K})$ denote the $r$-th largest eigenvalue of a suitable matrix $\mathbf{K}$. We also define the Euclidean norm balls $\mathbb{B}^m \triangleq \{\boldsymbol{u} \in \mathbb{R}^m : \|\boldsymbol{u}\|_2 \leq 1\}$ and $\mathbb{B}^m(R) \triangleq R\mathbb{B}^m$ for each $m \in \mathbb{N}$ and $R > 0$. For an event $\mathcal{E}$ and an integrable random variable $X$, we define $\mathbb{E}_{\mathcal{E}}[X] \triangleq \mathbb{E}[X \cdot \mathbf{1}[\mathcal{E}]]$. We write $a_n \leq \widetilde{O}(b_n)$ to mean $a_n \leq b_n \operatorname{polylog}(n)$.

## 2. Sub-Gaussian Thinning

We begin with a formal definition of our problem setting. Consider a fixed collection of $n_{\mathrm{in}}$ input points $\mathcal{X}_{\mathrm{in}}$ belonging to a potentially larger universe of datapoints

Table 1: **Examples of $(\mathbf{K}, \nu, \delta)$-sub-Gaussian thinning algorithms.** For input size $n_{\text{in}}$, output size $n_{\text{out}} \geq \sqrt{n_{\text{in}}}$, and $\|\mathbf{K}\|_{\max} = 1$ we report each sub-Gaussian parameter $\nu$ and runtime up to constants independent of $(n_{\text{in}}, n_{\text{out}}, \delta, \mathbf{K})$.

| **Algorithm** | SUBSAMPLING | KH$(\delta)$ | KH-COMPRESS$(\delta)$ | GS-THIN | GS-COMPRESS |
| :---: | :---: | :---: | :---: | :---: | :---: |
| | Prop. B.1 | Prop. B.2 | Prop. B.5 | Prop. B.6 | Prop. B.10 |
| **Sub-Gaussian parameter $\nu$** | $\dfrac{1}{\sqrt{n_{\text{out}}}}$ | $\dfrac{\sqrt{\log(n_{\text{out}}/\delta)}}{n_{\text{out}}}$ | $\dfrac{\sqrt{\log(n_{\text{out}})\log(n_{\text{out}}/\delta)}}{n_{\text{out}}}$ | $\dfrac{1}{n_{\text{out}}}$ | $\dfrac{\sqrt{\log(n_{\text{out}})}}{n_{\text{out}}}$ |
| **Runtime** | $n_{\text{out}}$ | $n_{\text{in}}^2$ | $n_{\text{out}}^2$ | $n_{\text{in}}^3$ | $n_{\text{out}}^3$ |

$\mathcal{X} \triangleq \{\boldsymbol{x}_1, \ldots, \boldsymbol{x}_n\}$. The aim of a thinning algorithm is to select $n_{\text{out}}$ points from $\mathcal{X}_{\text{in}}$ that together accurately summarize $\mathcal{X}_{\text{in}}$. This is formalized by the following definition.

**Definition 1** (**Thinning algorithms**). *A thinning algorithm* ALG *takes as input* $\mathcal{X}_{\text{in}}$ *and returns a possibly random subset* $\mathcal{X}_{\text{out}}$ *of size* $n_{\text{out}}$. *We denote the input and output empirical distributions by* $\mathbb{P}_{\text{in}} \triangleq \frac{1}{n_{\text{in}}} \sum_{\boldsymbol{x} \in \mathcal{X}_{\text{in}}} \boldsymbol{\delta_x}$ *and* $\mathbb{P}_{\text{out}} \triangleq \frac{1}{n_{\text{out}}} \sum_{\boldsymbol{x} \in \mathcal{X}_{\text{out}}} \boldsymbol{\delta_x}$ *and define the induced probability vectors* $\boldsymbol{p}_{\text{in}}, \boldsymbol{p}_{\text{out}} \in \boldsymbol{\Delta}_{n-1}$ *over the indices* $[n]$ *by*

$$\boldsymbol{p}_{\text{in},i} = \frac{\mathbf{1}[\boldsymbol{x}_i \in \mathcal{X}_{\text{in}}]}{n_{\text{in}}} \;\; and \;\; \boldsymbol{p}_{\text{out},i} = \frac{\mathbf{1}[\boldsymbol{x}_i \in \mathcal{X}_{\text{out}}]}{n_{\text{out}}} \;\; for \; all \;\; i \in [n].$$

*When* $\mathcal{X} \subset \mathbb{R}^d$, *we use* $\mathbf{X} \triangleq [\boldsymbol{x}_1, \ldots, \boldsymbol{x}_n]^\top \in \mathbb{R}^{n \times d}$ *to denote the input point matrix so that*

$$\mathbb{E}_{\boldsymbol{x} \sim \mathbb{P}_{\text{in}}}[\boldsymbol{x}] = \mathbf{X}^\top \boldsymbol{p}_{\text{in}} \;\; and \;\; \mathbb{E}_{\boldsymbol{x} \sim \mathbb{P}_{\text{out}}}[\boldsymbol{x}] = \mathbf{X}^\top \boldsymbol{p}_{\text{out}}.$$

We will make use of two common measures of summarization quality.

**Definition 2** (**Kernel MMD and max seminorm**). *Given two distributions* $\mu, \tilde{\mu}$ *and a reproducing kernel* $\mathbf{k}$ *(Steinwart & Christmann, 2008, Def. 4.18), the associated kernel maximum mean discrepancy (MMD) is the worst-case difference in means for functions in the unit ball* $\mathbb{B}_{\mathbf{k}}$ *of the associated reproducing kernel Hilbert space:*

$$\text{MMD}_{\mathbf{k}}(\mu, \tilde{\mu}) \triangleq \sup_{f \in \mathbb{B}_{\mathbf{k}}} |\mathbb{E}_{\boldsymbol{x} \sim \mu} f(\boldsymbol{x}) - \mathbb{E}_{\boldsymbol{x} \sim \tilde{\mu}} f(\boldsymbol{x})|.$$

*When* $\mu = \mathbb{P}_{\text{in}}$ *and* $\tilde{\mu} = \mathbb{P}_{\text{out}}$ *as in Def. 1 and* $\mathbf{K} \triangleq (\mathbf{k}(\boldsymbol{x}_i, \boldsymbol{x}_j))_{i,j=1}^n \in \mathbb{R}^{n \times n}$ *denotes the induced kernel matrix, then the MMD can be expressed as a Mahalanobis distance between* $\boldsymbol{p}_{\text{in}}$ *and* $\boldsymbol{p}_{\text{out}}$:

$$\text{MMD}_{\mathbf{k}}(\mathbb{P}_{\text{in}}, \mathbb{P}_{\text{out}}) = \sqrt{(\boldsymbol{p}_{\text{in}} - \boldsymbol{p}_{\text{out}})^\top \mathbf{K}(\boldsymbol{p}_{\text{in}} - \boldsymbol{p}_{\text{out}})}$$
$$\triangleq \text{MMD}_{\mathbf{K}}(\boldsymbol{p}_{\text{in}}, \boldsymbol{p}_{\text{out}}).$$

*For any indices* $\mathcal{I} \subseteq [n]$, *we further define the* kernel max seminorm (KMS)

$$\|\mathbf{K}(\boldsymbol{p}_{\text{in}} - \boldsymbol{p}_{\text{out}})\|_{\mathcal{I}} \triangleq \max_{i \in \mathcal{I}} |\boldsymbol{e}_i^\top \mathbf{K}(\boldsymbol{p}_{\text{in}} - \boldsymbol{p}_{\text{out}})|. \quad (1)$$

Notably, when the input points lie in $\mathbb{R}^d$ and $\mathbf{k}(\boldsymbol{x}_i, \boldsymbol{x}_j)$ is the linear kernel $\langle \boldsymbol{x}_i, \boldsymbol{x}_j \rangle$ (so that $\mathbf{K} = \mathbf{X}\mathbf{X}^\top$), MMD measures the Euclidean discrepancy in datapoint means between the input and output distributions:

$$\text{MMD}_{\mathbf{K}}(\boldsymbol{p}_{\text{in}}, \boldsymbol{p}_{\text{out}}) = \|\mathbf{X}^\top \boldsymbol{p}_{\text{in}} - \mathbf{X}^\top \boldsymbol{p}_{\text{out}}\|_2.$$

A common strategy for bounding the error of a thinning algorithm is to establish its sub-Gaussianity.

**Definition 3** (**Sub-Gaussian thinning algorithm**). *We write* ALG $\in \mathcal{G}_{\nu,\delta}(\mathbf{K})$ *and say* ALG *is* $(\mathbf{K}, \nu, \delta)$-*sub-Gaussian, if* ALG *is a thinning algorithm,* $\mathbf{K}$ *is a symmetric positive semidefinite (SPSD) matrix,* $\nu > 0$, $\delta \in [0, 1)$, *and there exists an event* $\mathcal{E}$ *with probability at least* $1 - \delta/2$ *such that, the input and output probability vectors satisfy*

$$\mathbb{E}_{\mathcal{E}}[\exp(\langle \boldsymbol{u}, \mathbf{K}(\boldsymbol{p}_{\text{in}} - \boldsymbol{p}_{\text{out}})\rangle)] \leq \exp\left(\frac{\nu^2}{2} \boldsymbol{u}^\top \mathbf{K} \boldsymbol{u}\right), \forall \boldsymbol{u} \in \mathbb{R}^n.$$

Here, the sub-Gaussian parameter $\nu$ controls the summarization quality of the thinning algorithm, and we see from Tab. 1 that a variety of practical thinning algorithms are $(\mathbf{K}, \nu, \delta)$-sub-Gaussian for varying levels of $\nu$.

### 2.1. Examples of sub-Gaussian thinning algorithms

Perhaps the simplest sub-Gaussian thinning algorithm is *uniform subsampling*: by Prop. B.1, selecting $n_{\text{out}}$ points from $\mathcal{X}_{\text{in}}$ uniformly at random (without replacement) is $(\mathbf{K}, \nu, 0)$-sub-Gaussian with $\nu = \sqrt{\|\mathbf{K}\|_{\max}}/\sqrt{n_{\text{out}}}$. Unfortunately, uniform subsampling suffers from relatively poor summarization quality. As we prove in App. B.1.1, its root-mean-squared MMD and KMS are both $\Omega(1/\sqrt{n_{\text{out}}})$ meaning that $n_{\text{out}} = 10000$ points are needed to achieve 1% relative error.

**Proposition 1** (**Quality of uniform subsampling**). *For any* $\mathcal{I} \subseteq [n]$, *a uniformly subsampled thinning satisfies*

$$\mathbb{E}[\text{MMD}_{\mathbf{K}}^2(\boldsymbol{p}_{\text{in}}, \boldsymbol{p}_{\text{out}})] = \frac{1}{n_{\text{out}}} \frac{n_{\text{in}} - n_{\text{out}}}{n_{\text{in}} - 1} C_{\mathbf{K}} \;\; and$$
$$\mathbb{E}[\|\mathbf{K}(\boldsymbol{p}_{\text{in}} - \boldsymbol{p}_{\text{out}})\|_{\mathcal{I}}^2] \geq \frac{1}{n_{\text{out}}} \frac{n_{\text{in}} - n_{\text{out}}}{n_{\text{in}} - 1} \max_{i \in \mathcal{I}} C_{\mathbf{K} \boldsymbol{e}_i \boldsymbol{e}_i^\top \mathbf{K}}$$

*for any SPSD* $\mathbf{K}$ *with* $C_{\mathbf{K}} \triangleq \sum_{i=1}^n \boldsymbol{p}_{\text{in},i} \mathbf{K}_{ii} - \boldsymbol{p}_{\text{in}}^\top \mathbf{K} \boldsymbol{p}_{\text{in}}$.

Fortunately, uniform subsampling is not the only sub-Gaussian thinning algorithm available. For example, the Kernel Halving (KH($\delta$)) algorithm of Dwivedi & Mackey (2024) provides a substantially smaller sub-Gaussian parameter, $\nu = O(\sqrt{\log(n_{\text{out}}/\delta)}/n_{\text{out}})$, in $n_{\text{in}}^2$ time by selecting one out of every two points and biasing selection toward the point that yields smaller approximation error. Similarly, the KH-COMPRESS($\delta$) algorithm of Shetty et al. (2022, Ex. 3) delivers $\nu = O(\sqrt{\log(n_{\text{out}}) \log(n_{\text{out}}/\delta)}/n_{\text{out}})$ in only $n_{\text{out}}^2$ time by halving and concatenating coresets of geometrically increasing size until the target output size is met. We derive simplified versions of these algorithms with identical sub-Gaussian constants in Apps. B.2 and B.5 and a linear-kernel variant (LKH($\delta$)) with $n_{\text{in}}d$ runtime in App. B.3. To round out our set of examples, we show in App. B.6.1 that two new thinning algorithms based on the Gram-Schmidt walk of Bansal et al. (2018) yield even smaller $\nu$ at the cost of increased runtime. We call these algorithms Gram-Schmidt Thinning (GS-THIN) and GS-COMPRESS.

## 3. Low-rank Sub-Gaussian Thinning

One might hope that the improved sub-Gaussian constants of Tab. 1 would also translate into improved quality metrics. Our main result, proved in App. C, shows that this is indeed the case whenever the inputs are approximately low-rank.

**Theorem 1** (**Low-rank sub-Gaussian thinning**). *Fix any $\delta' \in (0,1)$, $r \leq n$, and $\mathcal{I} \subseteq [n]$. If $\text{ALG} \in \mathcal{G}_{\nu,\delta}(\mathbf{K})$, then the following bounds hold individually with probability at least $1 - \delta/2 - \delta'$:*

$$\text{MMD}_{\mathbf{K}}^2(\boldsymbol{p}_{\text{in}}, \boldsymbol{p}_{\text{out}}) \leq \nu^2 \left[ e^2 r + e \log(\tfrac{1}{\delta'}) \right]$$
$$+ \lambda_{r+1} \left( \tfrac{1}{n_{\text{out}}} - \tfrac{1}{n_{\text{in}}} \right) \quad and \quad (2)$$

$$\|\mathbf{K}(\boldsymbol{p}_{\text{in}} - \boldsymbol{p}_{\text{out}})\|_{\mathcal{I}} \leq \nu D_{\mathcal{I}} \sqrt{2 \log(\tfrac{2|\mathcal{I}|}{\delta'})}. \quad (3)$$

*Here, $\lambda_j$ denotes the $j$-th largest eigenvalue of $\mathbf{K}$, $\lambda_{n+1} \triangleq 0$, and $D_{\mathcal{I}} \triangleq \max_{i \in \mathcal{I}} \sqrt{\mathbf{K}_{ii}}$.*

*Suppose that, in addition, $\mathcal{X} \subset \mathbb{R}^d$ and $|\mathbf{K}_{il} - \mathbf{K}_{jl}| \leq L_{\mathbf{K}} \|\boldsymbol{x}_i - \boldsymbol{x}_j\|_2$ for some $L_{\mathbf{K}} > 0$ and all $i,j \in \mathcal{I}$ and $l \in \text{supp}(\boldsymbol{p}_{\text{in}})$. Then, with probability at least $1 - \delta/2 - \delta'$,*

$$\|\mathbf{K}(\boldsymbol{p}_{\text{in}} - \boldsymbol{p}_{\text{out}})\|_{\mathcal{I}} \leq \nu D_{\mathcal{I}} \sqrt{2 \log(4/\delta')} (1 + \tfrac{32}{\sqrt{3}})$$
$$+ \nu D_{\mathcal{I}} 32 \sqrt{\tfrac{2}{3} \text{rank}(\mathbf{X}_{\mathcal{I}}) \log(\tfrac{3e^2 R_{\mathcal{I}} L_{\mathbf{K}}}{D_{\mathcal{I}}^2 \wedge (R_{\mathcal{I}} L_{\mathbf{K}})})} \quad (4)$$

*for $R_{\mathcal{I}} \triangleq \max_{i \in \mathcal{I}} \|\boldsymbol{x}_i\|_2$ and $\mathbf{X}_{\mathcal{I}} \triangleq [\boldsymbol{x}_i]_{i \in \mathcal{I}}^\top$.*

Let us unpack the three components of this result. First, Thm. 1 provides a high-probability $O(\nu \sqrt{\log(|\mathcal{I}|)})$ bound (3) on the KMS for any kernel and any sub-Gaussian thinning algorithm on any space. In particular, the non-uniform

algorithms of Tab. 1 all enjoy $O(\log(n_{\text{out}}) \sqrt{\log(|\mathcal{I}|)}/n_{\text{out}})$ KMS, a significant improvement over the $\Omega(1/\sqrt{n_{\text{out}}})$ KMS of uniform subsampling. The bound (3) follows from the sub-Gaussianity of the thinning algorithm (Def. 3) and the union bound over $\pm \boldsymbol{e}_i$ for each $i \in \mathcal{I}$.

For datapoints in $\mathbb{R}^d$, Thm. 1 also provides a refined $O(\nu \sqrt{\text{rank}(\mathbf{X}_{\mathcal{I}}) \log(R_{\mathcal{I}} L_{\mathbf{K}})})$ bound (4) on KMS. For bounded data, this trades an explicit dependence on the number of query points $|\mathcal{I}|$ for a rank factor that is never larger (and sometimes significantly smaller) than $d$. This refinement follows from a more elaborate chaining argument that frames $(\boldsymbol{e}_i^\top \mathbf{K}(\boldsymbol{p}_{\text{in}} - \boldsymbol{p}_{\text{out}}))_{i \in \mathcal{I}}$ as a sub-Gaussian process (Lem. C.5) and uses the Lipschitzness of $K$ to control its entropy integral. We will make use of these results when approximating dot-product attention in Sec. 4.

Notably, Thm. 3.1 of Phillips & Tai (2020) implies that any thinning algorithm must incur at least $\Omega(\sqrt{d}/n_{\text{out}})$ KMS error for some dataset in $\mathbb{R}^d$ and many common kernels. Meanwhile, our Tab. 1 and Thm. 1 imply that GS-THIN has $\nu = O(1/n_{\text{out}})$ and hence KMS $O(\sqrt{d}/n_{\text{out}})$. Taken together, these results imply that no algorithm can have sub-Gaussian constant $\nu = o(1/n_{\text{out}})$ and that GS-THIN enjoys minimax rate-optimal KMS and a minimax rate-optimal sub-Gaussian constant.

Finally, Thm. 1 provides an $O(\nu \sqrt{r} + \sqrt{\lambda_{r+1}/n_{\text{out}}})$ high-probability bound on kernel MMD, where the approximate rank parameter $r$ can be freely optimized. We establish this result by projecting $\mathbf{K}^{1/2}(\boldsymbol{p}_{\text{in}} - \boldsymbol{p}_{\text{out}})$ onto the first $r$ eigenvectors of $\mathbf{K}$, bounding the residual error in terms of $\lambda_{r+1}$, and bounding the projection magnitude with high probability using the sub-Gaussianity of ALG and a union bound over a finite cover of a Euclidean ball in $\mathbb{R}^r$.

When $\mathbf{K} = (\mathbf{k}(\boldsymbol{x}_i, \boldsymbol{x}_j))_{i,j=1}^n$ is generated by a finite-rank kernel $\mathbf{k}$, like a linear kernel $\langle \boldsymbol{x}_i, \boldsymbol{x}_j \rangle$, a polynomial kernel $(1 + \langle \boldsymbol{x}_i, \boldsymbol{x}_j \rangle)^p$, or a random Fourier feature kernel (Rahimi & Recht, 2007), this guarantee becomes $O(\nu)$ and improves upon uniform subsampling whenever $\nu = o(1/\sqrt{n_{\text{out}}})$. In this case, the non-uniform algorithms of Tab. 1 all enjoy $O(\log(n_{\text{out}})/n_{\text{out}})$ MMD, a significant improvement over the $\Omega(1/\sqrt{n_{\text{out}}})$ MMD of uniform subsampling. We will revisit this finite-rank setting when studying stochastic gradient acceleration strategies in Sec. 5.

More generally, Thm. 1 guarantees improved MMD even for full-rank $\mathbf{K}$, provided that the eigenvalues of $\mathbf{K}$ decay sufficiently rapidly. For example, optimizing over the approximate rank parameter $r$ yields an $O(\nu \log^{p/2}(n_{\text{out}}))$ bound under exponential eigenvalue decay $\lambda_{r+1} = O(n e^{-cr^{1/p}})$ and an $O(\nu^{\frac{p}{p+1}}(\frac{n}{n_{\text{out}}})^{\frac{1}{2(p+1)}})$ bound under polynomial eigenvalue decay $\lambda_{r+1} = O(n/r^p)$. Fortunately, some of the most commonly-used kernels generate kernel matrices with rapid eigenvalue decay.

For example, the popular Gaussian kernel on $\mathbb{R}^d$,

$$\text{GAUSS}(\eta): \ \mathbf{k}(x,y) = \exp(-\eta\|x-y\|_2^2) \text{ for } \eta > 0, \quad (5)$$

generates $\mathbf{K} = (\mathbf{k}(\boldsymbol{x}_i, \boldsymbol{x}_j))_{i,j=1}^n$ satisfying

$$\lambda_{r+1} \le ne^{-\frac{d}{2e}r^{1/d}\log\left(\frac{dr^{1/d}}{4e^2\eta R^2}\right)} \ \text{ for } \ (2e)^d \le r < n \quad (6)$$

whenever $\mathcal{X} \subset \mathbb{B}^d(R)$ (Altschuler et al., 2019, Thm. 3). Combined with Thm. 1, this fact immediately yields an MMD guarantee for each algorithm in Tab. 1. We present a representative guarantee for $\text{KH}(\delta)$.

**Corollary 1** (**Gaussian MMD of KH**). *If $\mathcal{X}_{\text{in}} \subset \mathbb{B}^d(R)$ for $R > 0$, then $\text{KH}(\delta)$ with $\mathbf{k} = \text{GAUSS}(\eta)$, and $n = n_{\text{in}}$ delivers*

$$\text{MMD}_{\mathbf{K}}^2(\boldsymbol{p}_{\text{in}}, \boldsymbol{p}_{\text{out}}) \le$$
$$O\Big(\frac{\log(n_{\text{out}}/\delta)}{n_{\text{out}}^2}\Big(\Big(\frac{\log(n_{\text{out}})\vee(R^2\eta)}{d}\Big)^d + \log(\frac{1}{\delta'})\Big)\Big)$$

*with probability at least $1 - \delta/2 - \delta'$.*

The proof in App. D provides a fully explicit and easily computed bound on the Gaussian MMD. Under the same assumptions, the distinct analysis of Dwivedi & Mackey (2022, Thm. 2, Prop. 3) provides a squared MMD bound of size $\Theta\Big(\frac{\log(n_{\text{out}}/\delta)}{n_{\text{out}}^2}\Big(\frac{\log^{d+1}(n_{\text{out}})R^d\eta^{d/2}}{(\log\log(n_{\text{out}}))^d} + \log(\frac{1}{\delta'})\Big)\Big)$. Notably, Cor. 1 improves upon this best known $\text{KH}(\delta)$ guarantee whenever the datapoint radius $R = O(\log n_{\text{out}})$, a property that holds almost surely for any bounded, sub-Gaussian, or subexponential data sequence (see Dwivedi & Mackey, 2024, Prop. 2).

Altschuler et al. (2019, Thm. 4) additionally showed that Gaussian kernel matrix eigenvalues satisfy

$$\lambda_{r+1} \le ne^{-cr^{2/(5d^\star)}} \quad \text{for} \quad 1 \le r < n \quad (7)$$

for a constant $c$ independent of $\mathcal{X}$ when $\mathcal{X}$ belongs to a smooth compact manifold of dimension $d^\star < d$. In this case, our low-rank analysis yields adaptive MMD guarantees that scale with the potentially much smaller intrinsic dimension $d^\star$. We use Thm. 1 to prove the first such intrinsic-dimension guarantee for $\text{KH}(\delta)$ in App. E.

**Corollary 2** (**Intrinsic Gaussian MMD of KH**). *If $\mathcal{X}_{\text{in}}$ lies on a smooth manifold $\Omega \subset \mathbb{B}^d$ of dimension $d^\star < d$ (Assump. E.1), then $\text{KH}(\delta)$ with $\mathbf{k} = \text{GAUSS}(\eta)$, and $n = n_{\text{in}}$ delivers*

$$\text{MMD}_{\mathbf{K}}^2(\boldsymbol{p}_{\text{in}}, \boldsymbol{p}_{\text{out}}) \le O\Big(\frac{\log(\frac{n_{\text{out}}}{\delta})}{n_{\text{out}}^2}\Big(\Big(\frac{\log(n_{\text{out}})}{c}\Big)^{\frac{5d^\star}{2}} + \log(\frac{1}{\delta'})\Big)\Big)$$

*with probability at least $1 - \frac{\delta}{2} - \delta'$ for $c$ independent of $\mathcal{X}_{\text{in}}$.*

In Sec. 6, we will use Cors. 1 and 2 to establish new guarantees for distinguishing distributions in near-linear time.

With our core theory in hand, we now turn our attention to a series of impactful applications.

# 4. Approximating Attention

We will first use our analysis to accelerate attention approximation in transformers. Dot-product attention lies at the heart of the transformer neural network architecture that has revolutionized natural language processing, computer vision, and speech recognition over the last decade (Vaswani et al., 2017; Dosovitskiy et al., 2021; Dong et al., 2018). Given a collection of query, key, and value vectors $(\boldsymbol{q}_i, \boldsymbol{k}_i, \boldsymbol{v}_i)_{i=1}^n$ each in $\mathbb{R}^d$, dot-product attention computes the *softmax matrix*

$$\mathbf{T} \triangleq \text{ATTENTION}((\boldsymbol{q}_i)_{i=1}^n, (\boldsymbol{k}_j, \boldsymbol{v}_j)_{j=1}^n) \triangleq \mathbf{D}^{-1}\mathbf{A}\mathbf{V} \quad (8)$$

for $\mathbf{A}_{ij} \triangleq \exp(\frac{\langle \boldsymbol{q}_i, \boldsymbol{k}_j\rangle}{\sqrt{d}}), \mathbf{D} = \text{diag}(\mathbf{A}\mathbf{1}_n)$, and $\mathbf{V}_{ij} \triangleq \boldsymbol{v}_{ij}$.

While attention has enjoyed unprecedented success in capturing long-range dependencies amongst datapoints, its computation is expensive, requiring $\Theta(d\,n^2)$ time to construct and multiply the matrix $\mathbf{A}$. This quadratic-time bottleneck has inspired a plethora of practical approximate attention mechanisms (e.g., Kitaev et al., 2020; Choromanski et al., 2021; Chen et al., 2021), but, to our knowledge, only two guarantee accurate reconstruction of the softmax matrix $\mathbf{T}$ (Zandieh et al., 2023; Han et al., 2024).[1] In this section, we design a new fast attention approximation based on sub-Gaussian thinning and derive guarantees that improve upon the prior art.

### 4.1. Thinning attention in theory

---
**Algorithm 1:** Thinformer

**Input:** Queries, keys, and values $(\boldsymbol{q}_i, \boldsymbol{k}_i, \boldsymbol{v}_i)_{i=1}^n$ in $\mathbb{R}^d$, $n_{\text{out}}$

// Define key-value attention kernel

$\mathbf{k}_{\text{att}}((\tilde{\boldsymbol{k}}, \tilde{\boldsymbol{v}}), (\tilde{\boldsymbol{k}}', \tilde{\boldsymbol{v}}')) \triangleq \exp(\langle\tilde{\boldsymbol{k}}, \tilde{\boldsymbol{k}}'\rangle)\langle\tilde{\boldsymbol{v}}, \tilde{\boldsymbol{v}}'\rangle$

// Thin augmented key-value pairs using $\mathbf{k}_{\text{att}}$

$v_{\max} \leftarrow \max_{i\in[n]}\|\boldsymbol{v}_i\|_\infty$; $(\tilde{\boldsymbol{k}}_i, \tilde{\boldsymbol{v}}_i)_{i=1}^n \leftarrow (\boldsymbol{k}_i/d^{\frac{1}{4}}, (\boldsymbol{v}_i, v_{\max}))_{i=1}^n$

$\mathcal{X}_{\text{out}} \leftarrow \text{KH-COMPRESS}(0.5)(\mathcal{X}_{\text{in}} = (\tilde{\boldsymbol{k}}_i, \tilde{\boldsymbol{v}}_i)_{i=1}^n, \mathbf{k}_{\text{att}}, n_{\text{out}})$

// Return exact attention on selected key-value subset

**return** $\widehat{\mathbf{T}} \triangleq \text{ATTENTION}((\boldsymbol{q}_i)_{i=1}^n, \{(\boldsymbol{k}, \boldsymbol{v}) : (\tilde{\boldsymbol{k}}, \tilde{\boldsymbol{v}}) \in \mathcal{X}_{\text{out}}\})$

---

Alg. 1 summarizes our new *Thinformer* module. At its heart is a new key-value attention kernel $\mathbf{k}_{\text{att}}$ that mimics the special structure of the softmax matrix $\mathbf{T}$. Alg. 1 uses the attention kernel and a high-quality thinning algorithm, KH-COMPRESS(0.5), to subselect key-value pairs and then computes exact attention (8) for the key-value subset. In total, this requires only $O(d\,n_{\text{out}}^2)$ time to run KH-COMPRESS(0.5) and $O(d\,n\,n_{\text{out}})$ time to compute

---
[1]A third remarkable work (Alman & Song, 2024) establishes upper and lower bounds for attention approximation but without a practical implementation.

Table 2: **Practical approximations with guarantees.** For each approximation $\widehat{\mathbf{T}} \in \mathbb{R}^{n \times d}$ to the softmax matrix $\mathbf{T}$ (8), we report, up to a constant factor, the best worst-case error guarantee for $\|\widehat{\mathbf{T}} - \mathbf{T}\|_{\max}$ given $O(d\, n^{1+a})$ running time and $\gamma$-bounded (9) queries and keys. Here, the ratio $\|\mathbf{V}\|_{\mathrm{op}}/\|\mathbf{V}\|_{2,\infty}$ lies in $[1, \sqrt{n}]$ and $\tau = 0.173 + o(1)$.

| Approximation | Guarantee |
|---|---|
| **Thinformer** | $\dfrac{n^{2\gamma}\sqrt{d\log(n\|\mathbf{V}\|_{\max})\log n}}{n^a} \cdot \|\mathbf{V}\|_{2,\infty}$ |
| **KDEformer** | $\dfrac{n^{2\gamma+\frac{\tau}{2}(1+\frac{\gamma}{2})}}{n^{a/2}} \cdot \|\mathbf{V}\|_{\mathrm{op}}$ |
| **HyperAttention** | $\dfrac{n^{\frac{17\gamma}{3}}(\log n)^{\frac{1}{6}}}{n^{a/6}} \cdot \|\mathbf{V}\|_{\mathrm{op}}$ |

ATTENTION with $n$ queries and $n_{\mathrm{out}}$ key-value pairs. In contrast, computing the exact softmax matrix $\mathbf{T}$ with standard matrix multiplication requires $\Theta(d\, n^2)$ time. Our next result, proved in App. F, shows that Alg. 1 also admits a strong quality guarantee for approximating $\mathbf{T}$.

**Theorem 2** (**Quality of Thinformer**). *With probability at least $\frac{1}{2}$, Thinformer (Alg. 1) yields*

$$\|\widehat{\mathbf{T}} - \mathbf{T}\|_{\max} \leq \frac{c\exp(\frac{2R^2}{\sqrt{d}})\|\mathbf{V}\|_{2,\infty}\sqrt{\log_2(n_{\mathrm{out}})\log(8n_{\mathrm{out}}\log_2\frac{n_{\mathrm{in}}}{n_{\mathrm{out}}})}}{n_{\mathrm{out}}}$$

*for* $c \triangleq \frac{128}{\sqrt{3}}\sqrt{(d+1)\log(3e^2(\frac{R^2}{\sqrt{d}}+2)\|\mathbf{V}\|_{\max})} + \sqrt{\log(8)}(4+\frac{128}{\sqrt{3}})$ *and* $R \triangleq \max_{i\in[n]}\max(\|\boldsymbol{k}_i\|_2, \|\boldsymbol{q}_i\|_2)$.

To put this result into context, let us compare with the existing guarantees for practical attention approximation, summarized in Tab. 2. Under the $\gamma$-boundedness assumption,

$$\max_{i\in[n]}\max(\|\boldsymbol{k}_i\|_2^2, \|\boldsymbol{q}_i\|_2^2) \leq \gamma\sqrt{d}\log n, \quad (9)$$

the KDEformer approximation (Zandieh et al., 2023, Cor. 3.6) with $\tau = 0.173 + o(1)$, the HyperAttention approximation (Han et al., 2024, Thm. 1) with no masking, and the Thinformer approximation (Thm. 2) guarantee the $\|\widehat{\mathbf{T}} - \mathbf{T}\|_{\max}$ bounds of Tab. 2 with $O(dn^{1+a})$ runtime and probability at least $\frac{1}{2}$. The Thinformer guarantee exhibits four improvements over its predecessors. First, it establishes a significantly faster error decay rate ($n^{-a}$ versus $n^{-a/2}$ or $n^{-a/6}$) for a given subquadratic runtime $n^{1+a}$. Second, it reduces the dependence on the error inflation factor $\gamma$. Third, like the HyperAttention guarantee, it eliminates all dependence on the KDEformer penalty parameter $\tau$. Finally, it reduces dependence on the value matrix by a factor of $\frac{\|\mathbf{V}\|_{\mathrm{op}}}{\|\mathbf{V}\|_{2,\infty}} \in [1, \sqrt{n}]$.

Put otherwise, with bounded $\|\mathbf{V}\|_{2,\infty}$, $\widehat{\mathbf{T}}_{\mathrm{thin}}$ can provide consistent (i.e., $\|\widehat{\mathbf{T}}_{\mathrm{thin}} - \mathbf{T}\|_{\max} \to 0$ as $n \to \infty$) subquadratic estimation whenever $\gamma$ is bounded away from $1/2$

and guarantee, for example, $O(\frac{1}{\sqrt{n}})$ error in $\widetilde{O}(dn^{\frac{3}{2}+2\gamma})$ time. In contrast, the $\widehat{\mathbf{T}}_{\mathrm{kde}}$ and $\widehat{\mathbf{T}}_{\mathrm{hyp}}$ bounds require quadratic runtime to guarantee $O(\frac{1}{\sqrt{n}})$ error in the best case ($\|\mathbf{V}\|_{\mathrm{op}} = O(1)$) and cannot guarantee consistent subquadratic estimation in the worst case ($\|\mathbf{V}\|_{\mathrm{op}} = \Omega(\sqrt{n})$).

### 4.2. Thinning attention in practice

To gauge the practical effectiveness of Alg. 1, we recreate the benchmark Tokens-To-Token Vision Transformer (T2T-ViT) and BigGAN image generation experiments of Zandieh et al. (2023). In the T2T-ViT experiment, attention approximations are scored on their ImageNet classification accuracy and computational expense when used as drop-in replacements for the two most expensive attention layers in a pretrained T2T-ViT neural network (Yuan et al., 2021). In the BigGAN experiment, approximations are scored on their computational expense and two popular measures of image generation quality, the Frechet Inception Distance (FID, Heusel et al., 2017) and Inception Score (IS, Salimans et al., 2016). Using the exact implementations and settings provided by Zandieh et al. (2023), we benchmark our PyTorch implementation of Thinformer against exact attention and four leading attention approximations: Performer (Choromanski et al., 2021), Reformer (Kitaev et al., 2020), ScatterBrain (Chen et al., 2021), and KDEformer.

In Tab. 3, we find that Thinformer ($\mathfrak{g} = 2$) provides the highest Top-1 accuracy on the ImageNet 2012 validation set (Russakovsky et al., 2015), while running faster than all of the alternatives. In Tab. 4, Thinformer ($\mathfrak{g} = 2$) yields better FID and IS than all of the alternatives while running significantly faster than exact, KDEformer, Reformer, and ScatterBrain. Performer runs faster still but at the expense of substantially worse FID and IS. The final attention call of Thinformer can also be combined with optimized attention implementations like FlashAttention (Dao et al., 2022; Dao, 2024) to further reduce the time and memory footprint. We provide PyTorch code replicating this experiment at https://github.com/microsoft/thinformer and supplementary experiment details in App. L.1.

## 5. Faster SGD Training

We now turn to a second application, accelerating training through gradient reordering. To train a machine learning model parameterized by $\boldsymbol{w} \in \mathbb{R}^d$, a standard approach is to minimize the empirical risk $f(\boldsymbol{w}) \triangleq \frac{1}{n}\sum_{i=1}^n f_i(\boldsymbol{w})$ using stochastic gradient descent (SGD) updates,

$$\boldsymbol{w}^{k+\frac{i}{n}} = \boldsymbol{w}^{k+\frac{i-1}{n}} - \alpha\nabla f_{\pi_k(i)}(\boldsymbol{w}^{k+\frac{i-1}{n}}), \quad (10)$$

for each epoch $k \in [K]$ and datapoint $i \in [n]$. Here, $\alpha > 0$ is a step size, each $f_i$ is a datapoint-specific loss function,

Table 3: **Quality of T2T-ViT attention approximations on ImageNet.** We report mean Top-1 accuracy $\pm 1$ standard deviation across five random seeds and mean forward pass runtime $\pm 1$ standard deviation across 50 batches of 64 images.

| Attention Algorithm | Top-1 Accuracy (%) | Layer 1 Runtime (ms) | Layer 2 Runtime (ms) |
|---|---|---|---|
| Exact | 82.55 ± 0.00 | 18.48 ± 0.12 | 1.40 ± 0.01 |
| Performer | 80.56 ± 0.30 | 2.54 ± 0.01 | 0.60 ± 0.01 |
| Reformer | 81.47 ± 0.06 | 7.84 ± 0.03 | 1.53 ± 0.01 |
| KDEformer | 82.00 ± 0.07 | 5.39 ± 0.03 | 2.28 ± 0.03 |
| Scatterbrain | 82.05 ± 0.08 | 6.86 ± 0.02 | 1.55 ± 0.03 |
| Thinformer (Ours) | 82.18 ± 0.05 | 2.06 ± 0.01 | 0.54 ± 0.00 |

Table 4: **Quality of BigGAN attention approximations for image generation.** We report Frechet Inception Distance (FID) with the ImageNet validation set, Inception Scores (IS), and mean forward pass runtime ± 1 standard deviation across 10 batches of 32 images. A lower FID or higher IS indicates better image generation quality.

| Attention Algorithm | FID ($\downarrow$) | IS ($\uparrow$) | Runtime (ms) |
|---|---|---|---|
| Exact | 32.18 | 58.37 ± 4.21 | 5.79 ± 0.02 |
| Performer | 33.58 | 38.07 ± 3.43 | 2.29 ± 0.01 |
| Reformer | 72.23 | 19.14 ± 2.09 | 12.14 ± 0.02 |
| KDEformer | 30.70 | 56.91 ± 4.16 | 6.74 ± 0.46 |
| ScatterBrain | 38.47 | 36.84 ± 2.90 | 3.12 ± 0.02 |
| Thinformer (Ours) | 30.54 | 57.12 ± 3.96 | 2.69 ± 0.01 |

and $\pi_k$ is a permutation of $[n]$ representing the order in which datapoints are processed in the $k$-th epoch.

---

**Algorithm 2:** Thinned Reordering

**Input:** Stochastic gradients $(\boldsymbol{x}_i^k \triangleq \nabla f_{\pi_k(i)}(\boldsymbol{w}^{k+\frac{i-1}{n}}))_{i=1}^n$, prior ordering $\pi_k$, thinning algorithm ALG

// Select half of points using linear kernel
$\mathcal{X}_{\text{out}}^k \leftarrow \text{ALG}(\mathcal{X}_{\text{in}} = (\boldsymbol{x}_i^k)_{i=1}^n, n_{\text{out}} = \frac{n}{2}, \mathbf{k}(\boldsymbol{x}, \boldsymbol{y}) = \langle \boldsymbol{x}, \boldsymbol{y} \rangle)$
$\Pi \leftarrow []; \quad \Pi' \leftarrow []$            // Initialize empty start and end lists
**for** $i = 1, \ldots, n$ **do**
$\quad$ $\Pi$.append($\pi_k(i)$) if $\boldsymbol{x}_i^k \in \mathcal{X}_{\text{out}}^k$ else $\Pi'$.prepend($\pi_k(i)$)
**end**
**return** $\pi_{k+1} = \text{concatenate}(\Pi, \Pi')$

---

Typically, one selects the orderings $\pi_k$ uniformly at random, but recent work has demonstrated faster convergence using non-uniform, adaptively selected orderings. Specifically, Lu et al. (2022); Cooper et al. (2023) show that any sufficiently accurate thinning algorithm can be efficiently transformed into a reordering rule that improves the convergence rate of SGD by a substantial $\widetilde{O}(n^{-1})$ factor. Their approach, distilled in Alg. 2, uses an elegant construction of Harvey & Samadi (2014, Thm. 10) to translate a high-quality thinning of stochastic gradients into a higher-quality reordering. However, these prior studies leave two

problems unaddressed.

First, while the established convergence rates of Lu et al. (2022) nearly match the minimax lower bounds for permuted SGD algorithms (Cha et al., 2023, Thm. 4.5), a multiplicative gap of size $\Theta(d)$ remains in the worst case. This led Cha et al. (2023) to declare, "It is an open problem whether there exists a permutation-based SGD algorithm that gives a dimension-free upper bound while maintaining the same dependency on other factors."

Second, Lu et al. (2022) carry out their analysis using the self-balancing walk (SBW) thinning algorithm of Alweiss et al. (2021) but find its overhead to be too high in practice. Hence, in all experiments they instead employ a greedy thinning algorithm that often works well in practice but is not covered by their analysis.

### 5.1. Bridging the dimension gap

To address the first problem, we derive a new guarantee for SGD with LKH reordering that replaces the typical $\Theta(d)$ penalty with a soft notion of rank.

**Definition 4** ($\epsilon$-rank). *The $\epsilon$-rank, $\text{rank}_\epsilon(\mathbf{X})$, of a matrix $\mathbf{X}$ is the number of singular values greater than $\epsilon$.*

**Theorem 3** (**LKH-SGD convergence**). *Suppose that, for*

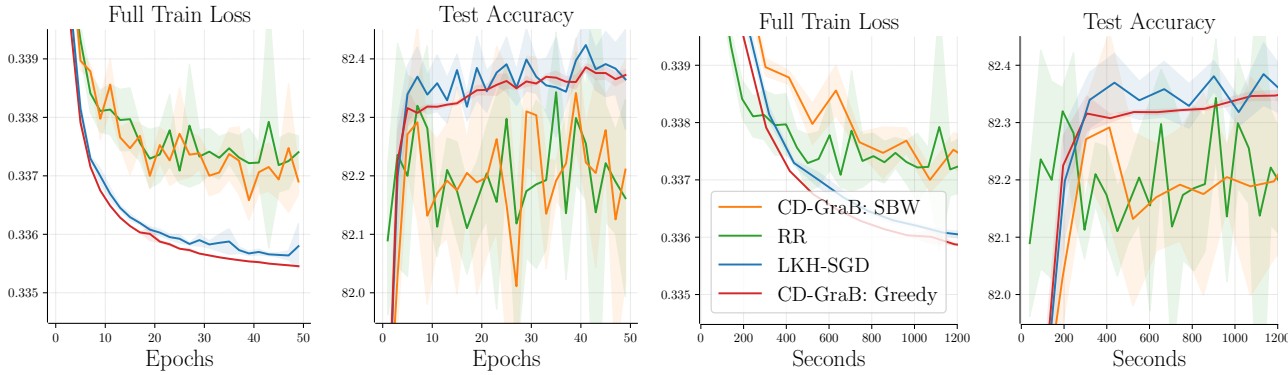

Figure 1: **Train and test convergence trajectories for mortgage classification with reordered SGD variants.** We display mean values $\pm 1$ standard deviation across 5 random seeds. See Sec. 5.2 for more details.

*all $i \in [n]$ and $\boldsymbol{w}, \boldsymbol{v} \in \mathbb{R}^d$, the losses $f$ and $f_i$ satisfy*

$\|\nabla f_i(\boldsymbol{w}) - \nabla f(\boldsymbol{w})\|_2^2 \leq \sigma^2$ **(bounded noise)**,

$\|\nabla f_i(\boldsymbol{w}) - \nabla f_i(\boldsymbol{v})\|_2 \leq L\|\boldsymbol{w} - \boldsymbol{v}\|_2$ **(smoothness)**, *and*

$f(\boldsymbol{w}) - f^\star \leq \frac{1}{2\mu}\|\nabla f(\boldsymbol{w})\|_2^2$ **(PL)** *for $f^\star \triangleq \inf_{\boldsymbol{v} \in \mathbb{R}^d} f(\boldsymbol{v})$.*

*Then, with probability at least $\frac{1}{2}$, SGD (10) with $\text{LKH}(\frac{1}{2K})$ reordering (Alg. 2) and step size $\alpha$ given in App. G satisfies*

$$f(\boldsymbol{w}_K) - f^\star \leq \widetilde{O}\left(\frac{r}{n^2 K^2}\right) \quad for$$

$r \triangleq \max_{k \in [K]} \text{rank}_{\epsilon_k}([\boldsymbol{x}_1^k, \ldots, \boldsymbol{x}_n^k])$, $\bar{\boldsymbol{x}}^k \triangleq \frac{1}{n}\sum_{i=1}^n \boldsymbol{x}_i^k$,

*and* $\epsilon_k \triangleq \max_{i \in [n]} \frac{\sqrt{9e\log(4Kn\log(en/2))\log(4Kn)}\|\boldsymbol{x}_i^k - \bar{\boldsymbol{x}}^k\|_2}{\sqrt{n}}$.

The proof of Thm. 3 in App. G simply uses Thm. 1 to bound the thinning quality of $\text{LKH}(\frac{1}{2K})$ and then adapts the prior SGD analysis of Cooper et al. (2023). Notably, the standard practice of *random reshuffling*, i.e., SGD with uniform reordering, can only guarantee a significantly slower $\Omega(\frac{1}{nK^2})$ rate under these assumptions (Rajput et al., 2020, Thm. 2), while Lu et al. (2022, Thm. 4) implies a similar but dimension-dependent $\widetilde{O}(\frac{d}{n^2 K^2})$ rate for SBW reordering. Thm. 3 matches the minimax lower bound of Cha et al. (2023, Thm. 4.5) up to the $\epsilon$-rank parameter and shows that dimension dependence can be avoided when the gradient update matrices $[\boldsymbol{x}_1^k, \ldots, \boldsymbol{x}_n^k]$ are low-rank, or, more generally, $\epsilon = O(\frac{\log(Kn)}{\sqrt{n}})$-approximable by low-rank matrices.

### 5.2. Bridging the theory-practice gap

Two criticisms levied by Lu et al. (2022) against the SBW algorithm were the need to estimate the maximum Euclidean norm of any possible gradient vector in advance and the need to tune its free hyperparameter. $\text{LKH}(\frac{1}{2K})$ has neither of these drawbacks as it automatically adapts

to the scale of each input and has no hyperparameters to tune. Moreover, with a linear kernel, $\text{LKH}(\frac{1}{2K})$ can be run online in $O(nd)$ time. Hence, $\text{LKH}(\frac{1}{2K})$ is a promising substitute for the greedy thinning of Lu et al. (2022); Cooper et al. (2023). Indeed, when we recreate the Home Mortgage Disclosure Act logistic regression experiment of Cooper et al. (2023) with a single worker (Fig. 1), we find that LKH-SGD strongly outperforms the standard practice of random reshuffling (RR) and the theoretically justified but overly conservative CD-GraB: SBW variant. In addition, LKH-SGD matches the state-of-the-art test accuracy of CD-GraB: Greedy and lags only slightly in terms of training convergence.

The accelerated convergence rate of LKH-SGD over the standard slow SGD rate of RR provides a direct verification of the Thm. 3 guarantee, and we further verify in Fig. L.1 that the singular values of the gradient update matrices drop off steeply, resulting in relatively small $\epsilon_k$-ranks (see Fig. L.1). See `https://github.com/microsoft/khsgd` for PyTorch code replicating this experiment and App. L.2 for supplementary experiment details.

## 6. Cheap Two-Sample Testing

Our final application is two-sample testing, determining whether two datasets are drawn from the same underlying distribution. We observe independent samples $\mathcal{X} \triangleq (\boldsymbol{x}_i)_{i=1}^m$ and $\mathcal{Y} \triangleq (\boldsymbol{y}_j)_{j=1}^n$ from the unknown distributions $\mathbb{P}$ and $\mathbb{Q}$ respectively, and we seek to accept or reject the null hypothesis that $\mathbb{P} = \mathbb{Q}$. Standard kernel MMD tests tackle this task by computing the empirical MMD

$\text{MMD}_\mathbf{k}(\mathbb{P}_\text{in}, \mathbb{Q}_\text{in})$ for $\mathbb{P}_\text{in}, \mathbb{Q}_\text{in} \triangleq \frac{1}{m}\sum_{\boldsymbol{x} \in \mathcal{X}}\boldsymbol{\delta_x}, \frac{1}{n}\sum_{\boldsymbol{y} \in \mathcal{Y}}\boldsymbol{\delta_y}$

for an appropriate kernel $\mathbf{k}$ and rejecting the null hypothesis whenever $\text{MMD}_\mathbf{k}(\mathbb{P}_\text{in}, \mathbb{Q}_\text{in})$ is sufficiently large (Gretton et al., 2012). Such tests are prized both for their broad

applicability and for their high discriminating *power*, that is, their probability of rejecting the null when $\mathbb{P} \neq \mathbb{Q}$. A standard way to summarize the power properties of a test is through its *detectable separation rate*.

**Definition 5** (**Detectable separation rate**). *We say a two-sample test has* detectable separation rate $\epsilon_{\mathbf{k},m,n}$ *if, for any detection probability* $1 - \beta \in (0,1)$, *there exists a constant* $c_{\mathbf{k},\beta} > 0$ *such that the test has power at least* $1 - \beta$ *of rejecting the null whenever* $\mathrm{MMD}_{\mathbf{k}}(\mathbb{P}, \mathbb{Q}) \geq c_{\mathbf{k},\beta} \cdot \epsilon_{\mathbf{k},m,n}$.

Standard MMD tests can detect distributional differences on the order of $\epsilon_{\mathbf{k},m,n} = \frac{1}{\sqrt{\min(m,n)}}$ (Gretton et al., 2012, Cor. 9), and this detectable separation rate is known to be the best possible for MMD tests (Domingo-Enrich et al., 2023, Prop. 2) and minimax optimal for translation invariant kernels (Kim & Schrab, 2023, Thm. 8). However, standard MMD tests also suffer from the $\Theta((m + n)^2)$ time burden of computing the empirical MMD. Recently, Domingo-Enrich et al. (2023) showed that one can improve scalability while preserving power by compressing $\mathbb{P}_{\mathrm{in}}$ and $\mathbb{Q}_{\mathrm{in}}$ using a high-quality thinning algorithm. However, their analysis applies only to a restricted class of distributions and kernels and exhibits a pessimistic dimension dependence on $\mathbb{R}^d$. Here, we offer a new analysis of their Compress Then Test approach that applies to any bounded kernel on any domain and, as an application, develop the first non-asymptotic power guarantees for testing with learned deep neural network kernels.

## 6.1. Low-rank analysis of Compress Then Test

---

**Algorithm 3:** Compress Then Test (CTT)

---

**Input:** Samples $(\mathcal{X}, \mathcal{Y})$, # coresets $s$, compression level $\mathfrak{g}$, kernel $\mathbf{k}$, failure probability $\delta$, # replicates $\mathcal{B}$, level $\alpha$

Partition $\mathcal{X}$ into $s_m = \frac{sm}{m+n}$ equal-sized bins $(\mathcal{X}^{(i)})_{i=1}^{s_m}$
Partition $\mathcal{Y}$ into $s_n = \frac{sn}{m+n}$ equal-sized bins $(\mathcal{Y}^{(i)})_{i=1}^{s_n}$

// Identify coreset of size $n_{\mathrm{out}} = 2^{\mathfrak{g}} \sqrt{\frac{m+n}{s}}$ for each bin

**for** $i = 1, \ldots, s_m$ **do** $\mathbb{P}_{\mathrm{out}}^{(i)} \leftarrow$ KT-COMPRESS$(\delta)(\mathcal{X}^{(i)}, \mathfrak{g}, \mathbf{k})$
**for** $i = 1, \ldots, s_n$ **do** $\mathbb{Q}_{\mathrm{out}}^{(i)} \leftarrow$ KT-COMPRESS$(\delta)(\mathcal{Y}^{(i)}, \mathfrak{g}, \mathbf{k})$

// Compute CORESETMMD test statistic
$$M_{\mathcal{B}+1} \leftarrow \mathrm{MMD}_{\mathbf{k}}(\tfrac{1}{s_m} \textstyle\sum_{i=1}^{s_m} \mathbb{P}_{\mathrm{out}}^{(i)}, \tfrac{1}{s_n} \textstyle\sum_{i=1}^{s_n} \mathbb{Q}_{\mathrm{out}}^{(i)}) \quad (11)$$

// Simulate null by randomly permuting the $s$ coresets $\mathcal{B}$ times
**for** $b = 1, \ldots, \mathcal{B}$ **do**
$\quad (\mathbb{P}_{\mathrm{out},b}^{(i)})_{i=1}^{s_m}, (\mathbb{Q}_{\mathrm{out},b}^{(i)})_{i=1}^{s_n} \leftarrow$ PERMUTE$((\mathbb{P}_{\mathrm{out}}^{(i)})_{i=1}^{s_m}, (\mathbb{Q}_{\mathrm{out}}^{(i)})_{i=1}^{s_n})$
$\quad M_b \leftarrow \mathrm{MMD}_{\mathbf{k}}(\tfrac{1}{s_m} \sum_{i=1}^{s_m} \mathbb{P}_{\mathrm{out},b}^{(i)}, \tfrac{1}{s_n} \sum_{i=1}^{s_n} \mathbb{Q}_{\mathrm{out},b}^{(i)})$
**end**

// Threshold test statistic
$R \leftarrow$ position of $M_{\mathcal{B}+1}$ in an increasing ordering of $(M_b)_{b=1}^{\mathcal{B}+1}$ with ties broken uniformly at random
**return Reject** with prob. $\min(1, \max(0, R - (1 - \alpha)(\mathcal{B} + 1)))$

---

Alg. 3 details the Compress Then Test (CTT) approach of Domingo-Enrich et al. (2023, Alg. 1). Given a coreset count $s \geq 2$, a compression level $\mathfrak{g} \geq 0$, and a nominal level $\alpha \in (0,1)$, CTT divides $\mathcal{X}$ and $\mathcal{Y}$ into datapoint bins of size $n_{\mathrm{in}} \triangleq \frac{m+n}{s}$, thins each bin down to size $n_{\mathrm{out}} \triangleq 2^{\mathfrak{g}} \sqrt{n_{\mathrm{in}}}$ using KT-COMPRESS$(\delta)$ (a refinement of KH-COMPRESS$(\delta)$ detailed in App. H), and uses the thinned coresets to cheaply approximate $\mathrm{MMD}_{\mathbf{k}}(\mathbb{P}_{\mathrm{in}}, \mathbb{Q}_{\mathrm{in}})$ and permuted versions thereof. Domingo-Enrich et al. (2023, (8)) showed that the total runtime of CTT is dominated by

$$O(4^{\mathfrak{g}}(m + n)(s + \log_4(\tfrac{m+n}{s} - \mathfrak{g})))$$

kernel evaluations, yielding a near-linear $O((m + n)\log^c(m+n))$ time algorithm whenever $s = O(\log_4(m+n))$ and $\mathfrak{g} \leq c\log_4\log(m+n)$. Moreover, Prop. 1 of Domingo-Enrich et al. (2023) ensures that CTT has probability at most $\alpha$ of falsely rejecting the null hypothesis.

Our next, complementary result shows that CTT also matches the detectable separation rate of standard MMD tests up to an inflation factor $\mathbf{R}_{\mathbf{k}}/2^{\mathfrak{g}}$ depending on the compression level $\mathfrak{g}$.

**Theorem 4** (**Low-rank analysis of CTT power**). *Suppose the parameters of* CTT *(Alg. 3) satisfy* $m \leq n$,

$$s_m \geq \tfrac{32}{9} \log(\tfrac{2e}{\gamma}), \quad \text{and} \quad \delta = \min(\tfrac{\widetilde{\beta}}{6}, (\tfrac{\widetilde{\beta}}{2})^{1/\lfloor \alpha(\mathcal{B}+1) \rfloor} \tfrac{\alpha}{30es})$$

*for* $\widetilde{\beta} \triangleq \frac{\beta}{1+\beta/2}$ *and* $\gamma \triangleq \frac{\alpha}{4e}(\tfrac{\tilde{\beta}}{4})^{1/\lfloor \alpha(\mathcal{B}+1) \rfloor}$. *Then* CTT *has detectable separation rate (Def. 5)*

$$\epsilon_{\mathbf{k},m,n} = (1 + \mathbf{R}_{\mathbf{k}}/2^{\mathfrak{g}})/\sqrt{m},$$

*where* $\mathbf{R}_{\mathbf{k}}^2$ *denotes the* $(1 - \frac{\widetilde{\beta}}{20s_n})$*-th quantile of*

$$\widehat{\mathbf{R}}_{\mathbf{k}}^2 \triangleq \log(\tfrac{m+n}{s})\log(\tfrac{n}{\beta}) \cdot \quad (12)$$
$$\min_{r \leq 2n_{\mathrm{out}}} \{\|\mathbf{k}\|_{\infty} r \log(\tfrac{n}{\beta}) + (\lambda_{r+1}(\mathbf{K}) + \lambda_{r+1}(\mathbf{K}'))n_{\mathrm{out}}\}.$$

*for* $\mathbf{K} \triangleq (\mathbf{k}(\boldsymbol{x}_i, \boldsymbol{x}_j))_{i,j=1}^m$, $\mathbf{K}' \triangleq (\mathbf{k}(\boldsymbol{y}_i, \boldsymbol{y}_j))_{i,j=1}^n$, *and* $\|\mathbf{k}\|_{\infty} \triangleq \sup_{x,y \in \mathrm{supp}(\mathbb{P}+\mathbb{Q})} |\mathbf{k}(x,y)|$.

The proof in App. I combines the low-rank sub-Gaussian error bounds of Thm. 1 with the generic compressed power analysis of Domingo-Enrich et al. (2023, App. B.1) to yield power guarantees for bounded kernels on any domain. Notably, when $\mathrm{rank}(\mathbf{K})$ and $\mathrm{rank}(\mathbf{K}')$ are bounded or, more generally, $\mathrm{polylog}(n)$ one can choose the compression level $\mathfrak{g} = \Theta(\log_4 \log(m + n))$ to exactly match the optimal quadratic-time detectable separation rates with a near-linear time CTT test. Moreover, the inflation factors remain well-controlled whenever the induced kernel matrices exhibit rapid eigenvalue decay.

As a concrete example, consider the learned deep neural network kernel of Liu et al. (2020),

$$\mathbf{k}_{\text{deep}}(\boldsymbol{x}, \boldsymbol{y}) \triangleq [(1 - \epsilon)\kappa(\phi(\boldsymbol{x}), \phi(\boldsymbol{y})) + \epsilon]q(\boldsymbol{x}, \boldsymbol{y}), \quad (13)$$

where $\phi : \mathbb{R}^d \to \mathbb{R}^{d_{\text{embd}}}$ is a pretrained neural network, $q$ and $\kappa$ are GAUSS($\eta$) kernels (5) on $\mathbb{R}^d$ and $\mathbb{R}^{d_{\text{embd}}}$ respectively, and $\epsilon \in (0, 1)$. This deep kernel generates full-rank kernel matrices (Liu et al., 2020, Prop. 5) but induces exponential eigenvalue decay due to its decomposition as a mixture of Gaussian kernels. Hence, as we show in App. J, CTT with $\mathbf{k}_{\text{deep}}$, $\mathfrak{g} = \Theta(\log_4 \log(m + n))$, and sub-Gaussian inputs matches the detection quality of a quadratic-time MMD test in near-linear time.

**Corollary 3** (**Power of deep kernel CTT**). *Instantiate the assumptions of Thm. 4 with* $\mathbf{k} = \mathbf{k}_{\text{deep}}$ *(13). If the inputs* $(\phi(\boldsymbol{x}_1), \boldsymbol{x}_1, \phi(\boldsymbol{y}_1), \boldsymbol{y}_1)$ *are sub-Gaussian, that is,*

$$\mathbb{E}[e^{c\|(\phi(\boldsymbol{x}_1), \boldsymbol{x}_1, \phi(\boldsymbol{y}_1), \boldsymbol{y}_1)\|_2^2}] < \infty \quad (14)$$

*for some* $c > 0$*, then* CTT *satisfies the conclusions of Thm. 4 with* $d' \triangleq d_{\text{embd}} + d$ *and*

$$\mathbf{R}_{\mathbf{k}_{\text{deep}}} = O(\log^{\frac{d'}{2} + \frac{3}{2}}(\tfrac{n}{\beta})).$$

Moreover, when the input and neural features lie on smooth compact manifolds (as, e.g., in Zhu et al., 2018), the error inflation of CTT adapts to the smaller intrinsic manifold dimension, enabling an improved trade-off between runtime and detection power. See App. K for our proof.

**Corollary 4** (**Power of deep manifold kernel CTT**). *Under the assumptions of Cor. 3, if* $\boldsymbol{x}_1$, $\boldsymbol{y}_1$, $(\boldsymbol{x}_1, \phi(\boldsymbol{x}_1))$, *and* $(\boldsymbol{y}_1, \phi(\boldsymbol{y}_1))$ *belong to smooth compact manifolds (Assump. E.1) with dimension* $d^\star < d'$ *then* CTT *satisfies the conclusions of Thm. 4 with*

$$\mathbf{R}_{\mathbf{k}_{\text{deep}}} = O(\log^{\frac{5d^\star}{4} + \frac{3}{2}}(\tfrac{n}{\beta})).$$

Cors. 3 and 4 follow from explicitly bounding the eigenvalues of the generated deep kernel matrices as in (6) and (7). By Kim & Schrab (2023, Thm. 8), the separation rate of Thm. 4 is minimax optimal up to the inflation factor $\mathbf{R}_{\mathbf{k}}/2^{\mathfrak{g}}$ and, hence, those of Cors. 3 and 4 are minimax optimal up to log factors.

One could alternatively bound the compression error of KT-COMPRESS($\delta$) using the covering number approach of Dwivedi & Mackey (2022, Thm. 2, Prop. 3). In the setting of Cor. 3, the argument of App. J combined with this distinct analysis would yield an alternative error inflation factor $\tilde{\mathbf{R}}_{\mathbf{k}_{\text{deep}}}/2^{\mathfrak{g}}$ with worse dimension dependence,

$$\tilde{\mathbf{R}}_{\mathbf{k}_{\text{deep}}} = \Theta(\log^{\frac{3d'}{4} + 2}(\tfrac{n}{\beta})),$$

and without known adaptivity to an intrinsic manifold dimension.

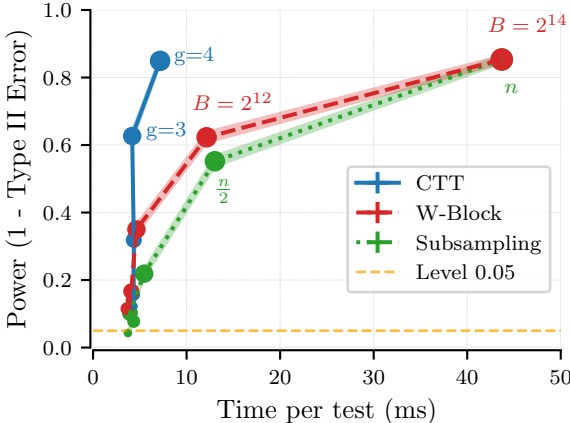

Figure 2: **Time-power trade-off curves for detecting Higgs bosons with deep kernel MMD tests.** We plot mean values $\pm 1$ standard error across 1000 independent trials with level $\alpha = 0.05$ and $\mathcal{B} = 100$ permutations.

### 6.2. Powerful deep kernel testing in near-linear time

To evaluate the practical utility of deep kernel CTT, we follow the Higgs mixture experiment of Domingo-Enrich et al. (2023, Sec. 5) and use the deep kernel training procedure of Liu et al. (2020, Tab. 1). Here, the aim is to distinguish a Higgs boson signal process $\mathbb{P}$ from a background process $\mathbb{Q}$ given $m = n = 16384$ observations, $d = 2$ particle-detector features, and a five-layer fully-connected neural network $\phi$ with softplus activations and embedding dimension $d_{\text{embd}} = 20$.

Fig. 2 compares the time-power trade-off curves induced by three fast kernel testing approaches to this problem: SUBSAMPLING, a standard wild-bootstrap MMD test (Chwialkowski et al., 2014) that simply evaluates empirical $\text{MMD}_{\mathbf{k}_{\text{deep}}}$ using $n_{\text{out}} = m_{\text{out}}$ uniformly subsampled points; W-BLOCK, a wild-bootstrap test that averages $\frac{n}{B}$ subsampled squared $\text{MMD}_{\mathbf{k}_{\text{deep}}}$ estimates based on $n_{\text{out}} = m_{\text{out}} = B$ points (Zaremba et al., 2013); and CTT with $s = 32$ bins and varying $\mathfrak{g}$. We find that the CTT curve dominates that of the alternative methods and matches the power of an exact MMD test (SUBSAMPLING with $n_{\text{out}} = n$) in a fraction of the time. The improvements of CTT over the standard power-runtime trade-off of SUBSAMPLING provides a direct verification of the Thm. 4 guarantee, and we additionally verify in Fig. L.2 that the empirical inflation factor $\widehat{\mathbf{R}}_{\mathbf{k}_{\text{deep}}} = O(\log^5(n))$ in this setting due to approximate low-rankness. See https://github.com/microsoft/deepctt for PyTorch code replicating this experiment and App. L.3 for supplementary experiment details.

## Impact Statement

This work introduced a new analysis of thinning algorithms that adapts to low-rank structures. We exploited this adaptivity to design fast algorithms with strong quality guarantees for three key applications in machine learning: dot-product attention in Transformers, stochastic gradient training in optimization, and deep kernel testing for distinguishing distributions. More broadly, our techniques provide a general framework for reducing computational resource use in machine learning. Such tools have the potential to reduce energy costs and environmental harms from model training, inference, and evaluation and to improve accessibility in resource-constrained settings, all while provably maintaining high quality.

**Acknowledgments** We thank Insu Han, A. Feder Cooper, and Wentao Guo for their assistance with their code bases and datasets.

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

# Appendix Contents

## A. Appendix Notation and Definitions

We often use the shorthand $(a)_+ \triangleq \max(a, 0)$ as well as the shorthand $\mathbf{k}(\mathcal{X}, \mathcal{X})$ to represent the matrix $(\mathbf{k}(\boldsymbol{x}_i, \boldsymbol{x}_j))_{i,j=1}^n$. In addition, for each kernel $\mathbf{k}$, we let $\mathcal{H}_{\mathbf{k}}$ and $\|\cdot\|_{\mathbf{k}}$ represent the associated reproducing kernel Hilbert space (RKHS) and RKHS norm, so that $\mathbb{B}_{\mathbf{k}} = \{f \in \mathcal{H}_{\mathbf{k}} : \|f\|_{\mathbf{k}} \leq 1\}$ and define

$$(\mathbb{P}_{\text{in}} - \mathbb{P}_{\text{out}})\mathbf{k} \triangleq \tfrac{1}{n_{\text{in}}} \sum_{x \in \mathcal{X}_{\text{in}}} \mathbf{k}(\boldsymbol{x}, \cdot) - \tfrac{1}{n_{\text{out}}} \sum_{x \in \mathcal{X}_{\text{out}}} \mathbf{k}(\boldsymbol{x}, \cdot).$$

We also relate our definition of a sub-Gaussian thinning algorithm (Def. 3) to several useful notions of sub-Gaussianity.

**Definition A.1 (Sub-Gaussian vector).** *We say that a random vector $\boldsymbol{w} \in \mathbb{R}^n$ is $(\mathbf{K}, \nu)$-sub-Gaussian on an event $\mathcal{E}$ if $\mathbf{K}$ is SPSD and $\nu > 0$ satisfies*

$$\mathbb{E}_{\mathcal{E}}\left[\exp(\boldsymbol{u}^\top \mathbf{K} \boldsymbol{w})\right] \leq \exp(\tfrac{\nu^2}{2} \cdot \boldsymbol{u}^\top \mathbf{K} \boldsymbol{u}) \quad \textit{for all} \quad \boldsymbol{u} \in \mathbb{R}^n. \tag{15}$$

*If, in addition, the event has probability $1$, we say that $\boldsymbol{w}$ is $(\mathbf{K}, \nu)$-sub-Gaussian.*

Notably, a thinning algorithm is $(\mathbf{K}, \nu, \delta)$-sub-Gaussian if and only if its associated vector $\boldsymbol{p}_{\text{in}} - \boldsymbol{p}_{\text{out}}$ is $(\mathbf{K}, \nu)$-sub-Gaussian on an event $\mathcal{E}$ of probability at least $1 - \delta/2$.

**Definition A.2 (Sub-Gaussian function).** *For a kernel $\mathbf{k}$, we say that a random function $\phi \in \mathcal{H}_{\mathbf{k}}$ is $(\mathbf{k}, \nu)$-sub-Gaussian on an event $\mathcal{E}$ if $\nu > 0$ satisfies*

$$\mathbb{E}_{\mathcal{E}}[\exp(\langle f, \phi \rangle_{\mathbf{k}})] \leq \exp(\tfrac{\nu^2}{2} \cdot \|f\|_{\mathbf{k}}^2) \quad \textit{for all} \quad f \in \mathcal{H}_{\mathbf{k}}. \tag{16}$$

*If, in addition, the event has probability $1$, we say that $\phi$ is $(\mathbf{k}, \nu)$-sub-Gaussian.*

Our next two lemmas show that for finitely-supported signed measures like $\mathbb{P}_{\text{in}} - \mathbb{P}_{\text{out}}$, this notion of functional sub-Gaussianity is equivalent to the prior notion of vector sub-Gaussianity, allowing us to use the two notions interchangeably. Hereafter, we say that $\mathbf{k}$ generates a SPSD matrix $\mathbf{K}$ if $\mathbf{k}(\mathcal{X}, \mathcal{X}) = \mathbf{K}$.

**Lemma A.1 (Functional sub-Gaussianity implies vector sub-Gaussianity).** *In the notation of Def. 3, if $(\mathbb{P}_{\text{in}} - \mathbb{P}_{\text{out}})\mathbf{k}$ is $(\mathbf{k}, \nu)$-sub-Gaussian on an event $\mathcal{E}$ and $\mathbf{k}$ generates $\mathbf{K}$, then the vector $\boldsymbol{p}_{\text{in}} - \boldsymbol{p}_{\text{out}}$ is $(\mathbf{K}, \nu)$-sub-Gaussian on $\mathcal{E}$.*

*Proof.* Suppose $(\mathbb{P}_{\text{in}} - \mathbb{P}_{\text{out}})\mathbf{k}$ is $(\mathbf{k}, \nu)$-sub-Gaussian on an event $\mathcal{E}$, fix a vector $\boldsymbol{u} \in \mathbb{R}^n$, and define the function

$$f_{\boldsymbol{u}} \triangleq \sum_{i=1}^n u_i \mathbf{k}(\cdot, x_i) \in \mathcal{H}_{\mathbf{k}}.$$

By the reproducing property,

$$\boldsymbol{u}^\top \mathbf{K}(\boldsymbol{p}_{\text{in}} - \boldsymbol{p}_{\text{out}}) = \langle f_{\boldsymbol{u}}, (\mathbb{P}_{\text{in}} - \mathbb{P}_{\text{out}})\mathbf{k} \rangle_{\mathbf{k}} \quad \text{and} \quad \|f_{\boldsymbol{u}}\|_{\mathbf{k}}^2 = \boldsymbol{u}^\top \mathbf{K} \boldsymbol{u}. \tag{17}$$

Invoking the representations (17) and the functional sub-Gaussianity condition (16) we therefore obtain

$$\mathbb{E}_{\mathcal{E}}\left[\exp(\boldsymbol{u}^\top \mathbf{K}(\boldsymbol{p}_{\text{in}} - \boldsymbol{p}_{\text{out}}))\right] = \mathbb{E}_{\mathcal{E}}[\exp(\langle f_{\boldsymbol{u}}, (\mathbb{P}_{\text{in}} - \mathbb{P}_{\text{out}})\mathbf{k} \rangle_{\mathbf{k}})] \leq \exp(\|f_{\boldsymbol{u}}\|_{\mathbf{k}}^2 \cdot \tfrac{\nu^2}{2}) = \exp(\boldsymbol{u}^\top \mathbf{K} \boldsymbol{u} \cdot \tfrac{\nu^2}{2}),$$

so that $\boldsymbol{p}_{\text{in}} - \boldsymbol{p}_{\text{out}}$ is $(\mathbf{K}, \nu)$-sub-Gaussian on the event $\mathcal{E}$ as claimed. $\qquad\square$

**Lemma A.2 (Vector sub-Gaussianity implies functional sub-Gaussianity).** *In the notation of Def. 3, if $\boldsymbol{p}_{\text{in}} - \boldsymbol{p}_{\text{out}}$ is $(\mathbf{K}, \nu)$-sub-Gaussian on an event $\mathcal{E}$ and $\mathbf{k}$ generates $\mathbf{K}$, then $(\mathbb{P}_{\text{in}} - \mathbb{P}_{\text{out}})\mathbf{k}$ is $(\mathbf{k}, \nu)$-sub-Gaussian on $\mathcal{E}$.*

*Proof.* Suppose $\boldsymbol{p}_{\text{in}} - \boldsymbol{p}_{\text{out}}$ is $(\mathbf{K}, \nu)$-sub-Gaussian on an event $\mathcal{E}$, fix a function $f \in \mathcal{H}_{\mathbf{k}}$, and consider the set

$$\mathcal{L} \triangleq \left\{ f_{\boldsymbol{u}} \triangleq \sum_{i=1}^n u_i \mathbf{k}(\cdot, x_i) : \boldsymbol{u} \in \mathbb{R}^n \right\}.$$

Since $\mathcal{L}$ is a closed linear subspace of $\mathcal{H}_{\mathbf{k}}$, we can decompose $f$ as $f = f_{\boldsymbol{u}} + f_\perp$, where $\boldsymbol{u} \in \mathbb{R}^n$ and $f_\perp$ is orthogonal to $\mathcal{L}$ (Rudin, 1991, Theorem 12.4), so that

$$\|f\|_{\mathbf{k}}^2 = \|f_{\boldsymbol{u}}\|_{\mathbf{k}}^2 + \|f_\perp\|_{\mathbf{k}}^2 \quad \text{and} \quad \|f_{\boldsymbol{u}}\|_{\mathbf{k}}^2 = \boldsymbol{u}^\top \mathbf{K} \boldsymbol{u}. \tag{18}$$

Invoking the orthogonality of $f_\perp$ and $(\mathbb{P}_{\text{in}} - \mathbb{P}_{\text{out}})\mathbf{k} \in \mathcal{L}$, the reproducing property representations (17), and the vector sub-Gaussianity condition (15), we find that

$$\mathbb{E}_{\mathcal{E}}[\exp(\langle f, (\mathbb{P}_{\text{in}} - \mathbb{P}_{\text{out}})\mathbf{k}\rangle_{\mathbf{k}})] = \mathbb{E}_{\mathcal{E}}[\exp(\langle f_{\boldsymbol{u}} + f_\perp, (\mathbb{P}_{\text{in}} - \mathbb{P}_{\text{out}})\mathbf{k}\rangle_{\mathbf{k}})] = \mathbb{E}_{\mathcal{E}}\left[\exp(\boldsymbol{u}^\top \mathbf{K}(\boldsymbol{p}_{\text{in}} - \boldsymbol{p}_{\text{out}}))\right]$$
$$\leq \exp(\boldsymbol{u}^\top \mathbf{K} \boldsymbol{u} \cdot \tfrac{\nu^2}{2}) \overset{(18)}{\leq} \exp(\|f\|_{\mathbf{k}}^2 \cdot \tfrac{\nu^2}{2}),$$

so that $(\mathbb{P}_{\text{in}} - \mathbb{P}_{\text{out}})\mathbf{k}$ is $(\mathbf{k}, \nu)$-sub-Gaussian on the event $\mathcal{E}$ as claimed. $\qquad\square$

We end our discussion about the versions of sub-Gaussianity considered above by presenting the standard fact about the additivity of sub-Gaussianity parameters under summation of independent sub-Gaussian random vectors, adapted to our setting.

**Lemma A.3** (**Vector sub-Gaussian additivity**). *Suppose that, for each $j \in [m]$, $\Delta_j \in \mathbb{R}^n$ is $(\mathbf{K}, \nu_j)$ on an event $\mathcal{E}_j$ given $\Delta_{1:(j-1)} \triangleq (\Delta_1, \ldots, \Delta_{j-1})$ and $\mathcal{E}_{\leq j-1} \triangleq \bigcap_{i=1}^{j-1} \mathcal{E}_i$. Then $\sum_{j=1}^m \Delta_j$ is $(\mathbf{K}, (\sum_{j=1}^m \nu_j^2)^{1/2})$-sub-Gaussian on $\mathcal{E}_{\leq m}$.*

*Proof.* Let $\mathcal{E}_{\leq s} = \bigcap_{j=1}^s \mathcal{E}_j$ for each $s \in [m]$. We prove the result for $\mathcal{Z}_s = \sum_{i=1}^s \Delta_j$ by induction on $s \in [m]$. The result holds for the base case of $s = 1$ by assumption. For the inductive case, suppose the result holds for $s \in [m-1]$. Fixing $\boldsymbol{u} \in \mathbb{R}^n$, we may apply the tower property, our conditional sub-Gaussianity assumption, and our inductive hypothesis in turn to conclude

$$\mathbb{E}\left[\exp(\langle \boldsymbol{u}, \mathbf{K}\textstyle\sum_{j=1}^{s+1}\Delta_j\rangle)\mathbf{1}[\mathcal{E}_{\leq s+1}]\right] = \mathbb{E}\left[\exp(\langle \boldsymbol{u}, \mathbf{K}\textstyle\sum_{j=1}^s\Delta_j\rangle)\mathbf{1}[\mathcal{E}_{\leq s}]\mathbb{E}[\exp(\langle \boldsymbol{u}, \Delta_{s+1}\rangle)\mathbf{1}[\mathcal{E}_{s+1}] \mid \Delta_{1:s}, \mathcal{E}_{\leq s}]\right]$$
$$\leq \mathbb{E}\left[\exp(\langle \boldsymbol{u}, \mathbf{K}\textstyle\sum_{j=1}^s\Delta_j\rangle)\mathbf{1}[\mathcal{E}_{\leq s}]\right]\exp\left(\tfrac{\nu_{s+1}^2}{2} \cdot \boldsymbol{u}^\top \mathbf{K} \boldsymbol{u}\right) \leq \exp\left(\tfrac{\sum_{j=1}^{s+1}\nu_j^2}{2} \cdot \boldsymbol{u}^\top \mathbf{K} \boldsymbol{u}\right).$$

Hence, $\mathcal{Z}_{s+1}$ is $(\mathbf{K}, (\sum_{j=1}^{s+1} \nu_j^2)^{1/2})$-sub-Gaussian on $\mathcal{E}_{\leq s+1}$, and the proof is complete. $\qquad\square$

## B. Proof of Tab. 1: Sub-Gaussian Thinning Examples

This section provides supplementary details for each of the sub-Gaussian thinning algorithms of Tab. 1.

### B.1. SUBSAMPLING

#### B.1.1. PROOF OF PROP. 1: QUALITY OF UNIFORM SUBSAMPLING

We begin by computing the first and second moments of $\boldsymbol{p}_{\text{out}}$: $\mathbb{E}[\boldsymbol{p}_{\text{out}}] = \boldsymbol{p}_{\text{in}}$ and

$$\mathbb{E}[\boldsymbol{p}_{\text{out}}\boldsymbol{p}_{\text{out}}^\top] = \tfrac{1}{n_{\text{out}}}\operatorname{diag}(\boldsymbol{p}_{\text{in}}) + \tfrac{n_{\text{in}}(n_{\text{out}}-1)}{n_{\text{out}}(n_{\text{in}}-1)}(\boldsymbol{p}_{\text{in}}\boldsymbol{p}_{\text{in}}^\top - \tfrac{1}{n_{\text{in}}}\operatorname{diag}(\boldsymbol{p}_{\text{in}})) = \tfrac{1}{n_{\text{out}}}(\tfrac{n_{\text{in}}-n_{\text{out}}}{n_{\text{in}}-1})\operatorname{diag}(\boldsymbol{p}_{\text{in}}) + \tfrac{n_{\text{in}}(n_{\text{out}}-1)}{n_{\text{out}}(n_{\text{in}}-1)}\boldsymbol{p}_{\text{in}}\boldsymbol{p}_{\text{in}}^\top.$$

Hence,

$$\mathbb{E}[\operatorname{MMD}_{\mathbf{K}}^2(\boldsymbol{p}_{\text{in}}, \boldsymbol{p}_{\text{out}})] = \boldsymbol{p}_{\text{in}}^\top \mathbf{K} \boldsymbol{p}_{\text{in}} - 2\boldsymbol{p}_{\text{in}}^\top \mathbf{K}\mathbb{E}[\boldsymbol{p}_{\text{out}}] + \mathbb{E}[\boldsymbol{p}_{\text{out}}^\top \mathbf{K} \boldsymbol{p}_{\text{out}}] = \operatorname{tr}(\mathbf{K}\mathbb{E}[\boldsymbol{p}_{\text{out}}\boldsymbol{p}_{\text{out}}^\top]) - \boldsymbol{p}_{\text{in}}^\top \mathbf{K} \boldsymbol{p}_{\text{in}}$$
$$= \tfrac{1}{n_{\text{out}}}(\tfrac{n_{\text{in}}-n_{\text{out}}}{n_{\text{in}}-1})(\operatorname{tr}(\mathbf{K}\operatorname{diag}(\boldsymbol{p}_{\text{in}})) - \boldsymbol{p}_{\text{in}}^\top \mathbf{K} \boldsymbol{p}_{\text{in}}) = \tfrac{1}{n_{\text{out}}}(\tfrac{n_{\text{in}}-n_{\text{out}}}{n_{\text{in}}-1})C_{\mathbf{K}}. \tag{19}$$

To derive the second advertised result, we note that

$$\mathbb{E}[\|\mathbf{K}(\boldsymbol{p}_{\text{in}} - \boldsymbol{p}_{\text{out}})\|_{\mathcal{I}}^2] \geq \max_{i \in \mathcal{I}} \mathbb{E}[(\boldsymbol{e}_i^\top \mathbf{K}(\boldsymbol{p}_{\text{in}} - \boldsymbol{p}_{\text{out}}))^2] = \max_{i \in \mathcal{I}} \mathbb{E}[\operatorname{MMD}_{\mathbf{K}\boldsymbol{e}_i\boldsymbol{e}_i^\top \mathbf{K}}^2(\boldsymbol{p}_{\text{in}}, \boldsymbol{p}_{\text{out}})]$$

and invoke the initial result (19) to conclude.

### B.1.2. SUB-GAUSSIANITY OF SUBSAMPLING

**Proposition B.1** (**Sub-Gaussianity of uniform subsampling**). *For any SPSD* $\mathbf{K} \in \mathbb{R}^{n \times n}$*, uniform subsampling (without replacement) is a* $(\mathbf{K}, \nu, 0)$*-sub-Gaussian thinning algorithm with*

$$\nu \triangleq \frac{\sqrt{\|\mathbf{K}\|_{\max}}}{\sqrt{n_{\text{out}}}}.$$

*Proof.* Fix any vector $\boldsymbol{u} \in \mathbb{R}^n$, and let $J_1, \ldots, J_{n_{\text{out}}}$ be the random indices in $[n]$ selected by uniform subsampling. Since $\boldsymbol{u}^\top \mathbf{K}(\boldsymbol{p}_{\text{in}} - \boldsymbol{p}_{\text{out}}) = \frac{1}{n_{\text{out}}} \sum_{i=1}^{n_{\text{out}}} \boldsymbol{u}^\top \mathbf{K}(\boldsymbol{p}_{\text{in}} - \boldsymbol{e}_{J_i})$ is an average of mean-centered scalars drawn without replacement and satisfying

$$|\boldsymbol{u}^\top \mathbf{K} \boldsymbol{e}_{J_i}| \leq \sqrt{\boldsymbol{u}^\top \mathbf{K} \boldsymbol{u}} \sqrt{\boldsymbol{e}_{J_i}^\top \mathbf{K} \boldsymbol{e}_{J_i}} \leq \sqrt{\|\mathbf{K}\|_{\max}} \sqrt{\boldsymbol{u}^\top \mathbf{K} \boldsymbol{u}} \quad \text{with probability 1}$$

by Cauchy-Schwarz, Thm. 4 and equations (1.8) and (4.16) of Hoeffding (1963) imply that

$$\mathbb{E}[\exp(\boldsymbol{u}^\top \mathbf{K}(\boldsymbol{p}_{\text{in}} - \boldsymbol{p}_{\text{out}}))] \leq \exp(\tfrac{\|\mathbf{K}\|_{\max}}{2n_{\text{out}}} \boldsymbol{u}^\top \mathbf{K} \boldsymbol{u}).$$

□

## B.2. KH($\delta$)

---

**Algorithm B.1:** KH($\delta$): Kernel Halving with simplified swapping thresholds and failure probability $\delta/2$

---

**Input:** point sequence $\mathcal{X}_{\text{in}} = (\boldsymbol{x}_i)_{i=1}^{n_{\text{in}}}$ with even $n_{\text{in}}$, kernel $\mathbf{k}$

$\mathcal{S}^{(1)}, \mathcal{S}^{(2)} \leftarrow \{\}; \quad \widetilde{\psi}_0 \leftarrow \mathbf{0} \in \mathcal{H}_{\mathbf{k}}$   // Initialize empty coresets: $\mathcal{S}^{(1)}, \mathcal{S}^{(2)}$ have size $i$ after round $i$
$\mathfrak{b}_{\max,i} \leftarrow 0$                           // Max function norm so far
**for** $i = 1, 2, \ldots, n_{\text{in}}/2$ **do**

  // Construct kernel difference function using next two points
  $(\boldsymbol{x}, \boldsymbol{x}') \leftarrow (\boldsymbol{x}_{2i-1}, \boldsymbol{x}_{2i}); \quad f_i \leftarrow \mathbf{k}(\boldsymbol{x}_{2i-1}, \cdot) - \mathbf{k}(\boldsymbol{x}_{2i}, \cdot); \quad \eta_i \leftarrow -1$

  // Compute swapping threshold $\mathfrak{a}_i$
  $\mathfrak{b}_i^2 = \|f_i\|_{\mathbf{k}}^2 = \mathbf{k}(\boldsymbol{x}, \boldsymbol{x}) + \mathbf{k}(\boldsymbol{x}', \boldsymbol{x}') - 2\mathbf{k}(x, x'); \quad \mathfrak{b}_{\max,i} = \max(\mathfrak{b}_i, \mathfrak{b}_{\max,i-1})$
  $\mathfrak{a}_i \leftarrow \mathfrak{b}_i \mathfrak{b}_{\max,i}(\tfrac{1}{2} + \log(2n_{\text{in}}/\delta))$

  // Compute RKHS inner product $\left\langle \widetilde{\psi}_{i-1}, f_i \right\rangle_{\mathbf{k}}$, which has a simple form
  $\alpha_i \leftarrow \sum_{j=1}^{2i-2}(\mathbf{k}(\boldsymbol{x}_j, \boldsymbol{x}) - \mathbf{k}(\boldsymbol{x}_j, \boldsymbol{x}')) - 2\sum_{\boldsymbol{z} \in \mathcal{S}^{(1)}}(\mathbf{k}(\boldsymbol{z}, \boldsymbol{x}) - \mathbf{k}(\boldsymbol{z}, \boldsymbol{x}'))$

  // Assign one point to each coreset after probabilistic swapping
  $(x, x') \leftarrow (x', x)$ and $\eta_i \leftarrow 1$   *with probability*   $\min(1, \tfrac{1}{2}(1 - \tfrac{\alpha_i}{\mathfrak{a}_i})_+)$
  $\mathcal{S}^{(1)}.\texttt{append}(\boldsymbol{x}); \quad \mathcal{S}^{(2)}.\texttt{append}(\boldsymbol{x}'); \quad \widetilde{\psi}_i \leftarrow \widetilde{\psi}_{i-1} + \eta_i f_i$    // $\widetilde{\psi}_i = \sum_{x' \in \mathcal{S}^{(2)}} \mathbf{k}(x', \cdot) - \sum_{x \in \mathcal{S}^{(1)}} \mathbf{k}(x, \cdot)$

**end**
**return** $\mathcal{X}_{\text{out}} \triangleq \mathcal{S}^{(1)}$, coreset of size $n_{\text{out}} = n_{\text{in}}/2$

---

In this section, we analyze KH($\delta$) (Alg. B.1), a variant of the Kernel Halving algorithm (Dwivedi & Mackey, 2024, Alg. 2) with simplified swapping thresholds. Prop. B.2, proved in App. B.2.1, establishes the sub-Gaussianity of KH($\delta$) and its intermediate iterates.

**Proposition B.2** (**Sub-Gaussianity of KH($\delta$)**). *Suppose* $n_{\text{in}} \geq 2$*. In the notation of Alg. B.1, on a common event* $\mathcal{E}$ *of probability at least* $1 - \delta/2$*, for all* $i \in [n_{\text{in}}/2]$*,* $\frac{1}{2i}\widetilde{\psi}_i$ *is* $(\mathbf{k}, \nu_i)$*-sub-Gaussian with*

$$\nu_i = \mathfrak{b}_{\max,i} \frac{\sqrt{\log(2n_{\text{in}}/\delta)}}{2i} = \frac{\sqrt{\log(2n_{\text{in}}/\delta)}}{2i} \max_{j \in [i]} \text{MMD}_{\mathbf{k}}(\boldsymbol{\delta}_{\boldsymbol{x}_{2j-1}}, \boldsymbol{\delta}_{\boldsymbol{x}_{2j}}) \leq \frac{\sqrt{\log(2n_{\text{in}}/\delta)}}{2i} \max_{j \in [i]} \text{MMD}_{\mathbf{k}}(\boldsymbol{\delta}_{\boldsymbol{x}_{2j-1}}, \boldsymbol{\delta}_{\boldsymbol{x}_{2j}})$$
$$\leq \frac{\sqrt{\log(2n_{\text{in}}/\delta)}}{2i} 2\min(\max_{\boldsymbol{x} \in \mathcal{X}_{\text{in}}} \sqrt{\mathbf{k}(\boldsymbol{x}, \boldsymbol{x})}, \max_{\boldsymbol{x} \in \mathcal{X}_{\text{in}}} \text{MMD}_{\mathbf{k}}(\boldsymbol{\delta}_{\boldsymbol{x}}, \mathbb{P}_{\text{in}})).$$

Prop. B.2 and the triangle inequality imply that $(\mathbb{P}_{\text{in}} - \mathbb{P}_{\text{out}})\mathbf{k} = \frac{1}{n_{\text{in}}}\widetilde{\psi}_{n_{\text{in}}/2}$ is $(\mathbf{k}, \nu)$-sub-Gaussian on $\mathcal{E}$ with

$$\nu = \mathfrak{b}_{\max,n_{\text{in}}/2} \frac{\sqrt{\log(2n_{\text{in}}/\delta)}}{n_{\text{in}}} \leq \frac{\sqrt{\log(2n_{\text{in}}/\delta)}}{n_{\text{in}}} 2\min(\max_{\boldsymbol{x} \in \mathcal{X}_{\text{in}}} \sqrt{\mathbf{k}(\boldsymbol{x}, \boldsymbol{x})}, \max_{\boldsymbol{x} \in \mathcal{X}_{\text{in}}} \text{MMD}_{\mathbf{k}}(\boldsymbol{\delta}_{\boldsymbol{x}}, \mathbb{P}_{\text{in}})).$$

By Lem. A.1, we thus have that the KH($\delta$) output $\boldsymbol{p}_{\text{in}} - \boldsymbol{p}_{\text{out}}$ is $(\mathbf{K}, \nu)$-sub-Gaussian on $\mathcal{E}$ for $\mathbf{K}$ generated by $\mathbf{k}$ and that $\text{KH}(\delta) \in \mathcal{G}_{\nu,\delta}(\mathbf{K})$.

### B.2.1. PROOF OF PROP. B.2: SUB-GAUSSIANITY OF KH($\delta$)

We begin by studying the sub-Gaussian properties of a related algorithm, the self-balancing Hilbert walk (SBHW) of Dwivedi & Mackey (2024, Alg. 3). By Dwivedi & Mackey (2024, Thm. 3(i)), when the SBHW is run on the RKHS $\mathcal{H}_{\mathbf{k}}$ with the same $f_i$ and $\mathfrak{a}_i$ sequences employed in KH($\delta$), the output $\psi_i$ of each round is $(\mathbf{k}, \sigma_i)$-sub-Gaussian for

$$\sigma_0^2 \triangleq 0 \quad \text{and} \quad \sigma_i^2 \triangleq \sigma_{i-1}^2 + \|f_i\|_{\mathbf{k}}^2 \big(1 + \tfrac{\sigma_{i-1}^2}{\mathfrak{a}_i^2}(\|f_i\|_{\mathbf{k}}^2 - 2\mathfrak{a}_i)\big)_+ \quad \forall i \geq 1. \tag{20}$$

The following lemma bounds the growth of the sub-Gaussian constants $\sigma_i$ in terms of the swapping thresholds $\mathfrak{a}_i$.

**Lemma B.1** (Growth of SBHW sub-Gaussian constants)**.** *For each $i$, the SBHW sub-Gaussian constants* (20) *satisfy*

$$\sigma_i^2 \leq c_i \quad \text{for} \quad c_i \triangleq \max_{j \in [i]} \max(\mathfrak{b}_j^2, r_j) \quad \text{and} \quad r_i \triangleq \frac{\mathfrak{a}_i^2}{2\mathfrak{a}_i - \mathfrak{b}_i^2} \leq \frac{\mathfrak{a}_i^2}{2\mathfrak{a}_i - \mathfrak{b}_i \mathfrak{b}_{\max,i}}.$$

*Proof.* We will prove the result by induction on $i$.

**Base case.** $\sigma_1^2 = \mathfrak{b}_1^2 \leq c_1$ as desired.

**Inductive case.** Suppose $\sigma_{i-1}^2 \leq c_{i-1}$. Then $\sigma_i^2 = g(\sigma_{i-1}^2)$ for $g(x) = x + \mathfrak{b}_i^2(1 - x/r_i)_+$. Note that the slope of $g$ is $1 - \mathfrak{b}_i^2/r_i$ for $x < r_i$ and $1$ for $x > r_i$. If $1 - \mathfrak{b}_i^2/r_i \geq 0$, then $g$ is increasing and its maximum value over $[0, c_i]$ is at $c_i$. If, on the other hand, $1 - \mathfrak{b}_i^2/r_i < 0$, then $g$ first decreases and then increases so its maximum value over $[0, c_i]$ is either at $0$ or at $c_i$. Since $c_i \geq \max(r_i, c_{i-1})$, $\sigma_i^2 \leq \max(g(0), g(c_i)) = \max(\mathfrak{b}_i^2, c_i) = c_i$. The proof is complete. $\square$

Invoking Lem. B.1, the assumption $n_{\text{in}} \geq 2$, and the fact that $\delta \mapsto \frac{\frac{1}{2} + \log(2/\delta)}{\log(2/\delta)}$ is increasing on $(0, 1]$, we find that

$$\sigma_i^2 \leq \mathfrak{b}_{\max,i}^2 \log(2n_{\text{in}}/\delta) \frac{(\frac{1}{2} + \log(2n_{\text{in}}/\delta))^2}{2(\log(2n_{\text{in}}/\delta))^2} \leq \mathfrak{b}_{\max,i}^2 \log(2n_{\text{in}}/\delta) \frac{(\frac{1}{2} + \log(4))^2}{2(\log(4))^2} \leq \mathfrak{b}_{\max,i}^2 \log(2n_{\text{in}}/\delta). \tag{21}$$

The first inequality in (21) and the definition (20) further imply that

$$\mathfrak{a}_i = \mathfrak{b}_i \mathfrak{b}_{\max,i}(\tfrac{1}{2} + \log(2n_{\text{in}}/\delta)) \geq \sigma_i \mathfrak{b}_i \sqrt{2\log(2n_{\text{in}}/\delta)} \geq \sigma_{i-1} \mathfrak{b}_i \sqrt{2\log(2n_{\text{in}}/\delta)}.$$

Hence, by Dwivedi & Mackey (2024, Thm. 3(iii)), for each $i \in [n_{\text{in}}/2]$, the vector $\widetilde{\psi}_i$ of KH($\delta$) coincides with the vector $\psi_i$ of SBHW on a common event $\mathcal{E}$ of probability at least $1 - \delta/2$. Therefore, each $\frac{1}{2i}\widetilde{\psi}_i$ is $(\mathbf{k}, \frac{1}{2i}\sigma_i)$-sub-Gaussian on $\mathcal{E}$, implying the result.

### B.3. LKH($\delta$)

In this section, we analyze LKH($\delta$) (Alg. B.2), the Kernel Halving algorithm of (Dwivedi & Mackey, 2024, Alg. 2) with a linear kernel, $\mathbf{k}(\boldsymbol{x}, \boldsymbol{y}) = \langle \boldsymbol{x}, \boldsymbol{y} \rangle$, on $\mathbb{R}^d$ and failure probability $\delta/2$. Notably, Alg. B.2 can be carried out in only $O(nd)$ time thanks to the linear kernel structure. Prop. B.3, proved in App. B.3.1, establishes the sub-Gaussianity of LKH($\delta$) and its intermediate iterates.

**Proposition B.3** (Sub-Gaussianity of LKH($\delta$))**.** *Suppose $n_{\text{in}} \geq 2$. In the notation of Alg. B.2, on a common event $\mathcal{E}$ of probability at least $1 - \delta/2$, for all $i \in [n_{\text{in}}/2]$, $\frac{1}{2i}\widetilde{\psi}_i$ is $(\mathbf{k}, \nu_i)$-sub-Gaussian with $\mathbf{k}(\boldsymbol{x}, \boldsymbol{y}) = \langle \boldsymbol{x}, \boldsymbol{y} \rangle$ and*

$$\begin{aligned}\nu_i &= \frac{\sqrt{\log(2n_{\text{in}}(\log(n_{\text{in}}/2)+1)/\delta)}}{2i} \max_{j \in [i]} \|\boldsymbol{x}_{2j-1} - \boldsymbol{x}_{2j}\|_2 \\ &\leq \frac{\sqrt{\log(2n_{\text{in}}(\log(n_{\text{in}}/2)+1)/\delta)}}{2i} 2\min(\max_{\boldsymbol{x} \in \mathcal{X}_{\text{in}}} \sqrt{\|\boldsymbol{x}\|_2}, \max_{\boldsymbol{x} \in \mathcal{X}_{\text{in}}} \|\boldsymbol{x} - \bar{\boldsymbol{x}}\|_2) \quad \text{for} \quad \bar{\boldsymbol{x}} = \tfrac{1}{n_{\text{in}}} \sum_{\boldsymbol{x} \in \mathcal{X}_{\text{in}}} \boldsymbol{\delta}_{\boldsymbol{x}}.\end{aligned}$$

Prop. B.3 and the triangle inequality imply that $(\mathbb{P}_{\text{in}} - \mathbb{P}_{\text{out}})\mathbf{k} = \frac{1}{n_{\text{in}}}\psi_{n_{\text{in}}/2}$ is $(\mathbf{k}, \nu)$-sub-Gaussian on $\mathcal{E}$ with

$$\begin{aligned}\nu &= \frac{\sqrt{\log(2n_{\text{in}}(\log(n_{\text{in}}/2)+1)/\delta)}}{n_{\text{in}}} \max_{j \in [n_{\text{in}}/2]} \|\boldsymbol{x}_{2j-1} - \boldsymbol{x}_{2j}\|_2 \\ &\leq \frac{\sqrt{\log(2n_{\text{in}}(\log(n_{\text{in}}/2)+1)/\delta)}}{n_{\text{in}}} 2\min(\max_{\boldsymbol{x} \in \mathcal{X}_{\text{in}}} \sqrt{\|\boldsymbol{x}\|_2}, \max_{\boldsymbol{x} \in \mathcal{X}_{\text{in}}} \|\boldsymbol{x} - \bar{\boldsymbol{x}}\|_2) \quad \text{for} \quad \bar{\boldsymbol{x}} = \tfrac{1}{n_{\text{in}}} \sum_{\boldsymbol{x} \in \mathcal{X}_{\text{in}}} \boldsymbol{\delta}_{\boldsymbol{x}}.\end{aligned}$$

---

**Algorithm B.2:** LKH($\delta$): Kernel Halving with linear kernel and failure probability $\delta/2$

---

**Input:** point sequence $\mathcal{X}_{\text{in}} = (\boldsymbol{x}_i)_{i=1}^{n_{\text{in}}}$ with even $n_{\text{in}}$ and $\boldsymbol{x}_i \in \mathbb{R}^d$

$\mathcal{S}^{(1)}, \mathcal{S}^{(2)} \leftarrow \{\};\quad \psi_0 \leftarrow \mathbf{0} \in \mathbb{R}^d \quad$ // Initialize empty coresets: $\mathcal{S}^{(1)}, \mathcal{S}^{(2)}$ have size $i$ after round $i$
$\sigma_0 \leftarrow 0 \qquad\qquad\qquad\qquad\qquad$ // Keep track of sub-Gaussian constant
**for** $i = 1, 2, \ldots, n_{\text{in}}/2$ **do**
    // Consider two points
    $(\boldsymbol{x}, \boldsymbol{x}') \leftarrow (\boldsymbol{x}_{2i-1}, \boldsymbol{x}_{2i});\quad \eta_i \leftarrow -1$

    // Compute swapping threshold $\mathfrak{a}_i$
    $\mathfrak{b}_i^2 = \langle \boldsymbol{x} - \boldsymbol{x}', \boldsymbol{x} - \boldsymbol{x}' \rangle;\quad \delta_i = \frac{\delta}{2i(\log(n_{\text{in}}/2)+1)}$
    $(\mathfrak{a}_i, \sigma_i) \leftarrow \texttt{get\_swap\_params}\,(\sigma_{i-1}, \mathfrak{b}_i, \delta_i)$

    // Compute inner product
    $\alpha_i \leftarrow \langle \psi_{i-1}, \boldsymbol{x} - \boldsymbol{x}' \rangle$

    // Assign one point to each coreset after probabilistic swapping
    $(\boldsymbol{x}, \boldsymbol{x}') \leftarrow (\boldsymbol{x}', \boldsymbol{x})$ and $\eta_i \leftarrow 1 \quad$ *with probability* $\quad \min(1, \frac{1}{2}(1 - \frac{\alpha_i}{\mathfrak{a}_i})_+)$
    $\mathcal{S}^{(1)}.\texttt{append}(\boldsymbol{x});\quad \mathcal{S}^{(2)}.\texttt{append}(\boldsymbol{x}');\quad \widetilde{\psi}_i \leftarrow \widetilde{\psi}_{i-1} + \eta_i f_i$
**end**
**return** $\mathcal{X}_{\text{out}} \triangleq \mathcal{S}^{(1)}$, coreset of size $n_{\text{out}} = n_{\text{in}}/2$

---

**function** $\texttt{get\_swap\_params}\,(\sigma, \mathfrak{b}, \delta)$:
    $\mathfrak{a} \leftarrow \max(\mathfrak{b}\sigma\sqrt{2\log(2/\delta)}, \mathfrak{b}^2)$
    $\sigma^2 \leftarrow \sigma^2 + \mathfrak{b}^2(1 + (\mathfrak{b}^2 - 2\mathfrak{a})\sigma^2/\mathfrak{a}^2)_+$
**return** $(\mathfrak{a}, \sigma)$;

---

By Lem. A.1, we thus have that the LKH($\delta$) output $\boldsymbol{p}_{\text{in}} - \boldsymbol{p}_{\text{out}}$ is $(\mathbf{K}, \nu)$-sub-Gaussian on $\mathcal{E}$ for $\mathbf{K}$ generated by $\mathbf{k}$ and that LKH($\delta$) $\in \mathcal{G}_{\nu,\delta}(\mathbf{K})$.

### B.3.1. PROOF OF PROP. B.3: SUB-GAUSSIANITY OF LKH($\delta$)

We begin by studying the sub-Gaussian properties of a related algorithm, the self-balancing Hilbert walk (SBHW) of Dwivedi & Mackey (2024, Alg. 3). By Dwivedi & Mackey (2024, Thm. 3(i)), when the SBHW is run on the RKHS $\mathcal{H}_{\mathbf{k}}$ with the same $f_i$ and $\mathfrak{a}_i$ sequences employed in LKH($\delta$), the output $\psi_i$ of each round is $(\mathbf{k}, \sigma_i)$-sub-Gaussian. Moreover, since

$$\mathfrak{a}_i \geq \sigma_{i-1}\mathfrak{b}_i\sqrt{2\log(2/\delta_i)} \quad \text{for each} \quad i \in [n_{\text{in}}/2],$$

Dwivedi & Mackey (2024, Thm. 3(iii)) implies that, for each $i \in [n_{\text{in}}/2]$, the vector $\widetilde{\psi}_i$ of LKH($\delta$) coincides with the vector $\psi_i$ of SBHW on a common event $\mathcal{E}$ of probability at least $1 - \delta/2$. Therefore, each $\frac{1}{2i}\widetilde{\psi}_i$ is $(\mathbf{k}, \frac{1}{2i}\sigma_i)$-sub-Gaussian on $\mathcal{E}$. Finally, Dwivedi & Mackey (2024, (46)) shows that $\sigma_i \leq \nu_i$ for each $i \in [n_{\text{in}}/2]$, yielding the result.

### B.4. RKH($\delta$)

---

**Algorithm B.3:** RKH($\delta$): Repeated KH($\delta$)

---

**Input:** point sequence $\mathcal{X}_{\text{in}} = (\boldsymbol{x}_i)_{i=1}^{n_{\text{in}}}$, kernel $\mathbf{k}$, output size $n_{\text{out}} \in n_{\text{in}}/2^{\mathbb{N}}$

// Repeatedly divide coreset size in half
$m \leftarrow \log_2(n_{\text{in}}/n_{\text{out}})$
**for** $\ell = 1, 2, \ldots, m$ **do** $\mathcal{X}_{\text{in}} \leftarrow \text{KH}(\delta/m)(\mathcal{X}_{\text{in}}, \mathbf{k})$ ;
**return** $\mathcal{X}_{\text{out}} \triangleq \mathcal{X}_{\text{in}}$, coreset of size $n_{\text{out}} = n_{\text{in}}/2^m$

---

In this section, we analyze repeated KH($\delta$) (RKH($\delta$), Alg. B.3), a variant of the KT-SPLIT algorithm (Dwivedi & Mackey, 2024, Alg. 1a) with simplified swapping thresholds. Our next result, proved in App. B.4.1, establishes the sub-Gaussianity of RKH($\delta$).

---

**Algorithm B.4:** KH-COMPRESS($\delta$): Compress with KH halving and failure probability $\delta$

---

**Input:** point sequence $\mathcal{X}_{\text{in}} = (\boldsymbol{x}_i)_{i=1}^{n_{\text{in}}}$, kernel $\mathbf{k}$, $n_{\text{out}} \in \sqrt{n_{\text{in}}} \cdot 2^{\mathbb{N}}$

$\mathfrak{g} \leftarrow \log_2(n_{\text{out}}/\sqrt{n_{\text{in}}})$          // identify compression level

**function** compress($\mathcal{S}$)*:*
     **if** $|\mathcal{S}| = 4^{\mathfrak{g}}$ **then return** $\mathcal{S}$
     Partition $\mathcal{S}$ into four arbitrary subsequences $\{\mathcal{S}_i\}_{i=1}^{4}$ each of size $|\mathcal{S}|/4$
     **for** $i = 1, 2, 3, 4$ **do**
         $\widetilde{\mathcal{S}}_i \leftarrow$ compress($\mathcal{S}_i$)          // return coresets of size $2^{\mathfrak{g}} \cdot \sqrt{\frac{|\mathcal{S}|}{4}}$
     **end**
     $\widetilde{\mathcal{S}} \leftarrow$ CONCATENATE($\widetilde{\mathcal{S}}_1, \widetilde{\mathcal{S}}_2, \widetilde{\mathcal{S}}_3, \widetilde{\mathcal{S}}_4$);    $\ell \leftarrow 2 \cdot 2^{\mathfrak{g}} \cdot \sqrt{|\mathcal{S}|}$    // coreset of size $\ell$
     **return** KH$\left(\frac{\ell^2}{n_{\text{in}} 4^{\mathfrak{g}+1}(\log_4 n_{\text{in}} - \mathfrak{g})}\delta\right)(\widetilde{\mathcal{S}}, \mathbf{k})$      // coreset of size $2^{\mathfrak{g}}\sqrt{|\mathcal{S}|}$
**return** compress($\mathcal{X}_{\text{in}}$)          // coreset of size $n_{\text{out}} = 2^{\mathfrak{g}}\sqrt{n_{\text{in}}}$

---

**Proposition B.4 (Sub-Gaussianity of RKH($\delta$)).** *If $n_{\text{out}} \in n_{\text{in}}/2^{\mathbb{N}}$ then RKH($\delta$) (Alg. B.3) is $(\mathbf{k}, \nu)$-sub-Gaussian with*

$$\nu = \frac{2}{n_{\text{out}}\sqrt{3}}\sqrt{\log\left(\frac{6n_{\text{out}}\log_2(n_{\text{in}}/n_{\text{out}})}{\delta}\right)} \min(\max_{\boldsymbol{x} \in \mathcal{X}_{\text{in}}} \sqrt{\mathbf{k}(\boldsymbol{x}, \boldsymbol{x})}, \max_{\boldsymbol{x} \in \mathcal{X}_{\text{in}}} \text{MMD}_{\mathbf{k}}(\boldsymbol{\delta}_{\boldsymbol{x}}, \mathbb{P}_{\text{in}}))$$

*on an event $\mathcal{E}$ of probability at least $1 - \delta/2$.*

By Lem. A.1, we thus have that the RKH($\delta$) output $\boldsymbol{p}_{\text{in}} - \boldsymbol{p}_{\text{out}}$ is $(\mathbf{K}, \nu)$-sub-Gaussian on $\mathcal{E}$ for $\mathbf{K}$ generated by $\mathbf{k}$ and that RKH($\delta$) $\in \mathcal{G}_{\nu,\delta}(\mathbf{K})$. Finally, $\nu = O(\frac{\sqrt{\log(n_{\text{out}}/\delta)}}{n_{\text{out}}})$ when $n_{\text{out}} \geq \sqrt{n_{\text{in}}}$.

### B.4.1. PROOF OF PROP. B.4: SUB-GAUSSIANITY OF RKH($\delta$)

Let $c = 2\min(\max_{\boldsymbol{x} \in \mathcal{X}_{\text{in}}} \sqrt{\mathbf{k}(\boldsymbol{x}, \boldsymbol{x})}, \max_{\boldsymbol{x} \in \mathcal{X}_{\text{in}}} \text{MMD}_{\mathbf{k}}(\boldsymbol{\delta}_{\boldsymbol{x}}, \mathbb{P}_{\text{in}}))$, and, for each $\ell \in [m]$, let $\widetilde{\psi}^{(\ell)}$ represent the vector $\widetilde{\psi}_{n_{\text{in}}/2^{\ell}}$ produced at the end of the $\ell$-th call to KH($\delta$). By the proof of Prop. B.2 and the union bound, on an event $\mathcal{E}$ of probability at least $1 - \delta/2$, $(\widetilde{\psi}^{(\ell)})_{\ell \in [m]} = (\psi^{(\ell)})_{\ell \in [m]}$, where each $\frac{2^{\ell-1}}{n_{\text{in}}}\psi^{(\ell)}$ is $(\mathbf{k}, \nu^{(\ell)})$-sub-Gaussian given $(\psi^{(j)})_{j \in [\ell-1]}$ for

$$\nu^{(\ell)} = c\frac{\sqrt{\log(2n_{\text{in}}m/(2^{\ell-1}\delta))}}{n_{\text{in}}/2^{\ell-1}}.$$

Hence, on $\mathcal{E}$, the weighted sum

$$(\mathbb{P}_{\text{in}} - \mathbb{P}_{\text{out}})\mathbf{k} = \sum_{\ell \in [m]} \frac{2^{\ell-1}}{n_{\text{in}}}\widetilde{\psi}^{(\ell)} = \sum_{\ell \in [m]} \frac{2^{\ell-1}}{n_{\text{in}}}\psi^{(\ell)}$$

is $(\mathbf{k}, \sqrt{\sum_{\ell \in [m]}(\nu^{(\ell)})^2})$-sub-Gaussian by Dwivedi & Mackey (2024, Lem. 14). Finally, by Dwivedi & Mackey (2024, Eq. (63)), $\sqrt{\sum_{\ell \in [m]}(\nu^{(\ell)})^2} \leq \nu$.

### B.5. KH-COMPRESS($\delta$)

In this section, we analyze KH-COMPRESS($\delta$) (Alg. B.4), a variant of the KT-SPLIT-COMPRESS algorithm (Shetty et al., 2022, Ex. 3) with simplified swapping thresholds.

**Proposition B.5 (Sub-Gaussianity of KH-COMPRESS($\delta$)).** *If $n_{\text{out}} \in \sqrt{n_{\text{in}}} 2^{\mathbb{N}}$ then KH-COMPRESS($\delta$) (Alg. B.4) is $(\mathbf{k}, \nu)$-sub-Gaussian with*

$$\nu = \frac{1}{n_{\text{out}}}\sqrt{\log_2(n_{\text{out}})\log\left(\frac{4n_{\text{out}}\log_2(n_{\text{in}}/n_{\text{out}})}{\delta}\right)} \max_{\boldsymbol{x} \in \mathcal{X}_{\text{in}}} \sqrt{\mathbf{k}(\boldsymbol{x}, \boldsymbol{x})}$$

*on an event $\mathcal{E}$ of probability at least $1 - \delta/2$.*

*Proof.* Since the original Kernel Halving algorithm of Dwivedi & Mackey (2024, Alg. 2) is equal to the KT-SPLIT algorithm of Dwivedi & Mackey (2024, Alg. 1a) with $m = 1$ halving round, KH-COMPRESS($\delta$) is simply the KT-SPLIT-COMPRESS algorithm of (Shetty et al., 2022, Ex. 3) with KH($\delta$) of Alg. B.1 substituted for KT-SPLIT($\delta, m = 1$). The

result now follows immediately from the KH($\delta$) sub-Gaussian constant of Prop. B.2 and the argument of Shetty et al. (2022, Rem. 2, Ex. 3). □

Prop. B.5 and Lem. A.1 imply that KH-COMPRESS($\delta$) $\in \mathcal{G}_{\nu,\delta}(\mathbf{K})$ for any kernel $\mathbf{k}$ that generates $\mathbf{K}$. In addition, $\nu = O(\frac{\sqrt{\log(n_{\text{out}})\log(n_{\text{out}}/\delta)}}{n_{\text{out}}})$ when $n_{\text{out}} \geq \sqrt{n_{\text{in}}}$. Furthermore, Shetty et al. (2022, Rem. 1) implies that KH-COMPRESS($\delta$) has a runtime less than $4^{\mathfrak{g}+1}n_{\text{in}}(\log_4(n_{\text{in}}) - \mathfrak{g}) = 4n_{\text{out}}^2\log_2(n_{\text{in}}/n_{\text{out}}) = O(n_{\text{out}}^2)$ when $n_{\text{out}} \geq \sqrt{n_{\text{in}}}$.

## B.6. GS-THIN

The section introduces and analyzes the Gram-Schmidt Thinning algorithm (GS-THIN, Alg. B.5). GS-THIN repeatedly divides an input sequence in half using, GS-HALVE (Alg. B.6), a symmetrized and kernelized version of the Gram-Schmidt (GS) Walk of Bansal et al. (2018). We will present two different implementations of GS-HALVE: a quartic-time implementation (Alg. B.6) based on the GS Walk description of Bansal et al. (2018) and a cubic-time implementation based on local updates to the matrix inverse (Alg. B.7). While both the algorithms lead to the same output given the same source of randomness, we present the original implementation[2] for conceptual clarity and the optimized implementation for improved runtime. Throughout, for a matrix $\mathbf{Q}$ and vector $\boldsymbol{u}$, we use the notation $\mathbf{Q}_{\mathcal{I}\times\mathcal{J}}$ and $\boldsymbol{u}_{\mathcal{I}}$ to represent the submatrix $(\mathbf{Q}_{ij})_{i\in\mathcal{I},j\in\mathcal{J}}$ and subvector $(\boldsymbol{u}_i)_{i\in\mathcal{I}}$.

---

**Algorithm B.5:** GS-THIN: Gram-Schmidt Thinning

---

**Input:** point sequence $\mathcal{X}_{\text{in}} = (\boldsymbol{x}_i)_{i=1}^{n_{\text{in}}}$, kernel $\mathbf{k}$, output size $n_{\text{out}} \in n_{\text{in}}/2^{\mathbb{N}}$, HALVE $\in \{$GS-HALVE, GS-HALVE-CUBIC$\}$

// Repeatedly divide coreset size in half
$m \leftarrow \log_2(n_{\text{in}}/n_{\text{out}})$
**for** $\ell = 1, 2, \ldots, m$ **do** $\mathcal{X}_{\text{in}} \leftarrow$ HALVE$(\mathcal{X}_{\text{in}}, \mathbf{k})$ ;
**return** $\mathcal{X}_{\text{out}} \triangleq \mathcal{X}_{\text{in}}$, coreset of size $n_{\text{out}} = n_{\text{in}}/2^m$

---

Our first result, proved in App. B.6.1, shows that GS-THIN is a sub-Gaussian thinning algorithm.

**Proposition B.6** (**GS-THIN sub-Gaussianity**). *For* $\mathbf{K}$ *generated by* $\mathbf{k}$*,* GS-THIN *(Alg. B.5) is a* $(\mathbf{K}, \nu, 0)$*-sub-Gaussian thinning algorithm with parameter*

$$\nu \triangleq \tfrac{2}{\sqrt{3}}\frac{\sqrt{\|\mathbf{K}\|_{\max}}}{n_{\text{out}}}. \tag{22}$$

Our second result, proved in App. B.6.2, shows that GS-THIN with the GS-HALVE implementation has $O(n_{\text{in}}^4)$ runtime.

**Proposition B.7** (**Runtime of GS-THIN with GS-HALVE**). *The runtime of* GS-THIN *with implementation* GS-HALVE *(Alg. B.6) is* $O(n_{\text{in}}^4)$*.*

Our third result, proved in App. B.6.3, establishes the equivalence between GS-HALVE and GS-HALVE-CUBIC. More precisely, we show that the sequence of partial assignment vectors generated by kernel_gs_walk($\cdot$) of Alg. B.6 and kernel_gs_walk_cubic($\cdot$) of Alg. B.7 are identical given identical inputs, an invertible induced kernel matrix, and an identical source of randomness.

**Proposition B.8** (**Agreement of GS-HALVE and GS-HALVE-CUBIC**). *Let* $\boldsymbol{z}_1, \boldsymbol{z}_2, \ldots$ *be the fractional assignment sequence generated by* kernel_gs_walk($(\boldsymbol{x}_i)_{i=1}^{n_{\text{in}}}$) *in Alg. B.6 and* $\boldsymbol{z}_1', \boldsymbol{z}_2', \ldots$ *be the fractional assignment sequence generated by* kernel_gs_walk_cubic($(\boldsymbol{x}_i)_{i=1}^{n_{\text{in}}}$) *in Alg. B.7 with an identical source of randomness. If the pairwise difference matrix*

$$\mathbf{Q} \triangleq (\mathbf{k}(x_{2i-1}, x_{2j-1}) + \mathbf{k}(x_{2i}, x_{2j}) - \mathbf{k}(x_{2i-1}, x_{2j}) - \mathbf{k}(x_{2i}, x_{2j-1}))_{i,j\in[n_{\text{in}}/2]}$$

*is positive definite, then* $\boldsymbol{z}_t = \boldsymbol{z}_t'$ *for all $t$.*

Our fourth result, proved in App. B.6.4, shows that GS-THIN with the GS-HALVE-CUBIC implementation has $O(n_{\text{in}}^3)$ runtime.

**Proposition B.9** (**Runtime of GS-THIN with GS-HALVE-CUBIC**). *The runtime of* GS-THIN *with implementation* GS-HALVE-CUBIC *(Alg. B.7) is* $O(n_{\text{in}}^3)$*.*

---

[2] Towards making this equivalence clear, Alg. B.6 has been expressed with the same variables that Alg. B.7 uses. Alg. B.6 can be slightly simplified if it were to be considered independently.

---

**Algorithm B.6:** GS-HALVE: Gram-Schmidt Halving

---

**Input:** point sequence $\mathcal{X}_{\text{in}} = (x_i)_{i=1}^{n_{\text{in}}}$ with even $n_{\text{in}}$, kernel $\mathbf{k}$

$\mathcal{X}_{\text{out}} \leftarrow \{\}$    // Initialize empty coreset

// Select one point to keep from each consecutive pair using kernelized GS Walk

$\boldsymbol{z} \leftarrow$ `kernel_gs_walk`$(\mathcal{X}_{\text{in}})$

**for** $i = 1, \ldots, n_{\text{in}}/2$ **do**

   **if** $\boldsymbol{z}_i = 1$ **then**

      |   $\mathcal{X}_{\text{out}}$.`append`$(x_{2i-1})$

   **else**

      |   $\mathcal{X}_{\text{out}}$.`append`$(x_{2i})$

   **end**

**end**

**return** $\mathcal{X}_{\text{out}}$, coreset of size $n_{\text{in}}/2$

---

**function** `kernel_gs_walk`$((\boldsymbol{x}_i)_{i=1}^{n_{\text{in}}})$:

   $t \leftarrow 1; \quad \boldsymbol{z}_t \leftarrow (0, 0, \ldots, 0) \in \mathbb{R}^{n_{\text{in}}/2}$       // Initialize fractional assignment vector

   $\mathcal{A} \leftarrow [n_{\text{in}}/2]$       // Initialize set of active coordinates

   $p \sim \mathcal{A}$       // Select a pivot uniformly at random

   **while** $\boldsymbol{z}_t \notin \{\pm 1\}^{n_{\text{in}}/2}$ **do**

      $\mathcal{A}' \leftarrow \mathcal{A} \setminus \left\{ \min \left( \{i \in [n_{\text{in}}/2] : |\boldsymbol{z}_{ti}| = 1\} \setminus ([n_{\text{in}}/2] \setminus \mathcal{A}) \right) \right\}$

      // Update set of active coordinates by removing smallest index set to $\pm 1$

      **if** $p \notin \mathcal{A}'$ **then**

         |   $p' \sim \text{Unif}(\mathcal{A}')$    // Select a new pivot from $\mathcal{A}'$ uniformly at random

      **else**

         |   $p' \leftarrow p$

      **end**

      // Compute step direction in which to update fractional assignment vector

      $\boldsymbol{u}_t \leftarrow \text{argmin}_{\boldsymbol{u} \in \mathbb{R}^{n_{\text{in}}/2}} \boldsymbol{u}^\top \mathbf{Q} \boldsymbol{u}$ subject to $\boldsymbol{u}_{p'} = 1$ and $\boldsymbol{u}_i = 0$ for all $i \notin \mathcal{A}'$,

         where $\mathbf{Q} \in \mathbb{R}^{(n_{\text{in}}/2) \times (n_{\text{in}}/2)}$ has entries $\mathbf{Q}_{ij} \triangleq \mathbf{k}(x_{2i-1}, x_{2j-1}) + \mathbf{k}(x_{2i}, x_{2j}) - \mathbf{k}(x_{2i-1}, x_{2j}) - \mathbf{k}(x_{2i}, x_{2j-1})$

      $\delta^+ \leftarrow |\max \Delta|$ and $\delta^- \leftarrow |\min \Delta|$, where $\Delta = \left\{ \delta \in \mathbb{R} : \boldsymbol{z}_t + \delta \boldsymbol{u}_t \in [-1, +1]^{n_{\text{in}}/2} \right\}$    // Select candidate step sizes

      $\delta_t \leftarrow \delta^+$ with probability $\delta^-/(\delta^+ + \delta^-)$; otherwise $\delta_t \leftarrow -\delta^-$    // Choose step size and sign at random

      $\boldsymbol{z}_{t+1} \leftarrow \boldsymbol{z}_t + \delta_t \boldsymbol{u}_t$    // Update fractional assignments

      $t \leftarrow t + 1; \quad \mathcal{A} \leftarrow \mathcal{A}'; \quad p \leftarrow p'$

   **end**

**return** $\boldsymbol{z}_t$, sign vector in $\{\pm 1\}^{n_{\text{in}}/2}$

---

---

**Algorithm B.7:** GS-HALVE-CUBIC: Gram-Schmidt Halving with cubic runtime

---

**Input:** point sequence $\mathcal{X}_{\text{in}} = (x_i)_{i=1}^{n_{\text{in}}}$ with even $n_{\text{in}}$, kernel $\mathbf{k}$ with positive definite $\mathbf{k}(\mathcal{X}_{\text{in}}, \mathcal{X}_{\text{in}})$

$\mathcal{X}_{\text{out}} \leftarrow \{\}$     // Initialize empty coreset

// Select one point to keep from each consecutive pair using kernelized GS Walk

$\boldsymbol{z} \leftarrow \texttt{kernel\_gs\_walk\_cubic}(\mathcal{X}_{\text{in}})$

**for** $i = 1, \ldots, n_{\text{in}}/2$ **do**

    **if** $\boldsymbol{z}_i = 1$ **then**

        |   $\mathcal{X}_{\text{out}}.\texttt{append}(x_{2i-1})$

    **else**

        |   $\mathcal{X}_{\text{out}}.\texttt{append}(x_{2i})$

    **end**

**end**

**return** $\mathcal{X}_{\text{out}}$, coreset of size $n_{\text{in}}/2$

---

**function** $\texttt{kernel\_gs\_walk\_cubic}((x_i)_{i=1}^{n_{\text{in}}})$:

    $t \leftarrow 1$;    $\boldsymbol{z}_t \leftarrow (0, 0, \ldots, 0) \in \mathbb{R}^{n_{\text{in}}/2}$          // Initialize fractional assignment vector

    $\mathcal{A} \leftarrow [n_{\text{in}}/2]$          // Initialize set of active coordinates

    $p \sim \mathcal{A}$          // Select pivot uniformly at random

    $\mathbf{Q} \leftarrow (\mathbf{k}(x_{2i-1}, x_{2j-1}) + \mathbf{k}(x_{2i}, x_{2j}) - \mathbf{k}(x_{2i-1}, x_{2j}) - \mathbf{k}(x_{2i}, x_{2j-1}))_{i,j=1}^{n_{\text{in}}/2}$      // Form paired difference kernel matrix

    $\mathbf{C} \leftarrow (\mathbf{Q}_{\mathcal{A}\setminus\{p\} \times \mathcal{A}\setminus\{p\}})^{-1}$

    **while** $\boldsymbol{z}_t \notin \{\pm 1\}^{n_{\text{in}}/2}$ **do**

        $\mathcal{A}' \leftarrow \mathcal{A} \setminus \big\{ \min \big( \{ i \in [n_{\text{in}}/2] : |\boldsymbol{z}_{ti}| = 1 \} \setminus ([n_{\text{in}}/2] \setminus \mathcal{A}) \big) \big\}$

        // Update set of active coordinates by removing smallest index set to $\pm 1$

        **if** $p \notin \mathcal{A}'$ **then**

            |   $p' \sim \text{Unif}(\mathcal{A}')$     // Select a new pivot from $\mathcal{A}'$ uniformly at random

        **else**

            |   $p' \leftarrow p$

        **end**

        $\mathcal{A}_1 \leftarrow \mathcal{A} \setminus \{p\}$

        $\mathcal{A}_2 \leftarrow \mathcal{A}' \setminus \{p'\}$.

        $i \leftarrow \mathcal{A}_1 \setminus \mathcal{A}_2$ // Choose $i$ as the (unique) index that was removed from the active coordinates

        // Compute $(\mathbf{Q}_{\mathcal{A}_2 \times \mathcal{A}_2})^{-1}$ using block matrix inversion and the Sherman-Morrison formula

        $\mathbf{D} \leftarrow \mathbf{C}_{\mathcal{A}_2 \times \mathcal{A}_2}$

        $\mathbf{C} \leftarrow \mathbf{D} - \frac{\mathbf{D}\mathbf{Q}_{\mathcal{A}_2 \times \{i\}} \mathbf{Q}_{\{i\} \times \mathcal{A}_2} \mathbf{D}}{\mathbf{Q}_{ii} + \mathbf{Q}_{\{i\} \times \mathcal{A}_2} \mathbf{D} \mathbf{Q}_{\mathcal{A}_2 \times \{i\}}}$

        // Compute step direction in which to update fractional assignment vector

        Compute $\boldsymbol{u}_t$ as $(\boldsymbol{u}_t)_{\mathcal{A}_2} = -\mathbf{C}\mathbf{Q}_{\mathcal{A}_2 \times \{p'\}}$, $\boldsymbol{u}_{tp'} = 1$, and $\boldsymbol{u}_{ti} = 0$ for $i \notin \mathcal{A}'$

        $\delta^+ \leftarrow |\max \Delta|$ and $\delta^- \leftarrow |\min \Delta|$, where $\Delta = \big\{ \delta \in \mathbb{R} : \boldsymbol{z}_t + \delta \boldsymbol{u}_t \in [-1, +1]^{n_{\text{in}}/2} \big\}$     // Select candidate step sizes

        $\delta_t \leftarrow \delta^+$ with probability $\delta^-/(\delta^+ + \delta^-)$; otherwise $\delta_t \leftarrow -\delta^-$     // Choose step size and sign at random

        $\boldsymbol{z}_{t+1} \leftarrow \boldsymbol{z}_t + \delta_t \boldsymbol{u}_t$    // Update fractional assignments

        $t \leftarrow t+1$;    $\mathcal{A} \leftarrow \mathcal{A}'$;    $p \leftarrow p'$

    **end**

**return** $\boldsymbol{z}_t$, sign vector in $\{\pm 1\}^{n_{\text{in}}/2}$

---

### B.6.1. PROOF OF PROP. B.6: GS-THIN SUB-GAUSSIANITY

Our first lemma bounds the sub-Gaussian constant of GS-HALVE (Alg. B.6).

**Lemma B.2** (**GS-HALVE sub-Gaussianity**). *In the notation of Def. 1, consider the input and output vectors $\boldsymbol{p}_{\text{in}}, \boldsymbol{p}_{\text{out}} \in \mathbb{R}^n$ of GS-HALVE (Alg. B.6) for $\mathcal{X} \supseteq \mathcal{X}_{\text{in}}$ with $|\mathcal{X}| = n \geq n_{\text{in}}$. If $\mathbf{K} = \mathbf{k}(\mathcal{X}, \mathcal{X})$, then $\boldsymbol{p}_{\text{in}} - \boldsymbol{p}_{\text{out}}$ is $(\mathbf{K}, \nu)$-sub-Gaussian with*

$$\nu \triangleq \frac{2\|\mathbf{K}\|_{\max}^{1/2}}{n_{\text{in}}} = \frac{\|\mathbf{K}\|_{\max}^{1/2}}{n_{\text{out}}}.$$

*Proof.* Since $\mathbf{K}$ is SPSD, there exists a matrix $\boldsymbol{\Phi} \in \mathbb{R}^{n \times d}$ such that $\mathbf{K} = \boldsymbol{\Phi}\boldsymbol{\Phi}^\top$. Let $\mathbf{B} \in \mathbb{R}^{d \times (n_{\text{in}}/2)}$ be the matrix with entries

$$\mathbf{B}_{j,i} \triangleq \boldsymbol{\Phi}_{2i-1,j} - \boldsymbol{\Phi}_{2i,j} \quad \text{for} \quad i \in [n_{\text{in}}/2] \quad \text{and} \quad j \in [d].$$

Note that, for each $i \in [n_{\text{in}}/2]$,

$$\textstyle\sum_{j \in [d]} \mathbf{B}_{j,i}^2 = \mathbf{K}_{2i-1,2i-1} + \mathbf{K}_{2i,2i} - \mathbf{K}_{2i-1,2i} - \mathbf{K}_{2i,2i-1} \leq 4\|\mathbf{K}\|_{\max}.$$

Hence, by Harshaw et al. (2024, Thm. 6.6), $\frac{1}{n_{\text{in}}}\mathbf{B}\boldsymbol{z}$ is $(\mathbf{I}, \nu)$-sub-Gaussian where $\mathbf{I}$ is the identity matrix in $\mathbb{R}^{d \times d}$.

Now fix any $\boldsymbol{u} \in \mathbb{R}^d$. Since $\frac{1}{n_{\text{in}}}\mathbf{B}\boldsymbol{z} = -\boldsymbol{\Phi}^\top(\boldsymbol{p}_{\text{in}} - \boldsymbol{p}_{\text{out}})$ by construction,

$$\mathbb{E}\big[\exp\big(\boldsymbol{u}^\top\mathbf{K}(\boldsymbol{p}_{\text{in}} - \boldsymbol{p}_{\text{out}})\big)\big] \leq \mathbb{E}\Big[\exp\Big(-\langle\boldsymbol{\Phi}^\top\boldsymbol{u}, \tfrac{1}{n_{\text{in}}}\mathbf{B}\boldsymbol{z}\rangle\Big)\Big] \leq \exp\Big(\tfrac{\nu^2}{2} \cdot \|\boldsymbol{\Phi}^\top\boldsymbol{u}\|_2^2\Big) = \exp\Big(\tfrac{\nu^2}{2} \cdot \boldsymbol{u}^\top\mathbf{K}\boldsymbol{u}\Big).$$

$\square$

Now, for $\ell \in [m]$, let $\boldsymbol{p}_\ell \in \mathbb{R}^n$ denote the output probability vector produced by the $\ell$-th call to GS-HALVE. Defining $\boldsymbol{p}_0 \triangleq \boldsymbol{p}_{\text{in}}$ and $\boldsymbol{p}_{\text{out}} \triangleq \boldsymbol{p}_m$, we have

$$\boldsymbol{p}_{\text{in}} - \boldsymbol{p}_{\text{out}} = \textstyle\sum_{i=1}^m \Delta_i, \quad \text{for} \quad \Delta_i \triangleq \boldsymbol{p}_{i-1} - \boldsymbol{p}_i \quad \text{for} \quad i \in [m].$$

By Lem. B.2, each $\boldsymbol{p}_{i-1} - \boldsymbol{p}_i$ is $(\mathbf{K}, \frac{2\|\mathbf{K}\|_{\max}^{1/2}}{n_{\text{in}}/2^{i-1}})$-sub-Gaussian conditional on $(\Delta_1, \ldots, \Delta_{i-1})$. Applying Lem. A.3 to the sequence $(\Delta_j)_{j=1}^m$, we find that $\boldsymbol{p}_{\text{in}} - \boldsymbol{p}_{\text{out}}$ is $(\mathbf{K}, \nu)$-sub-Gaussian with parameter

$$\nu = \Big(\textstyle\sum_{j=1}^m \frac{4\|\mathbf{K}\|_{\max}}{(n_{\text{in}}/2^{j-1})^2}\Big)^{1/2} = \frac{2\|\mathbf{K}\|_{\max}^{1/2}}{n_{\text{in}}}\Big(\textstyle\sum_{j=1}^m 4^j\Big)^{1/2} \leq \frac{\|\mathbf{K}\|_{\max}^{1/2}}{n_{\text{in}}}\sqrt{\tfrac{4}{3}4^m}.$$

Simplifying the above using the fact that $n_{\text{out}} = n_{\text{in}}/2^m$ yields our desired result (22).

### B.6.2. PROOF OF PROP. B.7: RUNTIME OF GS-THIN WITH GS-HALVE

We essentially reproduce the argument from Bansal et al. (2018) for the runtime of the GS-HALVE algorithm in our kernelized context.

The main computational cost of GS-HALVE is the execution of the `kernel_gs_walk(·)` subroutine in Alg. B.6. The number of iterations in while loop for $\boldsymbol{z}_t$ is at most $n_{\text{in}}/2$. This is due to the fact that in each iteration, at least one new variable is set to $\{\pm 1\}$. Further, in each iteration, the main computational cost is the computation of

$$\boldsymbol{u}_t \leftarrow \operatorname{argmin}_{\boldsymbol{u} \in \mathbb{R}^{n_{\text{in}}/2}} \boldsymbol{u}^\top\mathbf{Q}\boldsymbol{u}$$

under the constraints that $\boldsymbol{u}_p = 1$ and $\boldsymbol{u}_i = 0$ for all $i \notin \mathcal{A}$. Since this can be implemented in $O(n_{\text{in}}^3)$ time using standard convex optimization techniques, GS-HALVE has total runtime

$$r_{\text{H}}(\ell) \leq C\ell^4$$

for an input sequence of size $\ell$ and a constant $C$ independent of $\ell$. Now, note that GS-THIN calls GS-HALVE iteratively on inputs of size $n_{\text{in}}2^{-i}$ for $i = 0, 1, \ldots, m-1$ where $m = \log_2(n_{\text{in}}/n_{\text{out}})$. Thus, GS-THIN has runtime

$$\textstyle\sum_{i=0}^{m-1} r_{\text{H}}(n_{\text{in}}/2^i) \leq \sum_{i=0}^{m-1} C(n_{\text{in}}/2^i)^4 = O(n_{\text{in}}^4).$$

### B.6.3. PROOF OF PROP. B.8: AGREEMENT OF GS-HALVE AND GS-HALVE-CUBIC

We want to reason that any round of partial coloring leads to the same output across the two algorithms. Fix any fractional assignment update round. Recall that $\mathcal{A}_1 = \mathcal{A}\backslash\{p\}$ and $\mathcal{A}_2 = \mathcal{A}' \backslash \{p'\}$. These represent the active set coordinates without the pivot before and after the update respectively.

The main difference between Algs. B.6 and B.7 is in the computation of the step direction $\boldsymbol{u}_t$, which is the solution of the program

$$\boldsymbol{u}_t \leftarrow \mathrm{argmin}_{\boldsymbol{u}\in\mathbb{R}^n}\, \boldsymbol{u}^\top \mathbf{Q}\boldsymbol{u} \quad \text{subject to} \quad \boldsymbol{u}_{p'} = 1 \quad \text{and} \quad \boldsymbol{u}_i = 0 \quad \text{for all} \quad i \notin \mathcal{A}'.$$

$\boldsymbol{u}_t$ has a closed form with entries

$$(\boldsymbol{u}_t)_{\mathcal{A}_2} = -(\mathbf{Q}_{\mathcal{A}_2\times\mathcal{A}_2})^{-1} \cdot \mathbf{Q}_{\mathcal{A}_2\times\{p'\}}.$$

Note that the invertibility of $\mathbf{Q}_{\mathcal{A}_2\times\mathcal{A}_2}$ follows from the positive-definiteness of $\mathbf{Q}$, as, for any $\boldsymbol{w} \in \mathbb{R}^{|\mathcal{A}_2|}$,

$$\boldsymbol{w}^\top \mathbf{Q}_{\mathcal{A}_2\times\mathcal{A}_2}\boldsymbol{w} = \tilde{\boldsymbol{w}}^\top \mathbf{Q}\tilde{\boldsymbol{w}} > 0$$

for a second vector $\tilde{\boldsymbol{w}}$ with $\tilde{\boldsymbol{w}}_{\mathcal{A}_2} = \boldsymbol{w}$ and all other entries equal to zero. Therefore, to compute $\boldsymbol{u}_t$, it suffices to keep track of the inverse of $\mathbf{Q}_{\mathcal{A}_2\times\mathcal{A}_2}$ as $\mathcal{A}'$ across iterations.

Let $i$ be the unique element in $\mathcal{A}_1\backslash\mathcal{A}_2$. Writing $\mathbf{Q}_{\mathcal{A}_1\times\mathcal{A}_1}$ in block form, we have

$$\mathbf{Q}_{\mathcal{A}_1\times\mathcal{A}_1} = \begin{bmatrix} \mathbf{Q}_{\mathcal{A}_2\times\mathcal{A}_2} & \mathbf{Q}_{\mathcal{A}_2\times\{i\}} \\ \mathbf{Q}_{\{i\}\times\mathcal{A}_2} & \mathbf{Q}_{ii} \end{bmatrix}.$$

By block matrix inversion (see, e.g., Saadetoglu & Dinsev, 2023, Thm. 2), the leading size $|\mathcal{A}_2| \times |\mathcal{A}_2|$ principal submatrix of $(\mathbf{Q}_{\mathcal{A}_1\times\mathcal{A}_1})^{-1}$ equals

$$\mathbf{D} \triangleq \left( \mathbf{Q}_{\mathcal{A}_2\times\mathcal{A}_2} - \frac{\mathbf{Q}_{\mathcal{A}_2\times\{i\}}\mathbf{Q}_{\{i\}\times\mathcal{A}_2}}{\mathbf{Q}_{ii}} \right)^{-1}.$$

Thus, by the Sherman-Morrison formula (Sherman & Morrison, 1950),

$$(\mathbf{Q}_{\mathcal{A}_2\times A_2})^{-1} = \left( \mathbf{D}^{-1} + \frac{\mathbf{Q}_{\mathcal{A}_2\times\{i\}}\mathbf{Q}_{\{i\}\times\mathcal{A}_2}}{\mathbf{Q}_{ii}} \right)^{-1} = \mathbf{D} - \frac{\mathbf{D}\mathbf{Q}_{\mathcal{A}_2\times\{i\}}\mathbf{Q}_{\{i\}\times\mathcal{A}_2}\mathbf{D}}{\mathbf{Q}_{ii}+\mathbf{Q}_{\{i\}\times\mathcal{A}_2}\mathbf{D}\mathbf{Q}_{\mathcal{A}_2\times\{i\}}}. \tag{23}$$

Hence, if we already have access to a matrix $\mathbf{C} = (\mathbf{Q}_{\mathcal{A}_1\times\mathcal{A}_1})^{-1}$, we can compute $\mathbf{D}$ by dropping the row and column of $\mathbf{C}$ corresponding to $i$ and then compute $(\mathbf{Q}_{\mathcal{A}_2\times\mathcal{A}_2})^{-1}$ using (23). Since in Alg. B.7 we begin by explicitly computing the inverse of $\mathbf{Q}_{\mathcal{A}'\times\mathcal{A}'}$, the update step in Alg. B.7 maintains the required inverse and thus its partial assignment updates match those of Alg. B.6.

### B.6.4. PROOF OF PROP. B.9: RUNTIME OF GS-THIN WITH GS-HALVE-CUBIC

We begin by establishing the runtime of `kernel_gs_walk_cubic(·)`.

**Lemma B.3 (Running time of `kernel_gs_walk_cubic(·)`).** *The routine* `kernel_gs_walk_cubic(·)` *runs in* $O(\ell^3)$ *time given a point sequence of size $\ell$.*

*Proof.* First, the initialization of $\mathbf{C}$ costs $O(\ell^3)$ time using standard matrix inversion algorithms. Second, the number of iterations in the while loop is at most $\ell/2$ since, in each iteration, at least one new variable is assigned a permanent sign in $\{\pm 1\}$. In each while loop iteration, the main computational costs are the update of $\mathbf{C}$ and the computation of the step direction $\boldsymbol{u}_t$, both of which cost $O(\ell^2)$ time using standard matrix-vector multiplication. Hence, together, all while loop iterations cost $O(\ell^3)$ time. $\qquad\square$

Given the above lemma, we have that GS-HALVE-CUBIC, on input of size $\ell$, has a running time

$$r_{\mathrm{H}}(\ell) \le C\ell^3$$

for some $C$ independent of $\ell$. When used in GS-THIN this yields the runtime

$$\textstyle\sum_{i=0}^{m-1} r_{\mathrm{H}}(n_{\mathrm{in}}/2^i) \le \sum_{i=0}^{m-1} C(n_{\mathrm{in}}/2^i)^3 = O(n_{\mathrm{in}}^3).$$

---

**Algorithm B.8:** GS-COMPRESS: Compress with GS-HALVE-CUBIC halving

---

**Input:** point sequence $\mathcal{X}_{\text{in}} = (\boldsymbol{x}_i)_{i=1}^{n_{\text{in}}}$, kernel $\mathbf{k}$, $n_{\text{out}} \in \sqrt{n_{\text{in}}} \cdot 2^{\mathbb{N}}$

$\mathfrak{g} \leftarrow \log_2(n_{\text{out}}/\sqrt{n_{\text{in}}})$             // identify compression level

**function** `compress(`$\mathcal{S}$`)`:

    **if** $|\mathcal{S}| = 4^{\mathfrak{g}}$ **then return** $\mathcal{S}$

    Partition $\mathcal{S}$ into four arbitrary subsequences $\{\mathcal{S}_i\}_{i=1}^4$ each of size $|\mathcal{S}|/4$

    **for** $i = 1, 2, 3, 4$ **do**

        $\widetilde{\mathcal{S}}_i \leftarrow$ `compress(`$\mathcal{S}_i$`)`          // return coresets of size $2^{\mathfrak{g}} \cdot \sqrt{\frac{|\mathcal{S}|}{4}}$

    **end**

    $\widetilde{\mathcal{S}} \leftarrow$ CONCATENATE$(\widetilde{\mathcal{S}}_1, \widetilde{\mathcal{S}}_2, \widetilde{\mathcal{S}}_3, \widetilde{\mathcal{S}}_4)$;    $\ell \leftarrow 2 \cdot 2^{\mathfrak{g}} \cdot \sqrt{|\mathcal{S}|}$    // coreset of size $\ell$

    **return** GS-HALVE-CUBIC$(\widetilde{\mathcal{S}}, \mathbf{k})$          // coreset of size $2^{\mathfrak{g}}\sqrt{|\mathcal{S}|}$

**return** `compress(`$\mathcal{X}_{\text{in}}$`)`              // coreset of size $n_{\text{out}} = 2^{\mathfrak{g}}\sqrt{n_{\text{in}}}$

---

### B.7. GS-COMPRESS

This section introduces and analyzes the new GS-COMPRESS algorithm (Alg. B.8) which combines the COMPRESS meta-algorithm of Shetty et al. (2022) with the GS-HALVE-CUBIC halving algorithm (Alg. B.7). The following result bounds the sub-Gaussian constant and runtime of GS-COMPRESS.

**Proposition B.10** (**GS-COMPRESS sub-Gaussianity and runtime**). *If* $\mathbf{K}$ *is generated by* $\mathbf{k}$*, then* GS-COMPRESS *is* $(\mathbf{K}, \nu, 0)$*-sub-Gaussian with*

$$\nu \triangleq \tfrac{1}{n_{\text{out}}} \sqrt{\log_2(n_{\text{out}})\|\mathbf{K}\|_{\max}}.$$

*Moreover,* GS-COMPRESS *has an* $O(n_{\text{out}}^3)$ *runtime.*

*Proof.* By Lem. B.2 and Prop. B.8, GS-HALVE-CUBIC is $(\mathbf{K}, \nu_{\text{H}}(\ell))$-sub-Gaussian for an input point sequence of size $\ell$ and $\nu_{\text{H}}(\ell) = 2\sqrt{\|K\|_{\max}}/\ell$. Hence, by Lem. A.2, GS-HALVE-CUBIC is also $\nu_{\text{H}}(\ell)$ $f$-sub-Gaussian in the sense of Shetty et al. (2022, Def. 2) for each $f \in \mathcal{H}_{\mathbf{k}}$. By Shetty et al. (2022, Rmk. 2), GS-COMPRESS is therefore $f$-sub-Gaussian with parameter

$$\nu \leq \sqrt{\log_2(n_{\text{in}}/n_{\text{out}})}\nu_{\text{H}}(2n_{\text{out}}) \leq \sqrt{\log_2(n_{\text{out}})}\tfrac{\|\mathbf{K}\|_{\max}^{1/2}}{n_{\text{out}}}$$

for each $f \in \mathcal{H}_{\mathbf{k}}$. Hence, Lem. A.1 implies that GS-COMPRESS is a $(\mathbf{K}, \nu, 0)$-sub-Gaussian thinning algorithm.

Furthermore, Shetty et al. (2022, Thm. 1) implies that GS-COMPRESS has a runtime of

$$\textstyle\sum_{i=0}^{\log_2(n_{\text{in}}/(2n_{\text{out}}))} 4^i \cdot r_{\text{H}}(2n_{\text{out}}2^{-i}).$$

where the GS-HALVE-CUBIC runtime $r_{\text{H}}(\ell) \leq C\ell^3$ for $C$ independent of the input size $\ell$ by Lem. B.3. Therefore, the GS-COMPRESS runtime is bounded by

$$\textstyle\sum_{i=0}^{\log_2(n_{\text{in}}/(2n_{\text{out}}))} 4^i \cdot (2n_{\text{out}})^3 2^{-3i} = O(n_{\text{out}}^3).$$

$\square$

**Remark 1** (**COMPRESS with GS-HALVE**). If the GS-HALVE implementation were used in place of GS-HALVE-CUBIC, parallel reasoning would yield an $O(n_{\text{out}}^4)$ runtime for GS-COMPRESS.

## C. Proof of Thm. 1: Low-rank sub-Gaussian thinning

We establish the MMD bound (2) in App. C.1, the first kernel max seminorm bound (3) in App. C.2, and the Lipschitz kernel max seminorm bound (4) in App. C.3. Throughout, we use the notation $\mathbb{P}_{\mathcal{E}}(\mathcal{E}') \triangleq \mathbb{P}(\mathcal{E}, \mathcal{E}')$ for events $(\mathcal{E}, \mathcal{E}')$.

## C.1. Proof of MMD bound (2)

Without loss of generality, we suppose that $r \leq \text{rank}(\mathbf{K})$. Let $\mathbf{V}\boldsymbol{\Lambda}\mathbf{V}^\top$ be an eigendecomposition of $\mathbf{K}$ with orthonormal $\mathbf{V} \in \mathbb{R}^{n \times n}$ and diagonal $\boldsymbol{\Lambda} = \text{diag}(\lambda_1, \cdots, \lambda_n) \in \mathbb{R}^{n \times n}$. Let $\mathbf{V}_r$ represent the first $r$ columns of $\mathbf{V}$, and let $\mathbf{V}_{-r}$ represent the last $n - r$ columns of $\mathbf{V}$. Introduce the shorthand

$$\boldsymbol{w} \triangleq \boldsymbol{p}_{\text{in}} - \boldsymbol{p}_{\text{out}} \in \mathbb{R}^n \quad \text{and} \quad \boldsymbol{\Phi} \triangleq \mathbf{V}\boldsymbol{\Lambda}^{1/2}\mathbf{V}^\top \in \mathbb{R}^{n \times n}. \tag{24}$$

We can directly verify that

$$\mathbf{V}\mathbf{V}^\top = \mathbf{V}^\top\mathbf{V} = \mathbf{I}, \quad \mathbf{V}\mathbf{V}^\top = \mathbf{V}_r\mathbf{V}_r^\top + \mathbf{V}_{-r}\mathbf{V}_{-r}^\top, \quad \text{and} \quad \mathbf{K} = \boldsymbol{\Phi}\boldsymbol{\Phi}^\top. \tag{25}$$

Using the above equalities, we decompose the squared MMD into two components,

$$\text{MMD}^2_{\mathbf{K}}(\boldsymbol{p}_{\text{in}}, \boldsymbol{p}_{\text{out}}) = \boldsymbol{w}^\top\mathbf{K}\boldsymbol{w} = \boldsymbol{w}^\top\boldsymbol{\Phi}\boldsymbol{\Phi}^\top\boldsymbol{w} = \boldsymbol{w}^\top\boldsymbol{\Phi}\mathbf{V}\mathbf{V}^\top\boldsymbol{\Phi}^\top\boldsymbol{w} = \boldsymbol{w}^\top\boldsymbol{\Phi}\mathbf{V}_r\mathbf{V}_r^\top\boldsymbol{\Phi}^\top\boldsymbol{w} + \boldsymbol{w}^\top\boldsymbol{\Phi}\mathbf{V}_{-r}\mathbf{V}_{-r}^\top\boldsymbol{\Phi}^\top\boldsymbol{w}$$
$$= \|\mathbf{V}_r^\top\boldsymbol{\Phi}^\top\boldsymbol{w}\|_2^2 + \|\mathbf{V}_{-r}^\top\boldsymbol{\Phi}^\top\boldsymbol{w}\|_2^2. \tag{26}$$

In Apps. C.1.1 and C.1.2 respectively, we will establish the bounds

$$\mathbb{P}(\|\mathbf{V}_r^\top\boldsymbol{\Phi}^\top\boldsymbol{w}\|_2^2 \leq e\nu^2(er + \log(1/\delta'))) \geq 1 - \delta/2 - \delta' \quad \text{and} \tag{27}$$
$$\mathbb{P}(\|\mathbf{V}_{-r}^\top\boldsymbol{\Phi}^\top\boldsymbol{w}\|_2^2 \leq \lambda_{r+1}(\tfrac{1}{n_{\text{out}}} - \tfrac{1}{n})) = 1, \tag{28}$$

which when combined with (26) yield the advertised claim (2) on the squared MMD.

### C.1.1. PROOF OF (27): BOUNDING $\|\mathbf{V}_r^\top\boldsymbol{\Phi}^\top\boldsymbol{w}\|_2^2$

Our first lemma bounds the Euclidean norm of a vector in terms of a finite number of inner products.

**Lemma C.1 (Euclidean norm cover).** *For any $\boldsymbol{v} \in \mathbb{R}^r$ and $\varepsilon \in (0, 1)$,*

$$\|\boldsymbol{v}\|_2 \leq \tfrac{1}{1-\varepsilon}\max_{\boldsymbol{u}\in\mathcal{C}_{\varepsilon,r}}\langle\boldsymbol{u}, \boldsymbol{v}\rangle \tag{29}$$

*for a set $\mathcal{C}_{\varepsilon,r}$ contained in the ball $\mathbb{B}^r$ with $|\mathcal{C}_{\varepsilon,r}| \leq (1 + 2/\varepsilon)^r$.*

*Proof.* Fix any $\varepsilon \in (0, 1)$, and let $\mathcal{C}_{\varepsilon,r}$ be a set of minimum cardinality satisfying

$$\mathcal{C}_{\varepsilon,r} \subset \mathbb{B}^r \quad \text{and} \quad \sup_{\boldsymbol{u}\in\mathbb{B}^r}\min_{\boldsymbol{u}'\in\mathcal{C}_{\varepsilon,r}}\|\boldsymbol{u} - \boldsymbol{u}'\|_2 \leq \varepsilon.$$

By Wainwright (2019, Lem. 5.2), $|\mathcal{C}_{\varepsilon,r}| \leq (1 + 2/\varepsilon)^r$. Now we invoke the variational representation of $\|\cdot\|_2$ and the Cauchy-Schwarz inequality to conclude that

$$\|\boldsymbol{v}\|_2 = \sup_{\boldsymbol{u}\in\mathbb{B}^r}\langle\boldsymbol{u}, \boldsymbol{v}\rangle = \sup_{\boldsymbol{u}\in\mathbb{B}^r}\min_{\boldsymbol{u}'\in\mathcal{C}_{\varepsilon,r}}[\langle\boldsymbol{u} - \boldsymbol{u}', \boldsymbol{v}\rangle + \langle\boldsymbol{u}', \boldsymbol{v}\rangle]$$
$$\leq \sup_{\boldsymbol{u}\in\mathbb{B}^r}\min_{\boldsymbol{u}'\in\mathcal{C}_{\varepsilon,r}}\|\boldsymbol{u} - \boldsymbol{u}'\|_2\|\boldsymbol{v}\|_2 + \max_{\boldsymbol{u}'\in\mathcal{C}_{\varepsilon,r}}\langle\boldsymbol{u}', \boldsymbol{v}\rangle$$
$$\leq \varepsilon\|\boldsymbol{v}\|_2 + \max_{\boldsymbol{u}'\in\mathcal{C}_{\varepsilon,r}}\langle\boldsymbol{u}', \boldsymbol{v}\rangle.$$

Rearranging terms yields the claimed bound (29). $\qquad\square$

Our next lemma uses this covering estimate to bound the exponential moments of $\|\mathbf{V}_r^\top\boldsymbol{\Phi}^\top\boldsymbol{w}\|_2$.

**Lemma C.2 (Norm sub-Gaussianity).** *For any $\varepsilon > 0$ and any $t > 0$,*

$$\mathbb{E}_\mathcal{E}[\exp(t\|\mathbf{V}_r^\top\boldsymbol{\Phi}^\top\boldsymbol{w}\|_2)] \leq (1 + \tfrac{2}{\varepsilon})^r\exp(\tfrac{\nu^2 t^2}{2(1-\epsilon)^2}).$$

*Proof.* Fix any $t > 0$. Since $x \mapsto \exp(tx)$ is increasing, Lem. C.1 implies that

$$\mathbb{E}_\mathcal{E}[\exp(t\|\mathbf{V}_r^\top\boldsymbol{\Phi}^\top\boldsymbol{w}\|_2)] \leq \mathbb{E}_\mathcal{E}[\exp(t \cdot \tfrac{1}{1-\varepsilon}\max_{\boldsymbol{u}\in\mathcal{C}_{\varepsilon,r}}\langle\boldsymbol{u}, \mathbf{V}_r^\top\boldsymbol{\Phi}^\top\boldsymbol{w}\rangle)]$$
$$= \mathbb{E}_\mathcal{E}[\max_{\boldsymbol{u}\in\mathcal{C}_{\varepsilon,r}}\exp(\tfrac{t}{1-\varepsilon}\langle\mathbf{V}_r\boldsymbol{u}, \boldsymbol{\Phi}^\top\boldsymbol{w}\rangle)]$$
$$\leq \sum_{\boldsymbol{u}\in\mathcal{C}_{\varepsilon,r}}\mathbb{E}_\mathcal{E}[\exp(\tfrac{t}{1-\varepsilon}\langle\mathbf{V}_r\boldsymbol{u}, \boldsymbol{\Phi}^\top\boldsymbol{w}\rangle)]$$

for a subset $\mathcal{C}_{\varepsilon,r}$ with $|\mathcal{C}_{\varepsilon,r}| \leq (1 + \frac{2}{\varepsilon})^r$ and $\|\boldsymbol{u}\|_2 \leq 1$ for each $\boldsymbol{u} \in \mathcal{C}_{\varepsilon,r}$.

Now fix any $\boldsymbol{u} \in \mathcal{C}_{\varepsilon,r}$ and let $\boldsymbol{\Lambda}_r = \mathrm{diag}(\lambda_1, \ldots, \lambda_r)$. Using (24) and (25), we have

$$\mathbf{V}_r = \boldsymbol{\Phi}^\top \mathbf{V}_r \boldsymbol{\Lambda}_r^{-1/2} \quad \text{and therefore}$$

$$\langle \mathbf{V}_r \boldsymbol{u}, \boldsymbol{\Phi}^\top \boldsymbol{w} \rangle = \langle \boldsymbol{\Phi}^\top \mathbf{V}_r \boldsymbol{\Lambda}_r^{-1/2} \boldsymbol{u}, \boldsymbol{\Phi}^\top \boldsymbol{w} \rangle = \langle \mathbf{V}_r \boldsymbol{\Lambda}_r^{-1/2} \boldsymbol{u}, \mathbf{K} \boldsymbol{w} \rangle.$$

In addition, we have

$$(\mathbf{V}_r \boldsymbol{\Lambda}_r^{-1/2} \boldsymbol{u})^\top \mathbf{K} (\mathbf{V}_r \boldsymbol{\Lambda}_r^{-1/2} \boldsymbol{u}) = \boldsymbol{u}^\top \boldsymbol{\Lambda}_r^{-1/2} \mathbf{V}_r^\top \mathbf{V} \boldsymbol{\Lambda} \mathbf{V}^\top \mathbf{V}_r \boldsymbol{\Lambda}_r^{-1/2} \boldsymbol{u} = \boldsymbol{u}^\top \boldsymbol{u}.$$

Next, we can invoke our sub-Gaussianity assumption (Def. 3) to conclude that

$$\mathbb{E}_{\mathcal{E}}[\exp(\tfrac{t}{1-\varepsilon} \langle \mathbf{V}_r \boldsymbol{u}, \boldsymbol{\Phi}^\top \boldsymbol{w} \rangle)] = \mathbb{E}_{\mathcal{E}}[\exp(\tfrac{t}{1-\varepsilon} \langle \mathbf{V}_r \boldsymbol{\Lambda}_r^{-1/2} \boldsymbol{u}, \mathbf{K} \boldsymbol{w} \rangle)] \leq \exp(\tfrac{\nu^2 t^2}{2(1-\varepsilon)^2} \langle \mathbf{V}_r \boldsymbol{\Lambda}_r^{-1/2} \boldsymbol{u}, \mathbf{K} \mathbf{V}_r \boldsymbol{\Lambda}_r^{-1/2} \boldsymbol{u} \rangle)$$

$$\leq \exp(\tfrac{\nu^2 t^2}{2(1-\varepsilon)^2} \|\boldsymbol{u}\|_2^2).$$

Since $\|\boldsymbol{u}\|_2 \leq 1$ and $|\mathcal{C}_{\varepsilon,r}| \leq (1 + \frac{2}{\varepsilon})^r$, the advertised result now follows. $\qquad\square$

By Markov's inequality (Markov, 1884) and Lem. C.2, for any $\alpha > 0$,

$$\begin{aligned}
\mathbb{P}(\|\mathbf{V}_r^\top \boldsymbol{\Phi}^\top \boldsymbol{w}\|_2 > \alpha) &= \mathbb{P}_{\mathcal{E}}(\|\mathbf{V}_r^\top \boldsymbol{\Phi}^\top \boldsymbol{w}\|_2 > \alpha) + \mathbb{P}(\|\mathbf{V}_r^\top \boldsymbol{\Phi}^\top \boldsymbol{w}\|_2 > \alpha, \mathcal{E}^c) \\
&\leq \mathbb{P}_{\mathcal{E}}(\|\mathbf{V}_r^\top \boldsymbol{\Phi}^\top \boldsymbol{w}\|_2 > \alpha) + \mathbb{P}(\mathcal{E}^c) \\
&\leq \inf_{t>0} \mathbb{E}_{\mathcal{E}}[\exp(t\|\mathbf{V}_r^\top \boldsymbol{\Phi}^\top \boldsymbol{w}\|_2)] / \exp(t\alpha) + \delta/2 \\
&\leq (1 + \tfrac{2}{\varepsilon})^r \inf_{t>0} \exp(\tfrac{\nu^2 t^2}{2(1-\varepsilon)^2} - t\alpha) + \delta/2 \\
&= (1 + \tfrac{2}{\varepsilon})^r \exp(\tfrac{-(1-\varepsilon)^2 \alpha^2}{2\nu^2}) + \delta/2.
\end{aligned}$$

Next, we have

$$(1 + \tfrac{2}{\varepsilon})^r \exp(\tfrac{-(1-\varepsilon)^2 \alpha^2}{2\nu^2}) \leq \delta' \quad \text{if} \quad \alpha \geq \tfrac{\nu\sqrt{2}}{1-\varepsilon} \sqrt{\log(\tfrac{1}{\delta'}) + r \log(1 + \tfrac{2}{\varepsilon})}.$$

Since this bound holds for any $\varepsilon$, choosing $\varepsilon = 1 - \sqrt{2/e}$, we find that

$$\|\mathbf{V}_r^\top \boldsymbol{\Phi}^\top \boldsymbol{w}\|_2^2 \leq e\nu^2 \Big[ r \log(1 + 2/(1 - \sqrt{2/e})) + \log(1/\delta') \Big] \leq e\nu^2 [er + \log(1/\delta')]$$

with probability at least $1 - \delta/2 - \delta'$ as claimed.

C.1.2. PROOF OF (28): BOUNDING $\|\mathbf{V}_{-r}^\top \boldsymbol{\Phi}^\top \boldsymbol{w}\|_2^2$

Since

$$\|\boldsymbol{w}\|_2^2 = \boldsymbol{p}_{\mathrm{in}}^\top \boldsymbol{p}_{\mathrm{in}} + \boldsymbol{p}_{\mathrm{out}}^\top \boldsymbol{p}_{\mathrm{out}} - 2\boldsymbol{p}_{\mathrm{in}}^\top \boldsymbol{p}_{\mathrm{out}} = \tfrac{n_{\mathrm{in}}}{n_{\mathrm{in}}^2} + \tfrac{n_{\mathrm{out}}}{n_{\mathrm{out}}^2} - \tfrac{2n_{\mathrm{out}}}{n_{\mathrm{in}} n_{\mathrm{out}}} = \tfrac{1}{n_{\mathrm{out}}} - \tfrac{1}{n_{\mathrm{in}}}, \tag{30}$$

we have, for $\boldsymbol{\Lambda}_{-r} \triangleq \mathrm{diag}(\lambda_{r+1}, \cdots, \lambda_n)$ and $\lambda_{\max}$ the maximum eigenvalue of a SPSD matrix,

$$\|\mathbf{V}_{-r}^\top \boldsymbol{\Phi}^\top \boldsymbol{w}\|_2^2 = \boldsymbol{w}^\top \mathbf{V}_{-r} \boldsymbol{\Lambda}_{-r} \mathbf{V}_{-r}^\top \boldsymbol{w} \leq \lambda_{\max}(\mathbf{V}_{-r} \boldsymbol{\Lambda}_{-r} \mathbf{V}_{-r}^\top) \|\boldsymbol{w}\|_2^2 \overset{(30)}{=} \lambda_{r+1}(\tfrac{1}{n_{\mathrm{out}}} - \tfrac{1}{n_{\mathrm{in}}}).$$

## C.2. Proof of kernel max seminorm bound (3)

We begin by establishing a general bound on the maximum discrepancy between input and output expectations over a collection of test functions admitting a finite cover.

**Lemma C.3** (**Discrepancy cover bound**). *Fix any kernel* $\mathbf{k}$, *subset* $\mathcal{F} \subset \mathcal{H}_{\mathbf{k}}$, *and scalars* $\varepsilon \geq 0$ *and* $\delta' \in (0,1)$. *Define*

$$a \triangleq \sup_{f \in \mathcal{F}} \|f\|_{\mathbf{k}} \quad and \quad \mathbb{B}_{\mathcal{F}} \triangleq \{f \in \mathcal{H}_{\mathbf{k}} : \|f\|_{\mathbf{k}} \leq a\},$$

and let $\mathcal{C}_{\epsilon,\mathcal{F}}$ be a set of minimum cardinality satisfying

$$\mathcal{C}_{\epsilon,\mathcal{F}} \subset \mathbb{B}_{\mathcal{F}} \quad and \quad \sup_{f\in\mathcal{F}} \min_{f'\in\mathcal{C}_{\epsilon,\mathcal{F}}} \max_{\boldsymbol{x}\in\mathcal{X}_{\text{in}}} |f(\boldsymbol{x}) - f'(\boldsymbol{x})| \leq \varepsilon. \tag{31}$$

If $(\mathbb{P}_{\text{in}} - \mathbb{P}_{\text{out}})\mathbf{k}$ is $(\mathbf{k},\nu)$-sub-Gaussian on an event $\mathcal{E}$ (Def. A.2), then, on $\mathcal{E}$,

$$\|\mathbb{P}_{\text{in}} - \mathbb{P}_{\text{out}}\|_{\mathcal{F}} \triangleq \sup_{f\in\mathcal{F}}(\mathbb{P}_{\text{in}} - \mathbb{P}_{\text{out}})f \leq 2\epsilon + \nu a\sqrt{2\log(|\mathcal{C}_{\epsilon,\mathcal{F}}|/\delta')} \quad \text{with probability at least } 1-\delta'.$$

*Proof.* The triangle inequality and the covering property (31) together imply that, with probability 1,

$$
\begin{aligned}
(\mathbb{P}_{\text{in}} - \mathbb{P}_{\text{out}})f &\leq \min_{f'\in\mathcal{C}_{\epsilon,\mathcal{F}}}(\mathbb{P}_{\text{in}} - \mathbb{P}_{\text{out}})f' + |(\mathbb{P}_{\text{in}} - \mathbb{P}_{\text{out}})(f - f')| \\
&\leq \|\mathbb{P}_{\text{in}} - \mathbb{P}_{\text{out}}\|_{\mathcal{C}_{\epsilon,\mathcal{F}}} + \min_{f'\in\mathcal{C}_{\epsilon,\mathcal{F}}} |\mathbb{P}_{\text{in}}(f - f')| + |\mathbb{P}_{\text{out}}(f - f')| \\
&\leq \|\mathbb{P}_{\text{in}} - \mathbb{P}_{\text{out}}\|_{\mathcal{C}_{\epsilon,\mathcal{F}}} + 2\min_{f'\in\mathcal{C}_{\epsilon,\mathcal{F}}} \max_{\boldsymbol{x}\in\mathcal{X}_{\text{in}}} |f(\boldsymbol{x}) - f'(\boldsymbol{x})| \\
&\leq \|\mathbb{P}_{\text{in}} - \mathbb{P}_{\text{out}}\|_{\mathcal{C}_{\epsilon,\mathcal{F}}} + 2\varepsilon
\end{aligned}
\tag{32}
$$

for each $f \in \mathcal{F}$. Since $s \mapsto e^{ts}$ is increasing, the bound (32), the assumed sub-Gaussianity (Def. A.2), and the fact that $\mathcal{C}_{\epsilon,\mathcal{F}}$ belongs to $\mathbb{B}_{\mathcal{F}}$ imply that

$$
\begin{aligned}
\mathbb{E}_{\mathcal{E}}[\exp(t\|\mathbb{P}_{\text{in}} - \mathbb{P}_{\text{out}}\|_{\mathcal{F}})] &\leq e^{2t\varepsilon}\mathbb{E}_{\mathcal{E}}[\exp(t\|\mathbb{P}_{\text{in}} - \mathbb{P}_{\text{out}}\|_{\mathcal{C}_{\epsilon,\mathcal{F}}})] \\
&\leq \sum_{f'\in\mathcal{C}_{\epsilon,\mathcal{F}}} e^{2t\varepsilon}\mathbb{E}_{\mathcal{E}}[\exp(t(\mathbb{P}_{\text{in}} - \mathbb{P}_{\text{out}})f')] \\
&\leq \sum_{f'\in\mathcal{C}_{\epsilon,\mathcal{F}}} \exp(\tfrac{t^2\nu^2\|f'\|_{\mathbf{k}}^2}{2} + 2t\epsilon) \leq |\mathcal{C}_{\epsilon,\mathcal{F}}| \exp(\tfrac{t^2\nu^2 a^2}{2} + 2t\epsilon).
\end{aligned}
$$

Now, by Markov's inequality (Markov, 1884), for any $\alpha > 0$,

$$
\begin{aligned}
\mathbb{P}_{\mathcal{E}}(\sup_{f\in\mathcal{F}}(\mathbb{P}_{\text{in}} - \mathbb{P}_{\text{out}})f > \alpha + 2\epsilon) &\leq \inf_{t>0} \mathbb{E}_{\mathcal{E}}[\exp(t\|\mathbb{P}_{\text{in}} - \mathbb{P}_{\text{out}}\|_{\mathcal{F}})]/\exp(t(\alpha + 2\epsilon)) \\
&\leq |\mathcal{C}_{\epsilon,\mathcal{F}}| \inf_{t>0} \exp(\tfrac{t^2\nu^2 a^2}{2} - t\alpha) = |\mathcal{C}_{\epsilon,\mathcal{F}}| \exp(\tfrac{-\alpha^2}{2\nu^2 a^2}).
\end{aligned}
$$

Finally, choosing $\alpha = \nu a\sqrt{2\log(|\mathcal{C}_{\epsilon,\mathcal{F}}|/\delta')}$ yields the desired claim. $\qquad\square$

Now fix any $\epsilon \geq 0$, $\delta' \in (0,1)$, and kernel $\mathbf{k}$ that generates $\mathbf{K}$, and consider the subset $\mathcal{F} = \{\pm\mathbf{k}(\boldsymbol{x}_i,\cdot) : i \in \mathcal{I}\}$. Since $\|\mathbf{K}(\boldsymbol{p}_{\text{in}} - \boldsymbol{p}_{\text{out}})\|_{\mathcal{I}} = \|\mathbb{P}_{\text{in}} - \mathbb{P}_{\text{out}}\|_{\mathcal{F}}$ and $\sup_{f\in\mathcal{F}} \|f\|_{\mathbf{k}} = D_{\mathcal{I}}$, Lem. C.3 implies that, on the event $\mathcal{E}$,

$$\|\mathbf{K}(\boldsymbol{p}_{\text{in}} - \boldsymbol{p}_{\text{out}})\|_{\mathcal{I}} \leq 2\epsilon + \nu D_{\mathcal{I}}\sqrt{2\log(|\mathcal{C}_{\epsilon,\mathcal{F}}|/\delta')} \quad \text{with probability at least } 1-\delta'.$$

Since $\mathbb{P}(\mathcal{E}^c) \leq \delta/2$ and $|\mathcal{F}| \leq 2|\mathcal{Z}|$, we use the estimate $|\mathcal{C}_{0,\mathcal{F}}| \leq 2|\mathcal{I}|$ with $\epsilon = 0$ to obtain the advertised bound (3).

### C.3. Proof of Lipschitz kernel max seminorm bound (4)

Introduce the query point set $\mathcal{Z} \triangleq \{\boldsymbol{x}_i : i \in \mathcal{I}\}$, fix any $\delta' \in (0,1)$ and $\boldsymbol{z}_0 \in \mathcal{Z}$, and define the symmetrized seminorm

$$\|(\mathbb{P}_{\text{in}} - \mathbb{P}_{\text{out}})\mathbf{k}\|_{\mathcal{Z},\mathcal{Z}} \triangleq \sup_{\boldsymbol{z},\boldsymbol{z}'\in\mathcal{Z}} |(\mathbb{P}_{\text{in}} - \mathbb{P}_{\text{out}})\mathbf{k}(\boldsymbol{z}) - (\mathbb{P}_{\text{in}} - \mathbb{P}_{\text{out}})\mathbf{k}(\boldsymbol{z}')|.$$

By the triangle inequality and the derivation of App. C.2, we have, on the event $\mathcal{E}$,

$$
\begin{aligned}
\|\mathbf{K}(\boldsymbol{p}_{\text{in}} - \boldsymbol{p}_{\text{out}})\|_{\mathcal{I}} &\leq \|(\mathbb{P}_{\text{in}} - \mathbb{P}_{\text{out}})\mathbf{k}\|_{\mathcal{Z},\mathcal{Z}} + |(\mathbb{P}_{\text{in}} - \mathbb{P}_{\text{out}})\mathbf{k}(\boldsymbol{z}_0)| \\
&\leq \|(\mathbb{P}_{\text{in}} - \mathbb{P}_{\text{out}})\mathbf{k}\|_{\mathcal{Z},\mathcal{Z}} + \nu\sqrt{\mathbf{k}(\boldsymbol{z}_0,\boldsymbol{z}_0)}\sqrt{2\log(4/\delta')} \quad \text{with probability at least } 1-\delta'/2.
\end{aligned}
\tag{33}
$$

Since $\mathbb{P}(\mathcal{E}^c) \leq \delta/2$, it only remains to upper bound $\|(\mathbb{P}_{\text{in}} - \mathbb{P}_{\text{out}})\mathbf{k}\|_{\mathcal{Z},\mathcal{Z}}$ on $\mathcal{E}$ with probability at least $1-\delta'/2$.

To this end, we first establish that $((\mathbb{P}_{\text{in}} - \mathbb{P}_{\text{out}})\mathbf{k}(\boldsymbol{z}))_{\boldsymbol{z}\in\mathcal{Z}}$ is a sub-Gaussian process on $\mathcal{E}$ with respect to a particular bounded-Hölder metric $\rho$.

**Definition C.1** (Sub-Gaussian process on an event). *We say an indexed collection of random variables $(X_\theta)_{\theta\in\Theta}$ is a sub-Gaussian process with respect to $\rho$ on an event $\mathcal{E}$ if $\rho$ is a metric on $\Theta$ and*

$$\mathbb{E}_{\mathcal{E}}\left[\exp\left(\tfrac{(X_\theta - X'_\theta)^2}{\rho(\theta,\theta')^2}\right)\right] \leq 2 \quad \text{for all} \quad \theta,\theta' \in \Theta.$$

**Lemma C.4** (**Bounded-Hölder sub-Gaussian process**). *Consider a kernel $\mathbf{k}$ on $\mathcal{X} = \mathbb{R}^d$ satisfying $|\mathbf{k}(\boldsymbol{z}, \boldsymbol{x}) - \mathbf{k}(\boldsymbol{z}', \boldsymbol{x})| \leq L_{\mathbf{k}} \|\boldsymbol{z} - \boldsymbol{z}'\|_2$ for all $\boldsymbol{z}, \boldsymbol{z}' \in \mathcal{Z} \subset \mathcal{X}$ and $\boldsymbol{x} \in \mathcal{X}_{\mathrm{in}}$. If $(\mathbb{P}_{\mathrm{in}} - \mathbb{P}_{\mathrm{out}})\mathbf{k}$ is $(\mathbf{k}, \nu)$-sub-Gaussian on an event $\mathcal{E}$ (Def. A.2), then $((\mathbb{P}_{\mathrm{in}} - \mathbb{P}_{\mathrm{out}})\mathbf{k}(\boldsymbol{z}))_{\boldsymbol{z} \in \mathcal{Z}}$ is a sub-Gaussian process on $\mathcal{E}$ with respect to the metric*

$$\rho(\boldsymbol{z}, \boldsymbol{z}') \triangleq \nu \sqrt{8/3} \min(2 \sup_{\boldsymbol{z} \in \mathcal{Z}} \sqrt{\mathbf{k}(\boldsymbol{z}, \boldsymbol{z})}, \sqrt{2 L_{\mathbf{k}} \|\boldsymbol{z} - \boldsymbol{z}'\|_2}). \tag{34}$$

The proof of Lem. C.4 can be found in App. C.4. Our next lemma, a slight modification of Wainwright (2019, Thm. 5.36), bounds the suprema of symmetrized sub-Gaussian processes on an event in terms of covering numbers.

**Lemma C.5** (**Sub-Gaussian process tails**). *Suppose $(X_\theta)_{\theta \in \Theta}$ is a sub-Gaussian process with respect to $\rho$ on an event $\mathcal{E}$, and define the diameter $\mathrm{diam}(\Theta, \rho) \triangleq \sup_{\theta, \theta' \in \Theta} \rho(\theta, \theta')$, the covering number*

$$\mathcal{N}(u; \Theta, \rho) \triangleq \min\{|\mathcal{C}_u| : \mathcal{C}_u \subseteq \Theta, \max_{\theta \in \Theta} \min_{\theta' \in \mathcal{C}_u} \rho(\theta, \theta') \leq u\} \quad \text{for all } u > 0,$$

*and the entropy integral $\mathcal{J}(\Theta, \rho) \triangleq \int_0^{\mathrm{diam}(\Theta, \rho)} \sqrt{\log(1 + \mathcal{N}(u; \Theta, \rho))} \, du$. Then,*

$$\mathbb{P}_{\mathcal{E}}(\sup_{\theta, \theta' \in \Theta} |X_\theta - X_{\theta'}| \geq 8(\mathcal{J}(\Theta, \rho) + t)) \leq 2 \exp(-t^2 / \mathrm{diam}(\Theta, \rho)^2) \quad \text{for all} \quad t > 0.$$

*Proof.* Since $\sqrt{\log(1 + xy)} \leq \sqrt{\log((1 + x)(1 + y))} \leq \sqrt{\log(1 + x)} + \sqrt{\log(1 + y)}$ for all $x, y > 0$, the proof is identical to that of Wainwright (2019, Thm. 5.36) with $c_1 = 8$ and $(\mathbb{E}_{\mathcal{E}}, \mathbb{P}_{\mathcal{E}})$ substituted for $(\mathbb{E}, \mathbb{P})$. $\qquad \square$

Our final lemma bounds the diameter, covering numbers, and entropy integral of $\mathcal{Z}$ using the metric $\rho$.

**Lemma C.6** (**Covering properties of bounded-Hölder metric**). *Consider the bounded-Hölder metric $\rho$ (34) for a kernel $\mathbf{k}$ on $\mathcal{X} = \mathbb{R}^d$ and a finite set $\mathcal{Z} \subset \mathcal{X}$. If $\mathbf{Z}$ is a matrix with one row corresponding to each element of $\mathcal{Z}$, $r = \mathrm{rank}(\mathbf{Z})$, and $R = \max_{\boldsymbol{z} \in \mathcal{Z}} \|\boldsymbol{z}\|_2$, then, in the notation of Lem. C.5,*

$$\mathcal{N}(u; \mathcal{Z}, \rho) \leq (1 + c^2/u^2)^r \quad \text{for} \quad c \triangleq \nu \sqrt{\tfrac{32}{3} R L_{\mathbf{k}}} \quad \text{and all} \quad u > 0, \tag{35}$$

$$\mathrm{diam}(\mathcal{Z}, \rho) \leq D \triangleq \min(c, \nu \sqrt{\tfrac{32}{3}} \max_{\boldsymbol{z} \in \mathcal{Z}} \sqrt{\mathbf{k}(\boldsymbol{z}, \boldsymbol{z})}), \quad \text{and} \tag{36}$$

$$\mathcal{J}(\mathcal{Z}, \rho) \leq D \sqrt{2r \log(\sqrt{3} ec/D)}.$$

*Proof.* The diameter bound (36) follows directly from the definition of $\rho$ (34) and the fact $\max_{\boldsymbol{z}, \boldsymbol{z}' \in \mathcal{Z}} \|\boldsymbol{z} - \boldsymbol{z}'\|_2 \leq 2R$.

To establish the covering number bound (35), we let $\mathbf{U} \boldsymbol{\Sigma} \mathbf{V}^\top$ be a compact singular value decomposition of $\mathbf{Z}$ so that

$$\mathbf{V} \in \mathbb{R}^{d \times r}, \quad \mathbf{Z} = \mathbf{Z} \mathbf{V} \mathbf{V}^\top, \quad \text{and} \quad \max_{\boldsymbol{z} \in \mathcal{Z}} \|\mathbf{V}^\top \boldsymbol{z}\|_2 = \max_{\boldsymbol{z} \in \mathcal{Z}} \|\boldsymbol{z}\|_2 = R.$$

Fix any $\epsilon > 0$, and let $\mathcal{C}$ and $\mathcal{C}_{\mathrm{ext}}$ be a sets of minimum cardinality satisfying

$$\mathcal{C} \subset \mathbb{B}^r(R), \qquad \max_{\boldsymbol{v} \in \mathbb{B}^r(R)} \min_{\boldsymbol{v}' \in \mathcal{C}} \|\boldsymbol{v}' - \boldsymbol{v}\|_2 \leq \epsilon^2/2,$$
$$\mathcal{C}_{\mathrm{ext}} \subset \mathbb{B}^d(R), \quad \text{and} \quad \max_{\boldsymbol{z} \in \mathcal{Z}} \min_{\boldsymbol{z}' \in \mathcal{C}_{\mathrm{ext}}} \|\boldsymbol{z}' - \boldsymbol{z}\|_2 \leq \epsilon^2/2. \tag{37}$$

Since $\mathbf{V}^\top \boldsymbol{z} \in \mathbb{B}^r(R)$ for each $\boldsymbol{z} \in \mathcal{Z}$ and $\mathbf{V} \boldsymbol{v}' \in \mathbb{B}^d$ for each $\boldsymbol{v}' \in \mathbb{B}^r$, we have

$$\max_{\boldsymbol{z} \in \mathcal{Z}} \min_{\boldsymbol{v}' \in \mathcal{C}} \|\mathbf{V} \boldsymbol{v}' - \boldsymbol{z}\|_2 = \max_{\boldsymbol{z} \in \mathcal{Z}} \min_{\boldsymbol{v}' \in \mathcal{C}} \|\mathbf{V}(\boldsymbol{v}' - \mathbf{V}^\top \boldsymbol{z})\|_2$$
$$= \max_{\boldsymbol{z} \in \mathcal{Z}} \min_{\boldsymbol{v}' \in \mathcal{C}} \|\boldsymbol{v}' - \mathbf{V}^\top \boldsymbol{z}\|_2 \leq \epsilon^2/2,$$

so that $\mathbf{V}\mathcal{C}$ satisfies the criteria of (37). Since $|\mathbf{V}\mathcal{C}| \leq |\mathcal{C}| \leq (1 + 4R/\epsilon^2)^r$ by Wainwright (2019, Lem. 5.2), we must also have $|\mathcal{C}_{\mathrm{ext}}| \leq (1 + 4R/\epsilon^2)^r$.

Now, since $\mathcal{C}_{\mathrm{ext}}$ has minimum cardinality amongst sets satisfying (37), for each $\boldsymbol{z}' \in \mathcal{C}_{\mathrm{ext}}$, there is some $\boldsymbol{z} \in \mathcal{Z}$ satisfying $\|\boldsymbol{z}' - \boldsymbol{z}\|_2 \leq \epsilon^2/2$ (or else $\boldsymbol{z}'$ would be superfluous). Hence, there exists a set $\mathcal{C}_{\mathrm{int}} \subseteq \mathcal{Z}$ satisfying

$$|\mathcal{C}_{\mathrm{int}}| \leq |\mathcal{C}_{\mathrm{ext}}| \leq (1 + 4R/\epsilon^2)^r \quad \text{and} \quad \max_{\boldsymbol{z} \in \mathcal{Z}} \min_{\boldsymbol{z}' \in \mathcal{C}_{\mathrm{int}}} \|\boldsymbol{z}' - \boldsymbol{z}\|_2 \leq \epsilon^2.$$

Moreover, by our metric definition (34),

$$\max_{\boldsymbol{z} \in \mathcal{Z}} \min_{\boldsymbol{z}' \in \mathcal{C}_{\text{int}}} \rho(\boldsymbol{z}, \boldsymbol{z}') \leq \frac{c}{2\sqrt{R}} \max_{\boldsymbol{z} \in \mathcal{Z}} \min_{\boldsymbol{z}' \in \mathcal{C}_{\text{int}}} \sqrt{\|\boldsymbol{z} - \boldsymbol{z}'\|_2} \leq \frac{c\epsilon}{2\sqrt{R}}.$$

Hence, for $u = \frac{c\epsilon}{2\sqrt{R}}$, $\mathcal{N}(u; \mathcal{Z}, \rho) \leq |\mathcal{C}_{\text{int}}| \leq (1 + c^2/u^2)^r$. Since $\epsilon > 0$ was arbitrary, we have established (35).

Finally, we bound the entropy integral using the inequality $1 \leq c^2/u^2$ for $u \in [0, D]$, the concavity of the square-root function, and Jensen's inequality:

$$\mathcal{J}(\mathcal{Z}, \rho) \leq \int_0^D \sqrt{\log(1 + (1 + c^2/u^2)^r)} \, du \leq \int_0^D \sqrt{\log((3c^2/u^2)^r)} \, du = \int_0^D \sqrt{2r \log(\sqrt{3}c/u)} \, du$$

$$\leq D\sqrt{\frac{1}{D} \int_0^D 2r \log(\sqrt{3}c/u) \, du} = D\sqrt{2r \log(\sqrt{3}ec/D)}.$$

$\square$

Together, Lems. C.4, C.5, and C.6 imply that, in the notation of Lem. C.6,

$$\|(\mathbb{P}_{\text{in}} - \mathbb{P}_{\text{out}})\mathbf{k}\|_{\mathcal{Z}, \mathcal{Z}} \leq 8D\sqrt{2r \log(\sqrt{3}ec/D)} + 8D\sqrt{\log(4/\delta')}$$

on $\mathcal{E}$ with probability at least $1 - \delta'/2$. Combining this bound with the inequality (33) yields the result.

### C.4. Proof of Lem. C.4: Bounded-Hölder sub-Gaussian process

Define $X_{\boldsymbol{z}} = (\mathbb{P}_{\text{in}} - \mathbb{P}_{\text{out}})\mathbf{k}(\boldsymbol{z})$ for each $\boldsymbol{z} \in \mathcal{Z}$, and fix any $\boldsymbol{z}, \boldsymbol{z}' \in \mathcal{Z}$. Our sub-Gaussianity assumption implies

$$\mathbb{E}_{\mathcal{E}}[\exp(\lambda(X_{\boldsymbol{z}} - X_{\boldsymbol{z}'}))] \leq \exp(\frac{\nu^2\lambda^2}{2}\|\mathbf{k}(\boldsymbol{z}, \cdot) - \mathbf{k}(\boldsymbol{z}', \cdot)\|_{\mathbf{k}}^2) \quad \text{for all} \quad \lambda \in \mathbb{R}.$$

Moreover, by our Lipschitz assumption,

$$\|\mathbf{k}(\boldsymbol{z}, \cdot) - \mathbf{k}(\boldsymbol{z}', \cdot)\|_{\mathbf{k}}^2 = \mathbf{k}(\boldsymbol{z}, \boldsymbol{z}) - \mathbf{k}(\boldsymbol{z}, \boldsymbol{z}') + \mathbf{k}(\boldsymbol{z}', \boldsymbol{z}') - \mathbf{k}(\boldsymbol{z}', \boldsymbol{z}) \leq \min(4\max_{\boldsymbol{z} \in \mathcal{Z}} \mathbf{k}(\boldsymbol{z}, \boldsymbol{z}), 2L_{\mathbf{k}}\|\boldsymbol{z} - \boldsymbol{z}'\|_2).$$

Finally, Lem. C.7 shows that $\mathbb{E}_{\mathcal{E}}[\exp(\frac{(X_{\boldsymbol{z}} - X_{\boldsymbol{z}'})^2}{\rho(\boldsymbol{z}, \boldsymbol{z}')^2})] \leq 2$ so that $(X_{\boldsymbol{z}})_{\boldsymbol{z} \in \mathcal{Z}}$ is a sub-Gaussian process on $\mathcal{E}$ with respect to $\rho$.

**Lemma C.7** (**Squared exponential moment bound**). *If* $\mathbb{E}_{\mathcal{E}}[\exp(\lambda X)] \leq \exp(\frac{\nu^2\lambda^2}{2})$ *for all* $\lambda \in \mathbb{R}$, *then* $\mathbb{E}_{\mathcal{E}}[\exp(\frac{3X^2}{8\nu^2})] \leq 2$.

*Proof.* The proof is identical to that in Wainwright (2019, Sec. 2.4) with $\mathbb{E}_{\mathcal{E}}$ substituted for $\mathbb{E}$. $\square$

## D. Proof of Cor. 1: Gaussian MMD of KH

Cor. 1 follows immediately from the following explicit, non-asymptotic bound.

**Corollary D.1** (**Detailed Gaussian MMD of KH**). *If* $\mathcal{X}_{\text{in}} \subset \mathbb{B}^d(R)$ *for* $R > 0$, *then* KH($\delta$) *with* $\mathbf{k} = $ GAUSS($\eta$), $n = n_{\text{in}} \geq (2e)^d$, *and* $b \triangleq \frac{1}{2}$ *delivers*

$$\text{MMD}_{\mathbf{K}}^2(\boldsymbol{p}_{\text{in}}, \boldsymbol{p}_{\text{out}}) \leq \frac{1}{n_{\text{out}}^2} \log(\frac{4n_{\text{out}}}{\delta})\Big[e^2 \max\Big\{\big[\frac{2e}{d}\log(n_{\text{in}}n_{\text{out}}b)\big]^d, (\frac{R^2\eta e^3 4}{d})^d\Big\} + e\log(\frac{1}{\delta'})\Big] + \frac{1}{n_{\text{out}}b}(\frac{1}{n_{\text{out}}} - \frac{1}{n_{\text{in}}})$$

*with probability at least* $1 - \delta/2 - \delta'$.

*Proof.* Consider the approximate rank parameter

$$r^{\star} \triangleq \max\Big\{\big[\frac{2e}{d}\log(n_{\text{in}}n_{\text{out}}b)\big]^d, (R^2\eta e^3 4/d)^d\Big\}.$$

The assumption $n_{\text{in}} \geq (2e)^d$ and the fact that $b \geq 1/(2^d n_{\text{out}})$ ensure that $\log(n_{\text{in}}n_{\text{out}}b) \geq d + \log(n_{\text{out}}b/2^d) \geq d$ and therefore that $r^{\star} \geq (2e)^d$. Hence, by Altschuler et al. (2019, Thm. 3), the $(r^{\star} + 1)$-th eigenvalue of $\mathbf{K}$ satisfies

$$\lambda_{r^{\star}+1} \leq n_{\text{in}} \exp\Big\{-\frac{d}{2e} \max\big\{\frac{2e}{d}\log(n_{\text{in}}n_{\text{out}}b), (R^2\eta e^3 4/d)\big\} \log\Big(\frac{d\max\big\{\frac{2e}{d}\log(n_{\text{in}}n_{\text{out}}b), (R^2\eta e^3 4/d)\big\}}{4e^2\eta R^2}\Big)\Big\}$$

$$\leq n_{\text{in}} \exp\{-\log(n_{\text{in}}n_{\text{out}}b)\log(e)\} \leq n_{\text{in}}\Big(\frac{1}{n_{\text{in}}n_{\text{out}}b}\Big) = \frac{1}{n_{\text{out}}b}.$$

Since $\|\mathbf{K}\|_{\max} = 1$ and $\mathrm{KH}(\delta) \in \mathcal{G}_\nu(\mathbf{K})$ with $\nu$ defined in Prop. B.2, the result now follows from Thm. 1.

$\square$

## E. Proof of Cor. 2: Intrinsic Gaussian MMD of KH

**Assumption E.1** ($d^\star$**-manifold with** $Q$**-smooth atlas** (Altschuler et al., 2019, Assum. 1))**.** *Let* $\Omega \subset \mathbb{R}^d$ *be a smooth compact manifold without boundary of dimension* $d^\star < d$. *Let* $(\boldsymbol{\Psi}_j, U_j)_{j \in [T]}$ *for* $T \in \mathbb{N}$ *be an atlas for* $\Omega$, *where* $(U_j)_j$ *are open sets covering* $\Omega$ *and* $\boldsymbol{\Psi}_j : U_j \mapsto \mathbb{B}^{d^\star}(r_j)$ *are smooth maps with smooth inverses, mapping* $U_j$ *bijectively to* $\mathbb{B}^{d^\star}(r_j)$. *Assume that there exists* $Q > 0$ *such that* $\sup_{u \in \mathbb{B}^{d^\star}(r_j)} \|D^{\boldsymbol{\alpha}} \boldsymbol{\Psi}_j^{-1}(u)\| \leq Q^{|\boldsymbol{\alpha}|}$ *for all* $\boldsymbol{\alpha} \in \mathbb{N}^{d^\star}$ *and* $j \in [T]$, *where* $|\boldsymbol{\alpha}| \triangleq \sum_{j=1}^{d^\star} \alpha_j$ *and* $D^{\boldsymbol{\alpha}} = \frac{\partial^{|\boldsymbol{\alpha}|}}{\partial u_1^{\alpha_1} \ldots \partial u_{d^\star}^{\alpha_{d^\star}}}$ *for* $\boldsymbol{\alpha} \in \mathbb{N}^{d^\star}$.

Cor. 2 follows immediately from the following more detailed result.

**Corollary E.1** (**Detailed Intrinsic Gaussian MMD of KH**)**.** *Suppose* $\mathcal{X}_{\mathrm{in}}$ *lies on a manifold* $\Omega \subset \mathbb{B}^d$ *satisfying Assump. E.1. Then* $\mathrm{KH}(\delta)$ *with* $\mathbf{k} = \mathrm{GAUSS}(\eta)$ *and* $n = n_{\mathrm{in}}$ *delivers*

$$\mathrm{MMD}_{\mathbf{K}}^2(\boldsymbol{p}_{\mathrm{in}}, \boldsymbol{p}_{\mathrm{out}}) \leq \tfrac{1}{n_{\mathrm{out}}^2} \log\big(\tfrac{4 n_{\mathrm{out}}}{\delta}\big) \big(\tfrac{e^2}{c^{5 d^\star/2}} \log^{\frac{5 d^\star}{2}}(n_{\mathrm{in}} n_{\mathrm{out}}) + e \log(\tfrac{1}{\delta'})\big) + \tfrac{1}{n_{\mathrm{out}}}\big(\tfrac{1}{n_{\mathrm{out}}} - \tfrac{1}{n_{\mathrm{in}}}\big)$$

*with probability at least* $1 - \frac{\delta}{2} - \delta'$ *for* $c$ *independent of* $\mathcal{X}_{\mathrm{in}}$.

*Proof.* Altschuler et al. (2019, Thm. 4) showed that the $(r+1)$-th eigenvalue of $\mathbf{K}$ satisfies (7) for a constant $c$ independent of $\mathcal{X} = \mathcal{X}_{\mathrm{in}}$. Since $\|\mathbf{K}\|_{\max} = 1$ and $\mathrm{KH}(\delta) \in \mathcal{G}_\nu(\mathbf{K})$ with $\nu$ defined in Prop. B.2, the result now follows from Thm. 1 with $r = (\log(n_{\mathrm{in}} n_{\mathrm{out}})/c)^{5 d^\star/2}$. $\square$

## F. Proof of Thm. 2: Quality of Thinformer

Throughout we will make use of the convenient representation

$$\widehat{\mathbf{T}} = \widehat{\mathbf{D}}^{-1} \widehat{\mathbf{A}} \mathbf{V} \text{ for } \mathcal{I}_{\mathrm{out}} \triangleq \{i \in [n] : (\tilde{\boldsymbol{k}}_i, \tilde{\boldsymbol{v}}_i) \in \mathcal{X}_{\mathrm{out}}\}, \quad \widehat{\mathbf{A}} \triangleq \tfrac{n}{n_{\mathrm{out}}} (\exp(\tfrac{\langle \boldsymbol{q}_i, \boldsymbol{k}_j \rangle}{\sqrt{d}}) \mathbf{1}[j \in \mathcal{I}_{\mathrm{out}}])_{i,j=1}^n, \text{ and } \widehat{\mathbf{D}} \triangleq \widehat{\mathbf{A}} \mathbf{1}_n. \quad (38)$$

Our proof makes use of three lemmas. The first, proved in App. F.1, bounds the approximation error for the attention matrix $\mathbf{T}$ in terms of the approximation error for $\mathbf{A}\mathbf{V}$ and $\mathbf{A}\mathbf{1}_n$.

**Lemma F.1** (**Decomposing attention approximation error**)**.** *In the notation of Alg. 1 and* (38),

$$\|\widehat{\mathbf{D}}^{-1} \widehat{\mathbf{A}} \mathbf{V} - \mathbf{D}^{-1} \mathbf{A} \mathbf{V}\|_{\max} \leq \min\big(\|(\tfrac{1}{n}\mathbf{D})^{-1}\|_{\max}, \|(\tfrac{1}{n}\widehat{\mathbf{D}})^{-1}\|_{\max}\big)\big(\tfrac{1}{n}\|\widehat{\mathbf{A}} \mathbf{V} - \mathbf{A} \mathbf{V}\|_{\max} + \tfrac{1}{n}\|\mathbf{A}\mathbf{1}_n - \widehat{\mathbf{A}}\mathbf{1}_n\|_\infty \|\mathbf{V}\|_{\max}\big).$$

The second, proved in App. F.2, bounds the approximation error for $\mathbf{A}\mathbf{V}$ and $\mathbf{A}\mathbf{1}_n$ in terms of the KMS (1) for a specific choice of attention kernel matrix.

**Lemma F.2** (**KMS bound on attention approximation error**)**.** *Instantiate the notation of Alg. 1 and* (38) *and define the query set*

$$\mathcal{X}' \triangleq \{\boldsymbol{x}_{i+nj} \triangleq (\tilde{\boldsymbol{q}}_i, \boldsymbol{e}_j^{d+1}) : i \in [n], j \in [d+1]\} \quad \text{where} \quad \tilde{\boldsymbol{q}}_i \triangleq \boldsymbol{q}_i / d^{\frac{1}{4}}$$

*and* $\boldsymbol{e}_j^{d+1}$ *is the* $j$-*th standard basis vector in* $\mathbb{R}^{d+1}$. *If* $\mathbf{K}_{\mathrm{att}} \triangleq \mathbf{k}_{\mathrm{att}}(\mathcal{X}, \mathcal{X})$ *for* $\mathcal{X} \triangleq \mathcal{X}' \cup \mathcal{X}_{\mathrm{in}}$, *then*

$$\max\big(\tfrac{1}{n}\|(\widehat{\mathbf{A}} - \mathbf{A}) \mathbf{V}\|_{\max}, \tfrac{1}{n}\|(\widehat{\mathbf{A}} - \mathbf{A})\mathbf{1}_n\|_\infty \|\mathbf{V}\|_{\max}\big) = \|\mathbf{K}_{\mathrm{att}}(\boldsymbol{p}_{\mathrm{in}} - \boldsymbol{p}_{\mathrm{out}})\|_{\mathcal{I}} \quad \text{for} \quad \mathcal{I} \triangleq [n(d+1)].$$

Our third lemma, proved in App. F.3, bounds the size of key parameters of the thinned attention problem.

**Lemma F.3** (**Thinned attention problem parameters**)**.** *Instantiate the notation of Lem. F.2, and define* $R \triangleq$

$\max_{i \in [n]} \max(\|\boldsymbol{q}_i\|_2, \|\boldsymbol{k}_i\|_2)$. *Then, for all $i, j \in \mathcal{I}$ and $l \in \operatorname{supp}(\boldsymbol{p}_{\text{in}})$,*

$$\|(\tfrac{1}{n}\mathbf{D})^{-1}\|_{\max} \leq \exp(\tfrac{R^2}{\sqrt{d}}), \quad \max_{\boldsymbol{x} \in \mathcal{X}_{\text{in}}} \sqrt{\mathbf{k}_{\text{att}}(\boldsymbol{x}, \boldsymbol{x})} \leq \exp(\tfrac{R^2}{2\sqrt{d}})\sqrt{\|\mathbf{V}\|_{2,\infty}^2 + \|\mathbf{V}\|_{\max}^2},$$

$$R_{\mathcal{I}} \triangleq \max_{i \in \mathcal{I}} \|\boldsymbol{x}_i\|_2 \leq \sqrt{\tfrac{R^2}{\sqrt{d}} + 1}, \quad D_{\mathcal{I}} \triangleq \max_{i \in \mathcal{I}} \sqrt{\mathbf{K}_{\text{att},ii}} \leq \exp(\tfrac{R^2}{2\sqrt{d}}),$$

$$\operatorname{rank}(\mathbf{X}_{\mathcal{I}}) \leq d + 1 \quad \text{for} \quad \mathbf{X}_{\mathcal{I}} \triangleq [\boldsymbol{x}_i]_{i \in \mathcal{I}}^{\top}, \quad \text{and}$$

$$|\mathbf{K}_{\text{att},il} - \mathbf{K}_{\text{att},jl}| \leq L_{\mathbf{K}_{\text{att}}}\|\boldsymbol{x}_i - \boldsymbol{x}_j\|_2 \quad \text{for} \quad L_{\mathbf{K}_{\text{att}}} \triangleq \exp(\tfrac{R^2}{\sqrt{d}})\sqrt{\tfrac{R^2}{\sqrt{d}} + 2}\|\mathbf{V}\|_{\max}.$$

Now instantiate the notation of Lem. F.2, and define the coefficient

$$c \triangleq 2\sqrt{2}\Big(32\sqrt{\tfrac{2}{3}(d+1)\log(3e^2(\tfrac{R^2}{\sqrt{d}} + 2)\|\mathbf{V}\|_{\max})} + \sqrt{2\log(8)}(1 + \tfrac{32}{\sqrt{3}})\Big).$$

Together, Lem. F.3, the KMS quality bound of Thm. 1, and the KH-COMPRESS(0.5) sub-Gaussian constant $\nu$ of Prop. B.5 imply that, with probability at least $\frac{1}{2}$,

$$\|\mathbf{K}_{\text{att}}(\boldsymbol{p}_{\text{in}} - \boldsymbol{p}_{\text{out}})\|_{\mathcal{I}} \leq \tfrac{c}{2\sqrt{2}}\exp(\tfrac{R^2}{\sqrt{d}})\sqrt{\|\mathbf{V}\|_{2,\infty}^2 + \|\mathbf{V}\|_{\max}^2}\frac{\sqrt{\log_2(n_{\text{out}})\log(8n_{\text{out}}\log_2\frac{n_{\text{in}}}{n_{\text{out}}})}}{n_{\text{out}}}.$$

Hence, by Lems. F.1 and F.2, with probability at least $\frac{1}{2}$,

$$\|\widehat{\mathbf{D}}^{-1}\widehat{\mathbf{A}}\mathbf{V} - \mathbf{D}^{-1}\mathbf{A}\mathbf{V}\|_{\max} \leq \tfrac{c}{\sqrt{2}}\exp(\tfrac{2R^2}{\sqrt{d}})\sqrt{\|\mathbf{V}\|_{2,\infty}^2 + \|\mathbf{V}\|_{\max}^2}\frac{\sqrt{\log_2(n_{\text{out}})\log(8n_{\text{out}}\log_2\frac{n_{\text{in}}}{n_{\text{out}}})}}{n_{\text{out}}}$$

$$\leq c\exp(\tfrac{2R^2}{\sqrt{d}})\|\mathbf{V}\|_{2,\infty}\frac{\sqrt{\log_2(n_{\text{out}})\log(8n_{\text{out}}\log_2\frac{n_{\text{in}}}{n_{\text{out}}})}}{n_{\text{out}}}.$$

### F.1. Proof of Lem. F.1: Decomposing attention approximation error

By the triangle inequality, we have

$$\|\widehat{\mathbf{D}}^{-1}\widehat{\mathbf{A}}\mathbf{V} - \mathbf{D}^{-1}\mathbf{A}\mathbf{V}\|_{\max} \leq \|\widehat{\mathbf{D}}^{-1}\widehat{\mathbf{A}}\mathbf{V} - \widehat{\mathbf{D}}^{-1}\mathbf{A}\mathbf{V}\|_{\max} + \|\widehat{\mathbf{D}}^{-1}\mathbf{A}\mathbf{V} - \mathbf{D}^{-1}\mathbf{A}\mathbf{V}\|_{\max}.$$

We bound the first term on the right-hand side using the submultiplicativity of the max norm under diagonal rescaling:

$$\|\widehat{\mathbf{D}}^{-1}\widehat{\mathbf{A}}\mathbf{V} - \widehat{\mathbf{D}}^{-1}\mathbf{A}\mathbf{V}\|_{\max} \leq \|\widehat{\mathbf{D}}^{-1}\|_{\max}\|\widehat{\mathbf{A}}\mathbf{V} - \mathbf{A}\mathbf{V}\|_{\max} = \|(\tfrac{1}{n}\widehat{\mathbf{D}})^{-1}\|_{\max}\tfrac{1}{n}\|\widehat{\mathbf{A}}\mathbf{V} - \mathbf{A}\mathbf{V}\|_{\max}.$$

To bound the second term we use the same submultiplicativity property and the fact that each entry of $\mathbf{D}^{-1}\mathbf{A}\mathbf{V}$ is the average of values in $\mathbf{V}$:

$$\|\widehat{\mathbf{D}}^{-1}\mathbf{A}\mathbf{V} - \mathbf{D}^{-1}\mathbf{A}\mathbf{V}\|_{\max} = \|\widehat{\mathbf{D}}^{-1}(\mathbf{D} - \widehat{\mathbf{D}})\mathbf{D}^{-1}\mathbf{A}\mathbf{V}\|_{\max} \leq \|\widehat{\mathbf{D}}^{-1}\|_{\max}\|\mathbf{D} - \widehat{\mathbf{D}}\|_{\max}\|\mathbf{D}^{-1}\mathbf{A}\mathbf{V}\|_{\max}$$

$$\leq \|(\tfrac{1}{n}\widehat{\mathbf{D}})^{-1}\|_{\max}\tfrac{1}{n}\|\mathbf{A}\mathbf{1}_n - \widehat{\mathbf{A}}\mathbf{1}_n\|_{\infty}\|\mathbf{V}\|_{\max}.$$

An identical argument reversing the roles of $(\mathbf{D}, \mathbf{A})$ and $(\widehat{\mathbf{D}}, \widehat{\mathbf{A}})$ yields the second bound.

### F.2. Proof of Lem. F.2: KMS bound on attention approximation error

Define the augmented value matrix $\widetilde{\mathbf{V}} = [\mathbf{V}, \|\mathbf{V}\|_{\max}\mathbf{1}_n] \in \mathbb{R}^{d+1}$. By the definition of $\mathbf{K}_{\text{att}}$ and $\widehat{\mathbf{A}}$,

$$\|\mathbf{K}_{\text{att}}(\boldsymbol{p}_{\text{in}} - \boldsymbol{p}_{\text{out}})\|_{\mathcal{I}} = \max_{i \in [n], j \in [d+1]}|\sum_{\ell \in [n]}\mathbf{A}_{i\ell}\widetilde{\mathbf{V}}_{\ell j}(\boldsymbol{p}_{\text{in}} - \boldsymbol{p}_{\text{out}})_{\ell}| = \tfrac{1}{n}\|(\mathbf{A} - \widehat{\mathbf{A}})\widetilde{\mathbf{V}}\boldsymbol{e}_j^d\|_{\infty} = \tfrac{1}{n}\|(\mathbf{A} - \widehat{\mathbf{A}})\widetilde{\mathbf{V}}\|_{\max}.$$

### F.3. Proof of Lem. F.3: Thinned attention problem parameters

First, by the Cauchy-Schwarz inequality and the nonnegativity of $\mathbf{D} = \mathbf{A}\mathbf{1}_n$ we have

$$\|(\tfrac{1}{n}\mathbf{D})^{-1}\|_{\max} = \frac{1}{\min_{i \in [n]}\frac{1}{n}\sum_{j \in [n]}\mathbf{A}_{ij}} \leq \frac{1}{\min_{i \in [n], j \in [n]}\exp(\frac{\langle \boldsymbol{q}_i, \boldsymbol{k}_j \rangle}{\sqrt{d}})} \leq \frac{1}{\min_{i \in [n], j \in [n]}\exp(\frac{-\|\boldsymbol{q}_i\|_2\|\boldsymbol{k}_j\|_2}{\sqrt{d}})} \leq \exp(\tfrac{R^2}{\sqrt{d}}).$$

Second, the $\max_{\boldsymbol{x}\in\mathcal{X}_{\mathrm{in}}}\sqrt{\mathbf{k}_{\mathrm{att}}(\boldsymbol{x},\boldsymbol{x})}$ inequality follows as

$$\mathbf{k}_{\mathrm{att}}((\tilde{\boldsymbol{k}}_i,\tilde{\boldsymbol{v}}_i),(\tilde{\boldsymbol{k}}_i,\tilde{\boldsymbol{v}}_i)) = \exp(\tfrac{\|\boldsymbol{k}_i\|_2^2}{\sqrt{d}})(\|\boldsymbol{v}_i\|_2^2 + \|\mathbf{V}\|_{\max}^2) \le \exp(\tfrac{R^2}{\sqrt{d}})(\|\mathbf{V}\|_{2,\infty}^2 + \|\mathbf{V}\|_{\max}^2).$$

Third, the $R_{\mathcal{I}}$ inequality follows as

$$\|(\tilde{\boldsymbol{q}}_i,\boldsymbol{e}_j^{d+1})\|_2 = \sqrt{\|\tilde{\boldsymbol{q}}_i\|_2^2 + 1} \le \sqrt{\tfrac{R^2}{\sqrt{d}} + 1} \quad \text{for all} \quad i\in[n], j\in[d+1].$$

Fourth, the $D_{\mathcal{I}}$ inequality follows as

$$\max_{i\in\mathcal{I}}\mathbf{K}_{\mathrm{att},ii} = \max_{i\in[n]}\exp(\tfrac{\|\boldsymbol{q}_i\|_2^2}{\sqrt{d}}) \le \exp(\tfrac{R^2}{\sqrt{d}}).$$

Fifth, the rank inequality follows as $\boldsymbol{x}_i\in\mathbb{R}^{d+1}$ for $i\in\mathcal{I}$. Finally, the Lipschitz inequality follows as, for any $i,k,l\in[n]$ and $j,m\in[d+1]$,

$$\begin{aligned}
&|\exp(\tfrac{\langle\boldsymbol{q}_i,\boldsymbol{k}_l\rangle}{\sqrt{d}})\langle\boldsymbol{e}_j^{d+1},\tilde{\boldsymbol{v}}_l\rangle - \exp(\tfrac{\langle\boldsymbol{q}_k,\boldsymbol{k}_l\rangle}{\sqrt{d}})\langle\boldsymbol{e}_m^{d+1},\tilde{\boldsymbol{v}}_l\rangle| \\
&\le \exp(\tfrac{\langle\boldsymbol{q}_i,\boldsymbol{k}_l\rangle}{\sqrt{d}})|\tilde{\boldsymbol{v}}_{lj} - \tilde{\boldsymbol{v}}_{lm}| + |\exp(\tfrac{\langle\boldsymbol{q}_i,\boldsymbol{k}_l\rangle}{\sqrt{d}}) - \exp(\tfrac{\langle\boldsymbol{q}_k,\boldsymbol{k}_l\rangle}{\sqrt{d}})||\tilde{\boldsymbol{v}}_{lm}| \\
&\le \exp(\tfrac{\|\boldsymbol{q}_i\|_2\|\boldsymbol{k}_l\|_2}{\sqrt{d}})\|\boldsymbol{e}_j^{d+1} - \boldsymbol{e}_m^{d+1}\|_2\tfrac{|\tilde{\boldsymbol{v}}_{lj}-\tilde{\boldsymbol{v}}_{lm}|}{\sqrt{2}} + \exp(\tfrac{\max(\|\boldsymbol{q}_i\|_2,\|\boldsymbol{q}_k\|_2)\|\boldsymbol{k}_l\|_2}{\sqrt{d}})|\tfrac{\langle\boldsymbol{q}_i-\boldsymbol{q}_k,\boldsymbol{k}_l\rangle}{\sqrt{d}}||\tilde{\boldsymbol{v}}_{lm}| \\
&\le \exp(\tfrac{R^2}{\sqrt{d}})\|\boldsymbol{e}_j^{d+1} - \boldsymbol{e}_m^{d+1}\|_2\tfrac{|\tilde{\boldsymbol{v}}_{lj}-\tilde{\boldsymbol{v}}_{lm}|}{\sqrt{2}} + \exp(\tfrac{R^2}{\sqrt{d}})\tfrac{\|\boldsymbol{q}_i-\boldsymbol{q}_k\|_2 R}{\sqrt{d}}|\tilde{\boldsymbol{v}}_{lm}| \\
&\le \exp(\tfrac{R^2}{\sqrt{d}})\|\boldsymbol{e}_j^{d+1} - \boldsymbol{e}_m^{d+1}\|_2\sqrt{2}\|\mathbf{V}\|_{\max} + \exp(\tfrac{R^2}{\sqrt{d}})\tfrac{\|\boldsymbol{q}_i-\boldsymbol{q}_k\|_2 R}{\sqrt{d}}\|\mathbf{V}\|_{\max} \\
&\le \exp(\tfrac{R^2}{\sqrt{d}})\sqrt{\tfrac{R^2}{\sqrt{d}}+2}\|\mathbf{V}\|_{\max}\|(\tilde{\boldsymbol{q}}_i,e_j^{d+1}) - (\tilde{\boldsymbol{q}}_k,\boldsymbol{e}_m^{d+1})\|_2
\end{aligned}$$

by the triangle inequality, multiple applications of Cauchy-Schwarz, and the mean-value theorem applied to $x\mapsto e^x$.

## G. Proof of Thm. 3: LKH-SGD convergence

Our proof makes use of three intermediate results. The first, inspired by Harvey & Samadi (2014, Thm. 10) and Cooper et al. (2023, Lem. 1), relates the quality of the ordering produced by Alg. 2 to the quality of the thinning.

**Lemma G.1 (Quality of thinned reordering).** *The output of thinned reordering (Alg. 2) satisfies*

$$\max_{j\in[n]}\|\textstyle\sum_{i=1}^j \boldsymbol{x}_{\pi_{k+1}(\pi_k^{-1}(i))}^k\|_2 \le \tfrac{1}{2}\max_{j\in[n]}\|\textstyle\sum_{i=1}^j \boldsymbol{x}_i^k\|_2 + \tfrac{1}{2}\max_{j\in[n]}\|\textstyle\sum_{i=1}^j \eta_i^k\boldsymbol{x}_i^k\|_2 + \|\textstyle\sum_{i=1}^n \boldsymbol{x}_i^k\|_2$$

*where $\pi_k^{-1}$ is the inverse permutation of $\pi_k$ and $\eta_i^k \triangleq 2(\mathbf{1}[\boldsymbol{x}_i^k\in\mathcal{X}_{\mathrm{out}}^k]-1)$.*

*Proof.* Fix any $j^\star\in\mathrm{argmax}_{j\in[n]}\|\sum_{i=1}^j \boldsymbol{x}_{\pi_{k+1}(\pi_k^{-1}(i))}^k\|_2$. If $j^\star\le n/2$, then

$$2\|\textstyle\sum_{i=1}^{j^\star} \boldsymbol{x}_{\pi_{k+1}(\pi_k^{-1}(i))}^k\|_2 \le 2\max_{j\in[n]}\|\textstyle\sum_{i=1}^j \mathbf{1}[\eta_i^k=1]\boldsymbol{x}_i^k\|_2 \le \max_{j\in[n]}\|\textstyle\sum_{i=1}^j \boldsymbol{x}_i^k\|_2 + \max_{j\in[n]}\|\textstyle\sum_{i=1}^j \eta_i^k\boldsymbol{x}_i^k\|_2$$

by the triangle inequality. Similarly, if $j^\star > n/2$, then,

$$\begin{aligned}
2(\|\textstyle\sum_{i=1}^{j^\star} \boldsymbol{x}_{\pi_{k+1}(\pi_k^{-1}(i))}^k\|_2 - \|\textstyle\sum_{i=1}^n \boldsymbol{x}_i^k\|_2) &\le 2\|\textstyle\sum_{i>j^\star} \boldsymbol{x}_{\pi_{k+1}(\pi_k^{-1}(i))}^k\|_2 \le 2\max_{j\in[n]}\|\textstyle\sum_{i=1}^j \mathbf{1}[\eta_i^k=-1]\boldsymbol{x}_i^k\|_2 \\
&\le \max_{j\in[n]}\|\textstyle\sum_{i=1}^j \boldsymbol{x}_i^k\|_2 + \max_{j\in[n]}\|-\textstyle\sum_{i=1}^j \eta_i^k\boldsymbol{x}_i^k\|_2.
\end{aligned}$$

$\square$

The second, a mild adaptation of Cooper et al. (2023, Thms. 2 and 3), bounds the convergence rate of SGD with thinned reordering in terms of the thinning quality.

**Theorem G.1 (Convergence of SGD with thinned reordering).** *Suppose that, for all $i \in [n]$ and $\boldsymbol{w}, \boldsymbol{v} \in \mathbb{R}^d$,*

$$\|\nabla f_i(\boldsymbol{w}) - \nabla f(\boldsymbol{w})\|_2^2 \leq \sigma^2 \quad and \quad \|\nabla f_i(\boldsymbol{w}) - \nabla f_i(\boldsymbol{v})\|_2 \leq L\|\boldsymbol{w} - \boldsymbol{v}\|_2$$

*and that SGD* (10) *with thinned reordering (Alg. 2) satisfies the* prefix discrepancy bound

$$\max_{j \in [n]} \|\textstyle\sum_{i=1}^j \eta_i^k \boldsymbol{x}_i^k\|_2 \leq 2\tilde{A} \max_{i \in [n]} \|\boldsymbol{x}_i^k - \bar{\boldsymbol{x}}^k\|_2 \quad for \quad \eta_i^k \triangleq 2(\mathbf{1}[\boldsymbol{x}_i^k \in \mathcal{X}_{\text{out}}^k] - 1), \quad \bar{\boldsymbol{x}}^k \triangleq \tfrac{1}{n}\textstyle\sum_{i=1}^n \boldsymbol{x}_i^k, \quad (39)$$

*and each epoch $k \in [K]$. Then the step size setting*

$$\alpha = \min\left\{ \frac{1}{16L(2n+\tilde{A})}, \left( \frac{4F_1}{42L^2\sigma^2\tilde{A}^2 nK + 18L^2 n^3 \sigma^2} \right)^{1/3} \right\} \quad with \quad F_1 \triangleq f(\boldsymbol{w}_1) - f^\star \quad and \quad f^\star \triangleq \inf_{\boldsymbol{v} \in \mathbb{R}^d} f(\boldsymbol{v})$$

*yields the convergence bound*

$$\tfrac{1}{K}\textstyle\sum_{k=1}^K \|\nabla f(\boldsymbol{w}_k)\|^2 \leq \frac{9(F_1 L \sigma \tilde{A})^{2/3}}{(nK)^{2/3}} + \frac{(72F_1 L \sigma)^{2/3} + 64F_1 L(2 + \tilde{A}/(n))}{K}.$$

*If, in addition, $f$ satisfies the $\mu$-Polyak-Łojasiewicz (PL) condition,*

$$\mu(f(\boldsymbol{w}) - f^\star) \leq \tfrac{1}{2}\|\nabla f(\boldsymbol{w})\|_2^2 \quad for\ all \quad \boldsymbol{w} \in \mathbb{R}^d,$$

*and the number of epochs satisfies*

$$K \geq 10 + \tfrac{1}{\mu}32L(2 + \tilde{A}/n)\tilde{W} \quad for \quad \tilde{W} \triangleq W_0(K^2 n^2 C_3) \quad and \quad C_3 \triangleq \frac{(F_1 + \sigma^2/L)\mu^2}{224L^2\sigma^2\tilde{A}^2},$$

*where $W_0$ denotes the Lambert W function, then the step size setting $\alpha = \frac{2\tilde{W}}{Kn\mu}$ yields the convergence bound*

$$f(\boldsymbol{w}_K) - f^\star \leq \frac{1}{(nK)^2}\left( \frac{(F_1 + L^2\sigma^2)\tilde{W}}{C_3} + \frac{112L^2\sigma^2\tilde{A}^2\tilde{W}^2}{\mu^3} \right).$$

*Proof.* The proof is identical to that of Cooper et al. (2023, Thms. 2 and 3) with $m = 1$ worker once each instance of $\|\cdot\|_\infty$ is replaced with $\|\cdot\|_2$, each instance of $L_{2,\infty}$ is replaced with $L$, each instance of $T$ is replaced with $K$, and Lem. G.1 is substituted for Cooper et al. (2023, Lem. 1). $\qquad\square$

The final result uses Thm. 1 to bound the prefix discrepancy of $\text{LKH}(\delta)$.

**Lemma G.2 ($\text{LKH}(\delta)$ prefix discrepancy).** *Fix any epoch $k \in [K]$. With probability at least $1 - \frac{\delta}{2} - \delta'$, thinned reordering (Alg. 2) with $\text{LKH}(\delta)$ satisfies the* prefix discrepancy bound (39) *with*

$$\tilde{A} = \sqrt{\log(\tfrac{2n(\log(n/2)+1)}{\delta})\big[e^2 \operatorname{rank}_{\epsilon_k}(\mathbf{X}^k) + 10e \log(\tfrac{n}{\delta'})\big]}$$

*for $\mathbf{X}^k \triangleq [\boldsymbol{x}_1^k, \ldots, \boldsymbol{x}_n^k]^\top$, $\epsilon_k \triangleq \max_{i \in [n]} \frac{\sqrt{9e \log(2n \log(en/2)/\delta) \log(n/\delta')}\|\boldsymbol{x}_i^k - \bar{\boldsymbol{x}}^k\|_2}{\sqrt{n}}$, and $\bar{\boldsymbol{x}}^k \triangleq \tfrac{1}{n}\sum_{i=1}^n \boldsymbol{x}_i^k$.*

*Proof.* Define $\mathcal{X}^k = \{\boldsymbol{x}_1^k, \ldots, \boldsymbol{x}_n^k\}$, $c = 2\max_{i \in [n]} \|\boldsymbol{x}_i^k - \bar{\boldsymbol{x}}^k\|_2$, and $r = \operatorname{rank}_{\epsilon_k}(\mathbf{X}^k)$. For any $j \in [n]$, we can write

$$\|\textstyle\sum_{i=1}^j \eta_i^k \boldsymbol{x}_i^k\|_2 = \|\textstyle\sum_{i=1}^j \boldsymbol{x}_i^k - \sum_{i=1}^j \mathbf{1}[\boldsymbol{x}_i^k \in \mathcal{X}_{\text{out},j}^k]\boldsymbol{x}_i^k\|_2 = 2j\|(\mathbf{X}^k)^\top(\boldsymbol{p}_{\text{in}}^j - \boldsymbol{p}_{\text{out}}^j)\|_2 = 2j \operatorname{MMD}_{\mathbf{X}^k(\mathbf{X}^k)^\top}(\boldsymbol{p}_{\text{in}}^j, \boldsymbol{p}_{\text{out}}^j)$$

where $\boldsymbol{p}_{\text{in}}^j$ and $\boldsymbol{p}_{\text{out}}^j$ are the empirical distributions over $\mathcal{X}_{\text{in},j}^k = (\boldsymbol{x}_i^k)_{i=1}^j$ and $\mathcal{X}_{\text{out},j}^k = \{\boldsymbol{x}_i^k \in \mathcal{X}_{\text{out}}^k : i \in [j]\}$.

Since $\text{LKH}(\delta)$ is an online algorithm that assigns signs $(\eta_i^k, \eta_{i+1}^k = 1 - \eta_i^k)$ to the points $(\boldsymbol{x}_i^k, \boldsymbol{x}_{i+1}^k)$ sequentially, we can view $\mathcal{X}_{\text{out},j}^k$ as the output of $\text{LKH}(\delta)$ applied to $\mathcal{X}_{\text{in},j}^k$ with $n_{\text{out}} = \frac{j}{2}$ and the linear kernel $\mathbf{k}(\boldsymbol{x}, \boldsymbol{y}) = \langle \boldsymbol{x}, \boldsymbol{y} \rangle$ for each $j \in [n]$. Therefore, we may invoke the established $\text{LKH}(\delta)$ sub-Gaussian constants $\nu_j$ of Prop. B.3, Thm. 1, the union bound, and the definition of $\epsilon$-rank (Def. 4) to deduce that

$$\max_{j \in [n]} \|\textstyle\sum_{i=1}^j \eta_i^k \boldsymbol{x}_i^k\|_2^2 \leq \max_{j \in [n]} 4j^2 \nu_j^2 \big[e^2 r + e\log(\tfrac{n}{\delta'})\big] + \sigma_{r+1}(\mathbf{X}^k)^2 \tfrac{4j^2}{j} \leq c^2\tilde{A}^2$$

with probability at least $1 - \frac{\delta}{2} - \delta'$. $\qquad\square$

Thm. 3 now follows directly from Thm. G.1 and Lem. G.2 applied to $\text{LKH}(\frac{1}{2K})$ with $\delta' = \frac{1}{4K}$ and a union bound over epochs.

## H. KT-COMPRESS($\delta$)

We describe the thinning algorithm KT-COMPRESS($\delta$) used in Alg. 3. We use KH($\delta$) for every halving round except for the last round, which thins a point sequence of size $2n_{\text{out}}$ to $n_{\text{out}}$. For this final halving round we use KH-REFINE($\delta$) (Alg. H.1) derived from the KT-SWAP algorithm of Dwivedi & Mackey (2024, Alg. 1b). The refinement stage of Alg. H.1 greedily improves the MMD of the initial KH($\delta$) output. Hence, $\text{MMD}_{\mathbf{k}}(\mathcal{X}_{\text{in}}, \mathcal{X}_{\text{out}}) \leq \text{MMD}_{\mathbf{k}}(\mathcal{X}_{\text{in}}, \mathcal{S}^{(1)})$ with probability 1.

---

**Algorithm H.1:** KH-REFINE($\delta$): KH($\delta$) with greedy refinement (Dwivedi & Mackey, 2024, Alg. 1a)

---

**Input:** point sequence $\mathcal{X}_{\text{in}} = (\boldsymbol{x}_i)_{i=1}^{n_{\text{in}}}$, kernel $\mathbf{k}$, input size $n_{\text{in}} \in 2\mathbb{N}$

$\mathcal{S} \leftarrow \text{KH}(\delta)(\mathcal{X}_{\text{in}}, \mathbf{k}); \quad n_{\text{out}} \triangleq n_{\text{in}}/2$

// Swap out each point in $\mathcal{X}_{\text{out}}$ for the best alternative in $\mathcal{X}_{\text{in}}$

$\mathcal{X}_{\text{out}} \leftarrow \mathcal{S}.\texttt{copy}()$

**for** $\boldsymbol{x} \in \mathcal{S}$ **do**

$\quad \mathcal{X}_{\text{out}} \leftarrow \mathcal{X}_{\text{out}} \setminus \{\boldsymbol{x}\} \cup \{\text{argmin}_{\boldsymbol{x}' \in \mathcal{X}_{\text{in}}} \text{MMD}_{\mathbf{k}}(\mathbb{P}_{\text{in}}, \mathbb{P}_{\text{out}} + \frac{1}{n_{\text{out}}}(\boldsymbol{\delta}_{\boldsymbol{x}'} - \boldsymbol{\delta}_{\boldsymbol{x}}))\}$

**end**

**return** $\mathcal{X}_{\text{out}}$, refined coreset of size $n_{\text{in}}/2$

---

## I. Proof of Thm. 4: Low-rank analysis of CTT power

Thm. 4 follows from the following more detailed statement, proved in App. I.1, as

$$\mathbf{R}_{\mathbf{K}}^2(n_{\text{in}}, \tfrac{\widetilde{\beta}}{20s_n}, \mathfrak{g}) + \mathbf{R}_{\mathbf{K}'}^2(n_{\text{in}}, \tfrac{\widetilde{\beta}}{20s_n}, \mathfrak{g}) = O(\widehat{\mathbf{R}}_{\mathbf{k}}^2).$$

**Theorem I.1** (**Low-rank analysis of CTT power, detailed**). *Under the assumptions of Thm. 4 with $n_{\text{in}} \triangleq \frac{m+n}{s}$, CTT (Alg. 3) rejects with probability at least $1 - \beta$ whenever $c' \text{MMD}_{\mathbf{k}}(\mathbb{P}, \mathbb{Q})/\sqrt{\log(1/\gamma)}$ exceeds*

$$2c_{\widetilde{\beta}/(20s)} \frac{\|\mathbf{k}\|_{\infty}^{\frac{1}{2}}}{\sqrt{m}} + \frac{\mathbf{R}_{\mathbf{k}}(\mathbb{P}, n_{\text{in}}, \frac{\widetilde{\beta}}{20s_m}, \mathfrak{g}) + \mathbf{R}_{\mathbf{k}}(\mathbb{Q}, n_{\text{in}}, \frac{\widetilde{\beta}}{20s_n}, \mathfrak{g})}{2^{\mathfrak{g}}\sqrt{m}}.$$

*Here, $c' > 0$ is a universal constant, $c_\delta \triangleq 2 + \sqrt{2\log(\frac{2}{\delta})}$, and $\mathbf{R}_{\mathbf{k}}(\mathbb{P}, n_{\text{in}}, \delta, \mathfrak{g})$ and $\mathbf{R}_{\mathbf{k}}(\mathbb{Q}, n_{\text{in}}, \delta, \mathfrak{g})$ respectively denote the $(1 - \frac{\delta}{2})$-th quantiles of $\mathbf{R}_{\mathbf{K}}(n_{\text{in}}, \delta, \mathfrak{g})$ and $\mathbf{R}_{\mathbf{K}'}(n_{\text{in}}, \delta, \mathfrak{g})$, where*

$$\mathbf{R}_{\widetilde{\mathbf{K}}}^2(n_{\text{in}}, \delta, \mathfrak{g}) \triangleq 256(\log_4 n_{\text{in}} - \mathfrak{g} - 1)(\sqrt{\log(n_{\text{in}} + 1)} + \sqrt{\log(2/\delta)})^2 \tag{40}$$
$$\cdot \left( \frac{2\sqrt{\|\widetilde{\mathbf{K}}\|_{\max}}}{\sqrt{3}} \left[ \sqrt{e \log(\frac{6 \cdot 2^{\mathfrak{g}} \sqrt{n_{\text{in}}}(\log_4 n_{\text{in}} - \mathfrak{g})}{\delta})} + \sqrt{\log(\frac{3n_{\text{in}}(\log_4 n_{\text{in}} - \mathfrak{g} - 1)}{\delta})} \right] \right.$$
$$\left. + \min_{r \leq 2^{\mathfrak{g}+1}\sqrt{n_{\text{in}}}} \left\{ \frac{2\sqrt{\|\widetilde{\mathbf{K}}\|_{\max}}}{\sqrt{3}} \sqrt{e^2 r \log\left(\frac{6 \cdot 2^{\mathfrak{g}} \sqrt{n_{\text{in}}}(\log_4 n_{\text{in}} - \mathfrak{g})}{\delta}\right)} + \sqrt{\lambda_{r+1}(\widetilde{\mathbf{K}}) \cdot 2^{\mathfrak{g}-1}\sqrt{n_{\text{in}}}} \right\}^2 \right)^2.$$

### I.1. Proof of Thm. I.1: Low-rank analysis of CTT power, detailed

Recall the following definition from Shetty et al. (2022, Def. 3).

**Definition I.1** (**k-sub-Gaussian thinning algorithm**). *We say a thinning algorithm* ALG *(satisfying Def. 1) is* **k-sub-Gaussian** *on an event $\mathcal{E}$ with shift $a$ and parameter $v$ if*

$$\mathbb{P}_{\mathcal{E}}(\text{MMD}_{\mathbf{k}}(\mathbb{P}_{\text{in}}, \mathbb{P}_{\text{out}}) \geq a + v\sqrt{t} \mid \mathcal{X}_{\text{in}}) \leq e^{-t} \quad \text{for all} \quad t \geq 0.$$

Fix $\widetilde{\mathbf{K}} \in \{\mathbf{K}, \mathbf{K}'\}$. To conclude our power result, it suffices, by Domingo-Enrich et al. (2023, Rmk. 2, App. B.1) and the failure probability setting of Domingo-Enrich et al. (2023, Lem. 11), to establish that

$$\mathbf{R}_{\widetilde{\mathbf{K}}}^2(n_{\text{in}}, \delta, \mathfrak{g}) = 256(\log_4 n_{\text{in}} - \mathfrak{g} - 1)(C_{\widetilde{\mathbf{K}}}(\delta, 2^{\mathfrak{g}+1}\sqrt{n_{\text{in}}}) + \mathfrak{M}_{\widetilde{\mathbf{K}}}(\delta, 2^{\mathfrak{g}+1}\sqrt{n_{\text{in}}})\sqrt{\log(\frac{3n_{\text{in}}(\log_4 n_{\text{in}} - \mathfrak{g} - 1)}{\delta})})^2$$
$$\cdot (\sqrt{\log(n_{\text{in}} + 1)} + \sqrt{\log(2/\delta)})^2, \tag{41}$$

for any scalars $C_{\widetilde{\mathbf{K}}}(\delta, 2^{\mathfrak{g}+1}\sqrt{n}_{\text{in}})$ and $\mathfrak{M}_{\widetilde{\mathbf{K}}}(\delta, 2^{\mathfrak{g}+1}\sqrt{n}_{\text{in}})$ satisfying the property that, on an event of probability at least $1-\delta/2$, every call to HALVE $\triangleq$ KH$(\frac{\ell^2}{n_{\text{in}}4^{\mathfrak{g}+1}(\log_4 n_{\text{in}}-\mathfrak{g})}\delta)$ with input size $\ell$ and output size $\ell/2$ is k-sub-Gaussian (Def. I.1) with shift $a_{\ell,n_{\text{in}},\widetilde{\mathbf{K}}}$ and parameter $v_{\ell,n_{\text{in}},\widetilde{\mathbf{K}}}$ satisfying

$$a_{\ell,n_{\text{in}},\widetilde{\mathbf{K}}} = \frac{C_{\widetilde{\mathbf{K}}}(\delta,\ell)}{\ell/2} \quad \text{and} \quad v_{\ell,n_{\text{in}},\widetilde{\mathbf{K}}} = \frac{\mathfrak{M}_{\widetilde{\mathbf{K}}}(\delta,\ell)}{\ell/2}\sqrt{\log(\frac{12n_{\text{in}}4^{\mathfrak{g}}(\log_4 n_{\text{in}}-\mathfrak{g})}{\ell\delta})}.$$

Substituting $\mathfrak{M}_{\widetilde{\mathbf{K}}}(\delta, 2^{\mathfrak{g}+1}\sqrt{n}_{\text{in}}) = (2^{\mathfrak{g}}\sqrt{n}_{\text{in}})v_{2^{\mathfrak{g}+1}\sqrt{n}_{\text{in}},n_{\text{in}},\widetilde{\mathbf{K}}}\left[\log(\frac{12n_{\text{in}}4^{\mathfrak{g}}(\log_4 n_{\text{in}}-\mathfrak{g})}{2^{\mathfrak{g}+1}\sqrt{n}_{\text{in}}\delta})\right]^{-\frac{1}{2}}$ and $C_{\widetilde{\mathbf{K}}}(\delta, 2^{\mathfrak{g}+1}\sqrt{n}_{\text{in}}) = (2^{\mathfrak{g}}\sqrt{n}_{\text{in}})a_{2^{\mathfrak{g}+1}\sqrt{n}_{\text{in}},n_{\text{in}},\widetilde{\mathbf{K}}}$ into (41), we obtain the sufficient condition

$$\mathbf{R}_{\widetilde{\mathbf{K}}}^2(n_{\text{in}}, \delta, \mathfrak{g}) = 256(\log_4 n_{\text{in}} - \mathfrak{g} - 1) \cdot (2^{\mathfrak{g}}\sqrt{n}_{\text{in}})^2 \cdot (\sqrt{\log(n_{\text{in}}+1)} + \sqrt{\log(2/\delta)})^2$$

$$\cdot \left(a_{2^{\mathfrak{g}+1}\sqrt{n}_{\text{in}},n_{\text{in}},\widetilde{\mathbf{K}}} + v_{2^{\mathfrak{g}+1}\sqrt{n}_{\text{in}},n_{\text{in}},\widetilde{\mathbf{K}}}\left[\log(\frac{12n_{\text{in}}4^{\mathfrak{g}}(\log_4 n_{\text{in}}-\mathfrak{g})}{2^{\mathfrak{g}+1}\sqrt{n}_{\text{in}}\delta})\right]^{-\frac{1}{2}}\sqrt{\log(\frac{3n_{\text{in}}(\log_4 n_{\text{in}}-\mathfrak{g}-1)}{\delta})}\right)^2. \tag{42}$$

We now identify suitable $a_{\ell,n_{\text{in}},\widetilde{\mathbf{K}}}$ and $v_{\ell,n_{\text{in}},\widetilde{\mathbf{K}}}$ with the aid of the following lemma, proved in App. I.2.

**Lemma I.1** (($\mathbf{K}, \nu, \delta$)-**sub-Gaussian thinning algorithms are** k-**sub-Gaussian**). *Suppose* ALG *is a* ($\mathbf{K}, \nu, \delta$)-*sub-Gaussian thinning algorithm, satisfying Def. 3 with an event $\mathcal{E}$ of probability at least $1-\delta/2$. Then* ALG *is* k-*sub-Gaussian (Def. I.1) on $\mathcal{E}$ with shift $a_{n_{\text{out}},n_{\text{in}},\mathbf{K}}$ and parameter $v_{n_{\text{out}},n_{\text{in}},\mathbf{K}}$ defined as*

$$a_{n_{\text{out}},n_{\text{in}},\mathbf{K}} \triangleq \nu\sqrt{e} + \min_{r \leq n_{\text{in}}}\left\{\nu\sqrt{e^2 r} + \sqrt{\lambda_{r+1}(\mathbf{K})(\frac{1}{n_{\text{out}}} - \frac{1}{n_{\text{in}}})}\right\} \quad \text{and} \quad v_{n_{\text{out}},n_{\text{in}},\mathbf{K}} \triangleq \nu\sqrt{e}.$$

By Prop. B.4 and Lem. A.1, KH$(\frac{\ell^2}{n_{\text{in}}4^{\mathfrak{g}+1}(\log_4 n_{\text{in}}-\mathfrak{g})}\delta)$ with input size $\ell$ and output size $\ell/2$ is a $(\widetilde{\mathbf{K}}, \nu, \frac{\ell^2}{n_{\text{in}}4^{\mathfrak{g}+1}(\log_4 n_{\text{in}}-\mathfrak{g})}\delta)$-sub-Gaussian thinning algorithm with

$$\nu \leq \frac{2}{(\ell/2)\sqrt{3}}\sqrt{\log\left(\frac{6(\ell/2)\log_2(\ell/(\ell/2))}{\delta} \cdot \frac{n_{\text{in}}4^{\mathfrak{g}+1}(\log_4 n_{\text{in}}-\mathfrak{g})}{\ell^2}\right)\|\widetilde{\mathbf{K}}\|_{\max}} = \frac{2}{(\ell/2)\sqrt{3}}\sqrt{\log\left(\frac{12n_{\text{in}}4^{\mathfrak{g}}(\log_4 n_{\text{in}}-\mathfrak{g})}{\ell\delta}\right)\|\widetilde{\mathbf{K}}\|_{\max}}.$$

By Lem. I.1, on an event of probability at least $1 - \frac{\ell^2}{2n_{\text{in}}4^{\mathfrak{g}+1}(\log_4 n_{\text{in}}-\mathfrak{g})}\delta$, KH$(\frac{\ell^2}{n_{\text{in}}4^{\mathfrak{g}+1}(\log_4 n_{\text{in}}-\mathfrak{g})}\delta)$ with input size $\ell$ and output size $\ell/2$ is a k-sub-Gaussian thinning algorithm with shift $a_{\ell,n_{\text{in}},\widetilde{\mathbf{K}}}$ and parameter $v_{\ell,n_{\text{in}},\widetilde{\mathbf{K}}}$ defined as

$$a_{\ell,n_{\text{in}},\widetilde{\mathbf{K}}} = \frac{2}{(\ell/2)\sqrt{3}}\sqrt{\log\left(\frac{12n_{\text{in}}4^{\mathfrak{g}}(\log_4 n_{\text{in}}-\mathfrak{g})}{\ell\delta}\right)\|\widetilde{\mathbf{K}}\|_{\max}}\sqrt{e\log 2}$$

$$+ \min_{r \leq \ell}\left\{\frac{2}{(\ell/2)\sqrt{3}}\sqrt{\log\left(\frac{12n_{\text{in}}4^{\mathfrak{g}}(\log_4 n_{\text{in}}-\mathfrak{g})}{\ell\delta}\right)\|\widetilde{\mathbf{K}}\|_{\max}}\sqrt{e^2 r} + \sqrt{\lambda_{r+1}(\widetilde{\mathbf{K}})(\frac{1}{\ell/2} - \frac{1}{\ell})}\right\} \quad \text{and} \tag{43}$$

$$v_{\ell,n_{\text{in}},\widetilde{\mathbf{K}}} = \frac{2}{(\ell/2)\sqrt{3}}\sqrt{\log\left(\frac{12n_{\text{in}}4^{\mathfrak{g}}(\log_4 n_{\text{in}}-\mathfrak{g})}{\ell\delta}\right)\|\widetilde{\mathbf{K}}\|_{\max}}\sqrt{e}. \tag{44}$$

Moreover, by the union bound, as detailed in Shetty et al. (2022, App. F.1), every HALVE call made by KT-COMPRESS is simultaneously k-sub-Gaussian with these input-size-dependent parameters on a common event of probability at least $1 - \frac{\delta}{2}$. Substituting (43) and (44) with $\ell = 2^{\mathfrak{g}+1}\sqrt{n}_{\text{in}}$ into (42), we obtain our error inflation factor expression (40), completing the proof.

### I.2. Proof of Lem. I.1: ($\mathbf{K}, \nu, \delta$)-sub-Gaussian thinning algorithms are k-sub-Gaussian

Fix any $t \geq 0$, and let $\delta' = e^{-t}$. By our sub-Gaussian assumption, Thm. 1 implies that, as advertised,

$$e^{-t} \geq \mathbb{P}_{\mathcal{E}}\left(\text{MMD}_{\mathbf{K}}^2(\boldsymbol{p}_{\text{in}}, \boldsymbol{p}_{\text{out}}) \geq \min_{r \leq n_{\text{in}}} \nu^2\left[e^2 r + et\right] + \lambda_{r+1}(\mathbf{K})(\frac{1}{n_{\text{out}}} - \frac{1}{n_{\text{in}}})\right)$$

$$= \mathbb{P}_{\mathcal{E}}\left(\text{MMD}_{\mathbf{K}}(\boldsymbol{p}_{\text{in}}, \boldsymbol{p}_{\text{out}}) \geq \min_{r \leq n_{\text{in}}} \sqrt{\nu^2[e^2 r + et] + \lambda_{r+1}(\mathbf{K})(\frac{1}{n_{\text{out}}} - \frac{1}{n_{\text{in}}})}\right)$$

$$\geq \mathbb{P}_{\mathcal{E}}\left(\text{MMD}_{\mathbf{K}}(\boldsymbol{p}_{\text{in}}, \boldsymbol{p}_{\text{out}}) \geq \nu\sqrt{e}\sqrt{t} + \min_{r \leq n_{\text{in}}} \nu\sqrt{e^2 r} + \sqrt{\lambda_{r+1}(\mathbf{K})(\frac{1}{n_{\text{out}}} - \frac{1}{n_{\text{in}}})}\right).$$

## J. Proof of Cor. 3: Power of deep kernel CTT

Define the radius

$$R' \triangleq \max_{\boldsymbol{y} \in \mathcal{Y} \cup \mathcal{X}} \|(\phi(\boldsymbol{y}), \boldsymbol{y})\|_2,$$

the augmented vectors $\mathcal{Y}' \triangleq \{(\phi(\boldsymbol{y}), \boldsymbol{y})\}_{\boldsymbol{y} \in \mathcal{Y}}$, and the augmented kernel

$$q'((\phi(\boldsymbol{x}), \boldsymbol{x}), (\phi(\boldsymbol{y}), \boldsymbol{y})) \triangleq \kappa(\phi(\boldsymbol{x}), \phi(\boldsymbol{y})) q(\boldsymbol{x}, \boldsymbol{y}) = \exp(-\eta \|(\phi(\boldsymbol{x}), \boldsymbol{x}) - (\phi(\boldsymbol{y}), \boldsymbol{y})\|_2^2).$$

Since the deep kernel (13) takes the form

$$\mathbf{k}_{\text{deep}}(\boldsymbol{x}, \boldsymbol{y}) = (1 - \epsilon) q'((\phi(\boldsymbol{x}), \boldsymbol{x}), (\phi(\boldsymbol{y}), \boldsymbol{y})) + \epsilon q(\boldsymbol{x}, \boldsymbol{y})$$

we also have

$$\mathbf{K}_{\text{deep}} \triangleq \mathbf{k}_{\text{deep}}(\mathcal{Y}, \mathcal{Y}) = (1 - \epsilon)\mathbf{Q}' + \epsilon\mathbf{Q} \quad \text{for} \quad \mathbf{Q}' \triangleq q'(\mathcal{Y}', \mathcal{Y}') \quad \text{and} \quad \mathbf{Q} \triangleq q(\mathcal{Y}, \mathcal{Y}).$$

Hence, by Weyl's inequality (Horn & Johnson, 1985, Thm. 4.3.1) and the Gaussian kernel matrix eigenvalue bound (6),

$$\lambda_{2r+1}(\mathbf{K}_{\text{deep}}) \le (1 - \epsilon)\lambda_{r+1}(\mathbf{Q}') + \epsilon\lambda_{r+1}(\mathbf{Q}) \le n e^{-\frac{d'}{2e} r^{1/d'} \log\left(\frac{d' r^{1/d'}}{4e^2 \eta R'^2}\right)} \quad \text{for} \quad (2e)^{d'} \le r < n.$$

Parallel reasoning and the assumption $m \le n$ yield the same bound for $\lambda_{2r+1}(\mathbf{k}_{\text{deep}}(\mathcal{X}, \mathcal{X}))$ and $(2e)^{d'} \le r < m$. Now consider the approximate rank parameter

$$r^\star \triangleq \max\left\{ \left[\frac{2e}{d'} \log(n n_{\text{out}} b)\right]^{d'}, (R'^2 \eta e^3 4/d')^{d'} \right\} \quad \text{for} \quad b \triangleq \frac{1}{2}. \tag{45}$$

Then, for $n \ge (2e)^{d'}$, we have, exactly as in App. D,

$$\lambda_{2r^\star+1}(\mathbf{K}_{\text{deep}}) + \lambda_{2r^\star+1}(\mathbf{k}_{\text{deep}}(\mathcal{X}, \mathcal{X})) \le \frac{2}{n_{\text{out}} b}$$

and therefore

$$\widehat{\mathbf{R}}_\mathbf{k} = O\left(\sqrt{\log(\tfrac{n}{s})} \log(\tfrac{n}{\widetilde{\beta}}) \max\left\{ \left[\frac{2e}{d'} \log(n n_{\text{out}} b)\right]^{d'/2}, (R'^2 \eta e^3 4/d')^{d'/2} \right\}\right).$$

Our final step is to bound the quantile of the sole remaining data-dependent term, $R'$. Since the inputs are $c$-sub-Gaussian (14), Lem. 1 of Dwivedi & Mackey (2024) with $\psi^{-1}(r) = \frac{\sqrt{\log r}}{\sqrt{c}}$ implies that the $1 - \frac{\widetilde{\beta}}{20 s_n}$ quantile of $R'$ is $O\left(\sqrt{\log(\tfrac{n}{\widetilde{\beta}})}\right)$, yielding the result.

## K. Proof of Cor. 4: Power of deep manifold kernel CTT

Our reasoning is identical to that in App. J with the manifold Gaussian kernel matrix eigenvalue bound (7) now substituted for the Euclidean ball bound (6) and the approximate rank setting $r^\star = (\log(n n_{\text{out}})/c)^{5d^\star/2}$ substituted for (45).

## L. Supplementary Experiment Details

### L.1. Approximating attention experiments

The experiment of Tab. 3 was carried out using Python 3.12.9, PyTorch 2.8.0.dev20250407+cu128 (Paszke et al., 2019), and an Ubuntu 22.04.5 LTS server with an AMD EPYC 7V13 64-Core Processor, 220 GB RAM, and a single NVIDIA A100 GPU (80 GB memory, CUDA 12.8, driver version 570.124.04). For reference, attention layer 1 has $(n, d) = (3136, 64)$ and attention layer 2 has $(n, d) = (784, 64)$. For each of the first 50 ImageNet 2012 validation set batches of size 64, we measured the time required to complete a forward pass through each the approximate softmax matrix (8) layer using CUDA events following 10 warm-up batches to initialize the GPU. Tab. L.1 provides the hyperparameter settings for each attention approximation in Tab. 3.

The experiment of Tab. 4 was carried out using Python 3.12.9, PyTorch 2.6.0, and an Ubuntu 22.04.5 LTS server with an Intel(R) Xeon(R) Gold 5218 CPU Processor, 100 GB RAM, and a single NVIDIA A6000 GPU (48 GB memory, CUDA 12.1, driver version 530.30.02). For reference, the BigGAN model contains a single attention layer with $4096$ queries in $\mathbb{R}^{64}$, 1024 keys in $\mathbb{R}^{64}$, and 1024 values in $\mathbb{R}^{256}$. We measured the time required to complete a forward pass through the approximate softmax matrix (8) layer using CUDA events following 10 warm-up batches to initialize the GPU. Tab. L.2 provides the hyperparameter settings for each attention approximation in Tab. 4.

The settings and implementations for all methods other than Thinformer were provided by Zandieh et al. (2023), and our experiment code builds on their open-source repository `https://github.com/majid-daliri/kdeformer`.

Table L.1: **Configurations for the attention approximation methods of Tab. 3.**

| Attention Algorithm | Layer 1 Configuration | Layer 2 Configuration |
|:---:|:---:|:---:|
| **Performer** | `num_features=49` | `num_features=12` |
| **Reformer** | `bucket_size=49`
`n_hashes=2` | `bucket_size=12`
`n_hashes=2` |
| **ScatterBrain** | `local_context=49`
`num_features=48` | `local_context=12`
`num_features=6` |
| **KDEformer** | `sample_size=64`
`bucket_size=32` | `sample_size=56`
`bucket_size=32` |
| **Thinformer (Ours)** | `g=2` | `g=4` |

Table L.2: **Configurations for the attention approximation methods of Tab. 4.**

| Attention Algorithm | Layer Configuration |
|:---:|:---:|
| **Performer** | `num_features=128` |
| **Reformer** | `bucket_size=64`
`n_hashes=8` |
| **ScatterBrain** | `local_context=32`
`num_features=128` |
| **KDEformer** | `sample_size=128`
`bucket_size=64` |
| **Thinformer (Ours)** | `g=2` |

## L.2. Faster SGD training experiments

The experiment of Sec. 5.2 was carried out using Python 3.10, PyTorch 2.0.1, a Rocky Linux 8.9 server with 64 CPU cores (Intel(R) Xeon(R) Platinum 8358 CPU @ 2.60GHz), and a NVIDIA A100 GPU (40 GB memory, CUDA 12.4, driver version 550.54.15).

Technically, the CD-GraB: SBW algorithm requires an a priori upper bound on the maximum Euclidean norm $b_{\max}$ of any stochastic gradient that it will encounter. To conduct our experiment, we first estimate $b_{\max}$ by calculating the maximum gradient Euclidean encountered across 10 epochs of running SGD with $\mathrm{LKH}(\frac{1}{2K})$ reordering. One would typically not choose to carry out such a two-step procedure in practice, but the experiment serves to demonstrate that the CD-GraB: SBW leads to overly conservative performance even if reasonable upper bound is known in advance.

The settings and implementation for both random reshuffling (RR) and CD-GraB: Greedy were those used in the original logistic regression on mortgage application experiment of Cooper et al. (2023). Our experiment code builds on the open-source CD-GraB repository `https://github.com/GarlGuo/CD-GraB`. As in Cooper et al. (2023), optimization was carried out with a learning rate of $\alpha = 0.01$, datapoints were loaded in batches of size 16, and stochastic gradients were reordered for each datapoint individually.

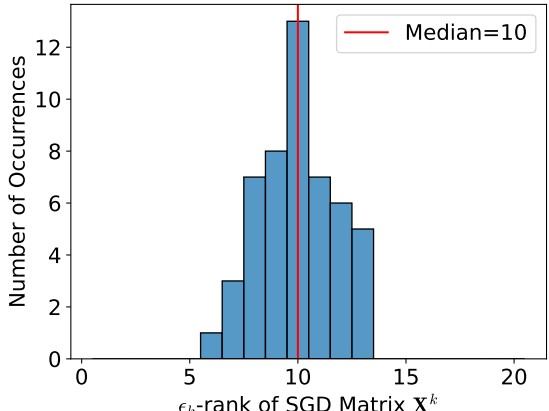 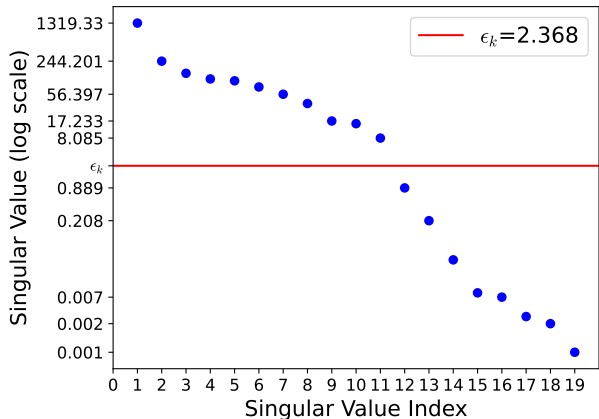

Figure L.1: **Approximate low-rank structure of stochastic gradient matrices.** *Left.* For each epoch $k$ of the $\mathrm{LKH}(\frac{1}{2K})$ experiment of Sec. 5.2, we record the $\epsilon_k$-rank of the stochastic gradient matrix $\mathbf{X}^k \triangleq [\boldsymbol{x}_1^k, \ldots, \boldsymbol{x}_n^k]^\top \in \mathbb{R}^{n \times d}$ (see Def. 4 and Thm. 3). Notably, the $\epsilon_k$-ranks are significantly smaller than the ambient dimension $d = 19$. *Right.* We display the singular values of the first stochastic gradient matrix, $\mathbf{X}^1$. The singular values drop off steeply, resulting in relatively small $\epsilon_k$-ranks.

## L.3. Cheap two-sample testing experiment

The experiment of Sec. 6.2 was carried out using Python 3.12.9, PyTorch 2.6.0+cu124, and an Ubuntu 20.04.6 LTS server with an AMD EPYC 9554 64-Core Processor, 100 GB RAM, and a single NVIDIA H100 GPU (96 GB memory, CUDA 12.2, driver version 535.154.05). Each test is run with replication count $\mathcal{B} = 100$, nominal level $\alpha = 0.05$, and failure probability $\delta = 0.5$. The neural network $\phi$ was trained exactly as in Liu et al. (2020) (with learning rate $5 \times 10^{-5}$ and batch size equal to the full training sample size), and runtime measurements exclude the time required to train $\phi$. Our experiment code builds on the open-source deep kernel testing (https://github.com/fengliu90/DK-for-TST) and Compress Then Test (https://github.com/microsoft/goodpoints) repositories.

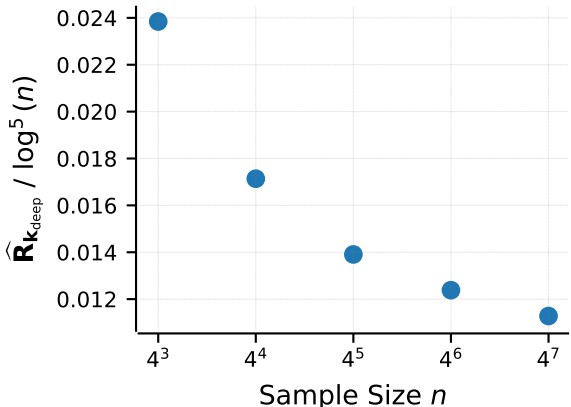

Figure L.2: **Slow-growing error inflation.** In the experimental setting of Sec. 6.2, the error inflation factor $\widehat{\mathbf{R}}_{\mathbf{k}_{\mathrm{deep}}}$ enjoys $O(\log^5(n))$ growth as the ratio $\widehat{\mathbf{R}}_{\mathbf{k}_{\mathrm{deep}}}/\log^5(n)$ decreases with $n$. Here, $\widehat{\mathbf{R}}_{\mathbf{k}_{\mathrm{deep}}}^2$ is defined by (12) with $n = m$, $n_{\mathrm{in}} \triangleq \frac{2n}{s}$ and $n_{\mathrm{out}} \triangleq 2^{\mathfrak{g}}\sqrt{n_{\mathrm{in}}}$ for constants $\beta = 0.05$, $s = 32$, and $\mathfrak{g} = 4$.

