# OpenReview forum: "Low-Rank Thinning"
_ICML.cc/2025/Conference — ICML 2025 poster_

### Official Review · Reviewer_XcrH · 2025-03-09

**Overall Recommendation:** 4

**Summary:**

This paper proposes Low-Rank Thinning, a method for selecting representative data points using sub-Gaussian thinning with improved efficiency. By leveraging low-rank structures, it enhances dataset summarization capability. Theoretical guarantees and empirical results demonstrate its ability to reduce computation while preserving data quality.


## update after rebuttal

My concerns are addressed by the authors, I improved my rating to 4.

**Claims And Evidence:**

yes

**Essential References Not Discussed:**

no

**Experimental Designs Or Analyses:**

The paper presents experiments evaluating Low-Rank Thinning on transformer attention, stochastic gradient training, and distribution testing. Theoretical guarantees are supported by empirical results. Although the experiments show gains in computational efficiency and accuracy, additional validation on other task apart from image classification would enhance the claims of generalizability and practical relevance, e.g., semantic segmentation, object detection.

**Methods And Evaluation Criteria:**

yes

**Other Comments Or Suggestions:**

Please refer to the weaknesses

**Other Strengths And Weaknesses:**

Strengths:

1. The proposed Low-Rank Thinning framework improves dataset summarization across diverse aspects, including transformer attention, stochastic gradient optimization, and two-sample testing, which can benefit the community.

2. The paper provides strong theoretical guarantees, extending sub-Gaussian thinning to more general distributions and kernels while addressing limitations in prior work.

3.Experimental results demonstrate that Low-Rank Thinning reduces computational costs while maintaining high accuracy, making it practical for large-scale machine learning tasks.


Weaknesses:

1. The experiments focus on constrained settings but lack extensive evaluation on other tasks, e.g., object detection, which may hinder the evaluation of the generalizability.

2, The method assumes a low-rank structure in the data, which may not always be present in real-world scenarios and can decrease its effectiveness for high-dimensional or complex datasets.

3. In Table 3, experiments are only conducted on T2T-ViT, more transformer architectures are suggested to be included while their variants should aslo be considered to test the cross backbone generalizability of the proposed method.

4. The training details regarding the hyperparamters, e.g., lr, batch size, are not well described in both of the main paper and appendix.

**Questions For Authors:**

My questions are all derived from the weaknesses I mentioned.

1. The experiments are conducted in constrained settings and do not include tasks like object detection. How would the proposed method perform on more diverse applications, and could additional benchmarks improve the evaluation of its generalizability?


2. The method relies on a low-rank structure in data, which may not always hold in practical settings. How does its performance degrade when applied to high-dimensional or non-low-rank data, and are there strategies to mitigate this limitation?

3. Table 3 only includes experiments on T2T-ViT, without testing on other transformer architectures or their variants. Would extending the evaluation to different transformer backbones provide stronger evidence of the method’s cross-architecture generalizability?

4. The paper lacks detailed descriptions of training hyperparameters such as learning rate and batch size in both the main text and appendix. Can the authors provide the details of the experiments in main text or appendix further?

**Relation To Broader Scientific Literature:**

This paper extends sub-Gaussian thinning to broader distributions and kernels, overcoming limitations in prior dataset summarization methods. By leveraging low-rank structures, it enhances attention approximation, stochastic optimization, and distribution testing with improved efficiency and theoretical guarantees.

**Theoretical Claims:**

yes I checked the proof of C part in the appendix and it looks ok

---

> ### Author Rebuttal · Authors · 2025-03-31
>
> We thank the reviewer for the positive and constructive feedback and are delighted that the reviewer found our theoretical guarantees strong, our methods practical, and our applications diverse and of benefit to the community.
>
> **A new application:** We presented three vignettes with three diverse applications (SGD training, two-sample testing, and attention approximation) to demonstrate the broad applicability and generalizability of the techniques, but we are happy to include an additional experiment with a different architecture and application as further evidence of generalizability. We have recreated the BigGAN image generation benchmarking experiment of Zandieh et al. where leading attention approximation methods are scored on computational expense and image generation quality. We used the experimental setup and released implementations of Zandieh et al. for all other methods. Our results table (https://imgur.com/a/F5Y7pfh) shows that Thinformer (g=2) yields better Frechet Inception Distance (FID) and Inception Scores (IS) than all of the alternatives while running significantly faster than exact, KDEformer, and Reformer. Performer runs faster still but at the expense of substantially worse FID and IS.
>
> **Low-rankness:** For attention approximation, even if the matrices Q and K are full rank, we still obtain the theoretical and practical benefits advertised (since their rank is d < n). The KMS guarantees (4) and (5) of Thm. 1 also ensure that we obtain a high-quality KMS approximation even if the kernel matrix is full-rank.
>
> For the other two applications, we have run two new analyses to test to what extent the relevant matrices are approximately low-rank in practice. In the CTT setting of Sec. 6.2, we find that the empirical inflation factor $R_k = O(\log^5(n))$ owing to approximate low-rankness: see https://imgur.com/a/hmAVQIU. In the LKH experiment setting of Sec. 5.2, we find that the stochastic gradient matrices $X_k$ have rapidly decaying singular values and a median $\epsilon_k^*$-rank of 10: see https://imgur.com/a/MyA7gR1.
>
> Thank you for the suggestion regarding hyperparameters! We initially omitted hyperparameters when recreating experiments fully specified in prior work, but we will include those hyperparameters in the revision. These include Sec. 6.2 (learning rate = 5e-5; batch size = full sample), Sec. 5.2 (learning rate = 1e-2; batch size = 16), and Sec. 4.2 (learning rate = N/A (inference only); batch size = 64).

---

### Official Review · Reviewer_UEYY · 2025-03-11

**Overall Recommendation:** 2

**Summary:**

This paper focuses on the thinning problem and proposes a low-rank method to simplify datasets with a limited number of data points, based on the analysis of sub-Gaussian thinning. The proposed low-rank analysis generalizes sub-Gaussian thinning to any distribution and kernel, ensuring high-quality compression of datasets.

## Update After Rebuttal

Thanks for the rebuttal. I will keep the initial score but with low confidence.

**Claims And Evidence:**

The claims in this paper are mainly supported by theoretical analysis.

**Essential References Not Discussed:**

The related work section not available in this paper.

**Experimental Designs Or Analyses:**

Yes

**Methods And Evaluation Criteria:**

Yes

**Other Comments Or Suggestions:**

N/A

**Other Strengths And Weaknesses:**

**Clarification: I don't have a background in thinning or kernel methods, so my assessment below comes with low confidence.**

## Strengths

1. The paper extends sub-Gaussian thinning beyond previous constraints, providing a comprehensive framework applicable to arbitrary kernels and distributions.
2. Based on the proposed method, the paper introduces Thinformer attention, a mechanism that provides a computationally efficient alternative to traditional self-attention while maintaining competitive accuracy. The Thinfomer layer is 2x faster than baselines like KDEformer while achieving a comparable accuracy.
3. In additional to the network architecture, the paper also involves the SGD acceleration, and shows significant convergence speed improvements over random reshuffling.

## Weaknesses

My main concern is the poor presentation of the paper, which makes it less accessible to non-expert readers. For instance, the thinning problem is not clearly defined at the outset. For example, the paper claims that *"This work is about thinning, finding a small set of
representative points to accurately summarize a larger dataset"*. At first glance, the paper appears to address the Core Set problem, which also aims to represent a dataset with a small subset. However, it is actually a general algorithm and is not related to DATASETS.

In addition, the main topic changes frequently in this paper. In the methodology section, the focus shifts to the Thinformer module for fast attention approximation. Then, in Section 5, the discussion pivots again—this time to fast SGD optimization. These transitions between seemingly different topics make the paper somewhat difficult to follow and may lead to confusion about its contribution.

**Questions For Authors:**

N/A

**Relation To Broader Scientific Literature:**

N/A

**Theoretical Claims:**

No.

---

> ### Author Rebuttal · Authors · 2025-03-31
>
> We thank the reviewer for the positive and constructive feedback and are glad that the reviewer found our framework “comprehensive” and our speed improvements “significant.”
>
> We apologize for any inaccessibility in our presentation. We aimed to showcase the broad applicability of theory developed in this paper by explicitly deriving sub-Gaussian thinning solutions with state-of-the-art guarantees for three practical problems. We will improve the coherence of the presentation and the accessibility for non-expert readers by providing a more detailed overarching roadmap in the introduction (with guidelines on what will be included in each section), improving transitions between the different application sections, summarizing our main proof arguments, and providing more high-level discussion of theoretical and practical takeaways.
>
> In addition, we will make the definition and broad scope of thinning clearer from the outset by highlighting that thinning flexibly allows one to summarize any of collection of points, be they datapoints (as in our testing application), stochastic gradients (as in our SGD application), or key-value pairs (as in our attention application).
>
> We will also clarify in the introduction that related work for the main results and applications are discussed separately in each section.

---

### Official Review · Reviewer_yDej · 2025-03-12

**Overall Recommendation:** 4

**Summary:**

This paper introduces a novel theoretical analysis of "thinning algorithms" that adapt effectively to low-rank structures present in data. The authors develop a theoretical upper bound on the discrepancy metrics (Theorem 1) applicable to any kernel and any data distribution, thereby demonstrating that thinning algorithms with sub-Gaussian guarantees (per Definition 3) achieve improved quality metrics especially when the kernel or data matrix is approximately low-rank. Three significant applications are also discussed: (1) approximating dot-product attention in Transformers efficiently, (2) expediting stochastic gradient descent (SGD) training by optimized data reordering, and (3) conducting two-sample kernel maximum mean discrepancy (MMD) tests efficiently via Compress-Then-Test approach. The authors provide theoretical guarantees for these applications (Theorems 2, 3, and 4, respectively) and extensive empirical experiments validating the efficacy of the proposed thinning-based approaches.

**Claims And Evidence:**

The paper makes four main claims: a general theorem providing high-probability upper bounds on the MMD and kernel max seminorm (Theorem 1), and three application-specific theorems demonstrating the utility of thinning-based approaches (Theorems 2, 3, and 4). Overall, these theoretical claims---particularly regarding improved complexity and performance bounds---are well-supported through rigorous proofs provided in the appendix and extensive empirical validation. However, the paper's presentation could be improved by succinctly summarizing the high-level ideas behind these proofs, especially for Theorem 1, in the main text. Currently, relegating all proofs to the appendix leaves readers without clear intuition on the core ideas underpinning the main results. Additionally, clearer explicit summaries and high-level discussions of theoretical takeaways, especially in Section 6.1 (e.g., how Algorithm 3's analysis compares with prior works), would significantly improve the readability and impact of the paper.

**Essential References Not Discussed:**

It seems essential references are properly cited and discussed.

**Experimental Designs Or Analyses:**

The experimental designs---e.g., especially the Transformer approximation experiments (Section 4.2) and SGD training (Section 5.2)---seem well-executed to support and complement the theoretical claims.

**Methods And Evaluation Criteria:**

The methods proposed and their associated evaluation criteria are appropriate and practical for the applications considered. For the Transformer approximation task, benchmark experiments on ImageNet provide relevant and convincing empirical evidence of efficiency and accuracy. Similarly, the logistic regression experiment for accelerating SGD and the Higgs boson detection task for two-sample testing are both well-chosen and pertinent.

**Other Comments Or Suggestions:**

* It would significantly benefit readability and clarity if the authors provided a concise, high-level summary of key proofs, particularly for Theorem 1, in the main body of the text.

* Presentation could be improved by ensuring algorithm references are not prematurely made before formal definitions, especially those relegated to appendices.

* Explicitly summarizing the key theoretical results in Section 6.1 and clearly situating Algorithm 3 relative to prior methods would enhance readability and interpretability of the contributions.

* $\nu$ is used to denote the sub-Gaussian constant in Definition 3, but it was also used to denote a distribution in Definition 2. Please try to avoid overloading notation.

* In the sentence surrounding Eq. (7), please specify that Eq. (7) applies to Gaussian kernel for clarity.

* It seems Table 2 is redundant as the three-line display at the bottom right of page 4 contains the same information.  Please consider removing one to use the saved space for other purposes.

**Other Strengths And Weaknesses:**

Strengths:
  - Originality in providing generally applicable theoretical guarantees for (sub-Gaussian) thinning algorithms that can effectively adapt to (approximate) low-rank structure.
  - Broad applicability showcased through multiple practical applications.
  - Solid and comprehensive theoretical guarantees, complemented by extensive empirical evidence.

Weaknesses:
  - Although the contributions are substantial and the manuscript is generally clear, the presentation could be improved (see suggestions in "Other Comments or Suggestions" below).

**Questions For Authors:**

1. Could you comment on the tightness of the upper bounds provided in Theorem 1? Do you have any insights or conjectures regarding their sharpness? Additionally, if feasible, could you empirically evaluate the tightness of your theoretical upper bounds through a simple experiment with synthetic datasets?

2. The numerical experiments in Sections 4–6 demonstrate the practical effectiveness of thinning-based approaches, yet the connection to the theoretical results (Theorems 2–4) seems unclear (or indirect). Could you clarify whether and how these experiments validate the tightness or relevance of your theoretical guarantees?  Could you also comment on the tightness of these theorems, and any room for further improvements, if any?

**Relation To Broader Scientific Literature:**

The authors clearly position their contributions within existing literature on kernel methods, coresets, and thinning algorithms, differentiating from recent related studies. The connections to practical applications in Transformers, SGD acceleration, and kernel two-sample tests are effectively highlighted.

**Theoretical Claims:**

I briefly checked Theorem 1 and its associated proofs in the appendix, which appear rigorous and sound. Nonetheless, I haven't checked the correctness of details in the proof.

---

> ### Author Rebuttal · Authors · 2025-03-31
>
> We thank the reviewer for the positive and constructive feedback and are delighted that the reviewer found our methods broadly applicable, our experiments extensive and convincing, and our contributions novel and clearly positioned within the literature.
>
> **Summaries:** Following the reviewer’s advice, we will add a summary of core ideas underpinning the main results. For example, for (3), we decompose the sub-Gaussian error vector $K^{1/2}w$ for $w = p_{in}-p_{out}$ into a rank-r projection $V_r^\top K^{1/2}w\in R^r$ and a residual. We bound the residual error in terms of $\lambda_{r+1}$ and $\vert\vert w\vert\vert_2=1/n_{out}-1/n_{in}$ and bound $\vert\vert V_r^\top K^{1/2}w\vert\vert_2=\sup_{\vert\vert u\vert\vert_2\le1} u^\top V_r^\top K^{1/2}w$ with high probability using the sub-Gaussianity of $w$ and the union bound over a finite cover of the unit ball in $R^r$. Similarly, (4) follows from the sub-Gaussianity of $w$ and the union bound over $\pm e_i$ for each $i\in\mathcal{I}$. Finally, (5) follows from a more elaborate chaining argument that frames $(e_i^\top Kw)_{i\in\mathcal{I}}$ as a sub-Gaussian process and uses the Lipschitzness of $K$ to control its entropy integral.
>
> We will also add summaries and high-level discussions of theoretical takeaways. For example, in Sec. 6.1, we will highlight that a standard way to summarize the discriminating power of a test is through its detectable separation rate, the smallest MMD separation between P and Q detectable with power at least $1-\beta$. Standard quadratic-time MMD tests have a minimax-optimal separation rate of order $1/\sqrt{\min(m,n)}$ (Kim & Schrab, 2023, Thm. 8), and our Thm. 4 shows that CTT matches this rate up to an inflation factor $R_k/2^g$ depending on the eigenvalue decay of the kernel. In other words, CTT provides minimax-optimal detectable separation in near-linear time whenever $\textup{rank}_{\epsilon}(K)$ is polylog(n) for $\epsilon=$polylog(n). As a practical concrete example, we derive the first power guarantees for deep kernel CTT in Cors. 3 and 4; these ensure that CTT can match the power of quadratic-time deep kernel testing in near-linear time. The original CTT analysis used the RKHS covering number bounds of Dwivedi & Mackey (DM) instead of kernel matrix eigenvalues to bound compression error, yielding less sharp results. For example, substituting DM bounds into our deep kernel analysis would yield a larger, order $\log^{3d/4+2}(n/\widetilde \beta)$ inflation factor that does not adapt to the intrinsic manifold dimension.
>
> We will also add a brief summarization of the methods analyzed in the revision and follow the reviewer’s other suggestions to improve clarity.
>
> **Tightness:** In the revision we will highlight that our Thm. 1 guarantee (5) with GS-Thin matches the minimax lower bound of Phillips & Tai (Thm. 3.1) for KMS coresets on $R^d$; that our Thm. 3 matches the minimax lower bound of Cha et al. (Thm. 4.5) for reordered SGD algorithms up to the epsilon rank parameter r; and that, by Kim & Schrab, 2023 (Thm. 8), the separation rate of our Thm. 4 is minimax optimal up to the inflation factor $R_k/2^g$ and that of Cors. 3 and 4 is minimax optimal up to log factors.
>
> We will also clarify that the observed accelerated convergence rate of LKH-SGD over the standard slow SGD rate of RR provides a direct verification of the Thm. 3 guarantee, while the improved power-runtime trade-off of CTT over the standard Monte Carlo tradeoff of subsampling provides a direct verification of the Thm. 4 guarantee.
>
> Following the reviewer’s advice, we have also conducted two new analyses to explicitly measure the approximate low-rankness of our matrices and thereby gauge the relevance of our guarantees. In the CTT setting of Sec. 6.2, we find that the empirical inflation factor $R_k = O(log^5(n))$ owing to approximate low-rankness: see https://imgur.com/a/hmAVQIU. In the LKH experiment setting of Sec. 5.2, we find that the stochastic gradient matrices $X_k$ have rapidly decaying singular values and a median $\epsilon_k^*$-rank of 10: see https://imgur.com/a/MyA7gR1. We will investigate whether another simple experiment could provide further insight into bound tightness.

---

> > ### Comment · Reviewer_yDej · 2025-04-04
> >
> > I thank the authors for their response with clarifications, additional discussions/experimental results, and revision plans.  I believe this paper would be a nice contribution to the ICML community with the promised revisions.  With the proviso that the authors would make appropriate revisions, I am willing to happily increase my evaluation rating from 3 to 4.

---

### Official Review · Reviewer_fAVp · 2025-03-27

**Overall Recommendation:** 5

**Summary:**

This paper presents a new analysis of sub-Gaussian thinning algorithms based on a low-rank assumption. It provides theoretical guarantees that apply to many data distributions and kernel functions, in contrast to previous work with more limited applicability and poor dimension dependence. The key insight is that high-quality dataset summarization can be achieved whenever the kernel or data matrix is approximately low-rank, enabling efficient identification of representative points. The authors introduce practical sub-Gaussian thinning methods. They apply them in three key applications with good performance: approximating attention in transformers, accelerating stochastic gradient descent, and distinguishing distributions using deep kernels.

**Claims And Evidence:**

The answer is a definite yes! The paper is very neatly organised, making the claims particularly clear and easy to follow. There is convincing evidence for all theoretical claims, with rigorous proofs in appendix.On top of the main theoretical results presented in section 3, the authors specify these in three quite diverse applications in the last three sections: approximating attention in transformers, accelerating stochastic gradient descent, and distinguishing distributions using deep kernels. Specifics theoretical results are proven in each of them, and numerical experiments are proposed as well. These evaluation criteria make sense, and I find the wealth and diversity of application quite impressive. I'm not an expert in these applications but found them interesting and informative.

**Essential References Not Discussed:**

Relevant literature is covered to a good extent. I am not familiar enough with the thinning literature to make a definite judgement.

**Experimental Designs Or Analyses:**

As said earlier, I am not much familiar with the applications of section 4 to 6. I did not check the soundness of the experimental designs or analyses, but all seemed very reasonable to me. A quick look at the code and some of the Python notebooks within the suppl materials convinced me that the experiments were conducted very thoroughly.

**Methods And Evaluation Criteria:**

On top of the main theoretical results presented in section 3, the authors specify these in three quite diverse applications in the last three sections: approximating attention in transformers, accelerating stochastic gradient descent, and distinguishing distributions using deep kernels. Specific theoretical results are proven in each of them, and numerical experiments are proposed as well. These evaluation criteria make sense, and I find the wealth and diversity of application quite impressive. I'm not an expert in these applications, but found them interesting and informative.

**Other Comments Or Suggestions:**

Just a few typos:
- last eqn page 12: what is meant by "surely"?
- the sentence starting with "Since" in App C.4 misses an end.

**Other Strengths And Weaknesses:**

Strengths:
- a paper thoroughly written;
- very solid and comprehensive theoretical results: every proposed algorithm comes with its own approximation/convergence guarantees and sub-Gaussian parameter \nu;
- the code provided seems well presented and complete (from skimming through it);
- applications and experiments are well motivated and show the interest of the proposed thinning algorithms.

Weakness
- I honestly do not see much to say as weaknesses. As my score and evaluation indicate, I am clearly championing the paper. I do feel, however, that the 8-page conference format makes the presentation somewhat compressed. A journal format with more space could have better served the exposition. That said, the authors make excellent use of content division between the main text versus the appendix.

**Questions For Authors:**

I don't think the response to this question would affect my score, but here it is anyway:
The authors partly characterize the performance of their thinning algorithms using sub-Gaussian parameters. These can in principle be optimized (ie, minimized) to yield a so-called optimal (i.e., minimal) variance proxy $\nu_opt^2$. For many standard distributions, this optimal variance proxy parameter is known. I was curious whether the sub-Gaussian parameters derived for the proposed thinning algorithms are optimal in this sense. Are there technical reasons preventing such a result? Establishing optimality—or proving lower bounds showing that the guarantees cannot be improved—would be a valuable addition and strengthen the theoretical contribution of the paper.

**Relation To Broader Scientific Literature:**

See below.

**Theoretical Claims:**

I set out to carefully check all of the proofs, as I am both familiar with and interested in sub-Gaussian properties, but did not manage to read every proof in full detail; I reviewed the main arguments through Appendix D with reasonable care. The parts I read were well-written and convincing.

---

> ### Author Rebuttal · Authors · 2025-03-31
>
> We thank the reviewer for championing this submission and are delighted that the reviewer found the paper “clear and easy to follow,” the theoretical results “solid and comprehensive,” and the “wealth and diversity of application quite impressive.”
>
> Thank you also for the interesting question concerning variance proxies. If we understand correctly, the question is whether for each thinning algorithm we have identified the smallest possible sub-Gaussian constant $\nu$. We have not yet had a chance to verify this, but one tractable lower-bounding strategy would be to compute (or lower-bound) each algorithm’s variance parameter $\sigma^2 = \sup_{u} E[u^\top K(p_{in} - p_{out})(p_{in} - p_{out})^\top K^\top u] / u^\top K u$ as a function of the sample size, $\delta$, and the kernel matrix. It seems at least plausible that the derived sub-Gaussian and variance parameters would match in the worst-case setting.
>
> While we do not yet have a precise answer to this question, we can currently comment on the rate optimality of our sub-Gaussian constants. Specifically, Thm. 3.1 of Phillips & Tai implies that any thinning algorithm must incur at least $\Omega(\sqrt{d}/n_{out})$ KMS error for some dataset in $R^d$ and many common kernels. Meanwhile, our Prop. B.6 and Thm. 1 imply that GS-Thin has $\nu = O(1/n_{out})$ and hence KMS $O(\sqrt{d}/n_{out})$. Thus, GS-Thin has a minimax rate-optimal sub-Gaussian constant, and no thinning algorithm can have sub-Gaussian constant $\nu = o(1/n_{out})$.
>
> Finally, thank you for flagging our typo. In the revision, we will replace "Since" with "Moreover" and "surely" with "with probability 1."

---

### Decision · Program_Chairs · 2025-05-01

**Decision:**

Accept (poster)

**Comment:**

The paper gives theoretical guarantees for  thinning (or subsampling data) algorithms that can adapt to low-rank structure. The bounds assume sub-Gaussian thinning, which is a broad class.  I enjoyed particularily the range of applications, from subsampling key-query pairs in an attention layer, to reordering the training data for SGD, and finally distinguishing two distributions in nearly linear time. Because the reviewers were also very positive, I recommend the paper be accepted.

The reviewers raised some issue around clarity, and that the paper is hard to access for non-experts. So I encourage the authors to make the paper more didactic, so a wider audience can appreciate their results. Though I know this is challenging, given the limited space, and all the results you would like to present. Furthermore, the assumptions in Theorem 3 are a bit restrictive, since you assume that loss is both Lipschitz, smooth and PL. Though I understand these assumptions hold for logistic regression with no L2 regularization, It would be nice to see if you could establish a proof in one of the more broad classes such as the smooth setting (smooth+PL) or non-smooth setting (Lipschitz + PL)